

# On the dynamics of free-fermionic $\tau$-functions at finite temperature

**Daniel M. Chernowitz[1] and Oleksandr Gamayun[2,3]⋆**

**1** Institute for Theoretical Physics, University of Amsterdam,
PO Box 94485, 1090 GL Amsterdam, The Netherlands
**2** Bogolyubov Institute for Theoretical Physics, 03143 Kyiv, Ukraine
**3** Faculty of Physics, University of Warsaw, ul. Pasteura 5, 02-093 Warsaw, Poland

⋆ oleksandr.gamayun@fuw.edu.pl

## Abstract

In this work we explore an instance of the $\tau$-function of vertex type operators, specified in terms of a constant phase shift in a free-fermionic basis. From the physical point of view this $\tau$-function has multiple interpretations: as a correlator of Jordan-Wigner strings, a Loschmidt Echo in the Aharonov-Bohm effect, or the generating function of the local densities in the Tonks-Girardeau gas. We present the $\tau$-function as a form-factors series and tackle it from four vantage points: (i) we perform an exact summation and express it in terms of a Fredholm determinant in the thermodynamic limit, (ii) we use bosonization techniques to perform partial summations of soft modes around the Fermi surface to acquire the scaling at zero temperature, (iii) we derive large space and time asymptotic behavior for the thermal Fredholm determinant by relating it to effective form-factors with an asymptotically similar kernel, and (iv) we identify and sum the important basis elements directly through a tailor-made numerical algorithm for finite-entropy states in a free-fermionic Hilbert space. All methods confirm each other. We find that, in addition to the exponential decay in the finite-temperature case the dynamic correlation functions exhibit an extra power law in time, universal over any distribution and time scale.

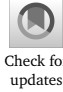

# 1 Introduction

The last decades have witnessed a huge success in the understanding of correlation functions in one-dimensional quantum systems [1, 2]. The first key idea came from the point of view of perturbation theory, which has been the identification of the important Feynman diagrams, upon which they can be resummed. Secondly, we have been fortunate in that these predictions could be tested against integrable models that, in principle, allow for full non-perturbative solutions [3]. The third and the most important ingredient was the formulation of the *Luttinger liquid* as a universal effective theory [4].

All these methods are inherently based on the ground state properties and describe low-energy physics. Conversely, in recent years the physics of highly excited states or non-equilibrium dynamics in one-dimensional systems has attracted more and more attention, both due to the tremendous advance in relevant experimental techniques, in particular with ultracold atoms [5–7], and novel theoretical concepts such as the quench action [8, 9], generalized hydrodynamics (GHD) [10–12], and others.

For integrable systems the most natural approach for obtaining correlation functions is via the spectral sum of the *form-factors* (matrix elements of physical operators), as they are known thanks to integrability. To implement this straightforward approach effective numerical

methods were developed [13] and successfully applied to various physical systems [14–17]. Unfortunately, these numerical methods cannot be directly applied to highly excited states, as the number of terms to be taken into account in the form-factor series grows exponentially in system size, for finite temperature. Some progress was made in e.g. [18].

Direct summation is possible only for free fermionic models. Therefore, partial summation methods were developed that allow us in particular to extract large time and space separation asymptotic behavior of the correlation functions [19–24]. When one focuses on partial summation of the excitations around the Fermi sea, one obtains an asymptotic that reproduces the prediction of the Luttinger model. As for dynamical correlation functions, new asymptotic terms have appeared, produced by single particle excitations that probe highly excited parts of Hilbert space. The corresponding effective field theory was dubbed a *non-linear* Luttinger liquid [25–27].

A straightforward generalization of these methods to thermal (finite-entropy) or highly excited states is not known. However, in order to tackle the situation, alternative methods were developed, less universal ones that relied on the detailed structure of the form-factors in their particular integrable theories. For instance, the finite temperature correlation functions of many observables in integrable lattice models of Yang–Baxter type can be evaluated by means of the Quantum Transfer Matrix (QTM) [28]. With this technique, the notion of the thermal form-factor was introduced [29], which turned out to be instrumental in the asymptotic analysis of two-point functions [29–31]. Thermal form-factors also appear naturally in the context of Integrable Quantum Field Theory [32–40].

Recently there have been numerous attempts to develop systematic methods to mount partial summations towards finite-entropy correlation functions, based either on the restriction of the spectral sums to a finite number of particle-hole pairs in the Lieb-Liniger and XXZ models [41–45], performing resummation of the most singular parts of the form-factors to describe the low-energy correlation functions in the Ising model [46], or $1/c$ expansion in the Lieb-Liniger model [47]. The whole machinery of the QTM methods was re-enlisted to address correlation functions of the XX spin-chain [48–50]. GHD methods were also adopted for correlation functions on the Euler scale [51] (for a review see [12]). After the generalization of Smirnov's form-factor axioms for thermodynamic states [52] the GHD description of the correlation functions was also successfully reproduced [53, 54].

This is the state of the art in broad strokes. In this paper we explore finite-entropy correlation functions in the free fermion model, however our operators are non-local in the spectral basis. This way, on the one hand we have all the entropic and combinatorial complexity associated with the techniques of direct evaluation of the form-factor series, while on the other hand we can evaluate the series exactly in the thermodynamic limit (TDL) and gauge various numerical approaches against the true answer.

To be more concrete, our main object of interest is the following $\tau$-function, formally defined as an infinite series

$$\tau_{\mathbf{g}}(x,t) := \sum_{\mathbf{k}} |\langle \mathbf{g} | \mathbf{k} \rangle|^2 \, e^{\sum_a \left( ix(g_a - k_a) - it(g_a^2 - k_a^2)/2 \right)}. \tag{1}$$

Here the state of the system $|\mathbf{g}\rangle$ is specified by $N$ distinct single particle momenta $g_a = \frac{2\pi}{L} n_a$. In turn, $\{n_a\}$ is a particular set of distinct integers. For instance, for the ground state, $\{n_a\}$ are adjacent and centered on zero. $L$ is a parameter associated with system size. The summed set $\mathbf{k}$ is specified by the $N$ shifted momenta $k_a = \frac{2\pi}{L}(m_a - \nu)$, $\nu \in [0, \frac{1}{2}]$. The scalar shift $\nu$ plays a central role in this work, as it interpolates between a trivial free-fermionic problem and a globally coupled one. Then summation over all available $\mathbf{k}$ is isomorphic to summation over all possible sets $\{m_a\}$ of $N$ distinct integers. Finally, the form-factor, which in our case is

simply an overlap, is given by

$$|\langle \mathbf{g}|\mathbf{k}\rangle|^2 = \left(\frac{2}{L}\sin(\pi \nu)\right)^{2N}\left(\det_{1\le a,b\le N}\frac{1}{k_a - g_b}\right)^2 .$$

(2)

This results from the inner product of two Slater determinants in position representation, one in the shifted and one in the unshifted fermion basis, see equation (12).

By the 'finite-temperature' (entropy) generalization of the $\tau$-function, we understand the situation where instead of a single state $|\mathbf{g}\rangle$, averaging is performed over a statistical ensemble. We conjecture below (see Sec. (4.2)) that in the TDL, being $N \to \infty$, while $N/L$ remains constant, one may replace the ensemble evaluation by the evaluation of $\tau$ on any thermodynamically large 'thermal' state $|\mathbf{g}\rangle_{\text{th}}$, whose integers are drawn independently according to the density $\rho(g)$ that defines the ensemble. The full thermal $\tau$-function can be presented in terms of a *Fredholm determinant* [55]

$$\tau(x,t) = \det_{\mathbb{R}^2}(\mathbb{1} + \hat{K}_\rho)$$

(3)

of the operator $\hat{K}_\rho$ acting on functions $y \in \mathcal{L}^2(\mathbb{R})$ as the convolution $\hat{K}_\rho y(p) = \int K_\rho(p,q)y(q)dq$, in turn specified by the kernel

$$K_\rho(q,p) := \sqrt{\rho(p)}\frac{e_+(q)e_-(p) - e_-(q)e_+(p)}{q - p}\sqrt{\rho(q)}$$

(4)

with

$$e_-(g) := e^{\frac{i}{2}xg - \frac{i}{4}tg^2}, \qquad e_+(g) := -\frac{\sin^2(\pi \nu)}{\pi e_-(g)}\left[\cot(\pi \nu) + i\,\mathrm{erf}\left(\frac{(i+1)(x-gt)}{2\sqrt{t}}\right)\right].$$

(5)

Why study this? Such kernels are found in many physical systems, we list here those of which we are aware.

Firstly, in the correlation functions of hardcore one-dimensional anyons [56–60], the variable $\nu$ plays the role of the anyonic exchange parameter. Secondly, similar kernels appear for the problem of the mobile impurity propagating through a gas of free fermions [61–64], when the coupling constant is infinite, $\nu$ can be related to the total momentum of the gas. Exactly such determinants as in (3), (4) and (5) can also be obtained as the correlation functions of Jordan-Wigner strings, as found in [65]. This last observation is a main motive of our investigation. In general, spinful interacting fermions in the infinite interaction limit are also often described in such terms [66–75]. In this case the correlation is found after taking the integral over $\nu$ of said determinant, which reflects the fact that after the spin-charge separation one has to 'average' over all possible anyonic phase statistics before obtaining effective spinless fermions. Finally, in section (2) we show a physical systems that produces our model even on the level of the form-factor series (1), as opposed to the previous examples that just share the same kernel in the TDL.

On to the properties of the object itself. The kernel in (4) is of a generalized time-dependent type introduced earlier in references [76, 77]. The Fredholm identity is not well suited to evaluate the large $x$ and $t$ asymptotics. The exponents in (5) oscillate rapidly, and many points are needed to correctly find the resulting interference, making it computationally challenging. Therefore, it is interesting to have alternative expressions for asymptotics at large times and distance from the origin. This has been done for comparable observables, by means of the Riemann-Hilbert method. An instance of the zero temperature case was considered in [73,77], of the finite temperature static case was solved in [78], and in [79, 80] one encounters a calculation for finite temperature dynamics of a similar kernel.

In this paper we explain, among other things, how the asymptotic can be obtained directly from the form-factor series. For zero temperature we perform the microscopic resummations of the *soft modes*, excitations around the Fermi surface, using methodology developed in references [21, 23, 24]. We relate this calculation to the bosonization approach, where the $\tau$-function is recognized as a vertex-type operator. We conjecture the asymptotic behavior when the critical point in momentum, $q = x/t$ coincides with the Fermi momentum. As for finite temperature, we employ a method of effective form-factors inspired by [81]. We modify this work appropriately in order to account for dynamics, and the spatial continuum, as opposed to particles constrained to the lattice, which is the case in [81]. It turns out that the phase shift of the effective fermions is momentum dependent, and contains a discontinuity which requires regularization around momenta $|q - x/t| \sim O(1/\sqrt{t})$. We could not identify the appropriate form of the regularization function, but show that it influences only the overall constant factor in the asymptotic, which moreover is empirically close to unity. We argue that the presence of the discontinuity leads to an additional power law scaling in time, which seems always to be present in the time-like region at finite temperature (151). This type of behavior was also observed for similar observables, thanks to the Riemann-Hilbert treatment of such determinants, for instance in [79, 80, 82].

The final method by which we evaluate the $\tau$-function is numerics. Instead of simply brute-forcing the calculation, which would prove very inefficient, we explore the summation over a basis of Hilbert space in an informed way, allowing results of earlier states in the series (1) to guide the direction in which new states are chosen. This idea is not new, techniques such as these have been in use for more than a decade. We refer specifically to the ABACUS algorithm for dynamical correlation functions of integrable models [13]. In this case, ABACUS is thought to be ill-suited for the task: it produces states at low temperatures, and entropies, where mostly the Fermi sea is filled and *descendents* are chosen by placing small numbers of particles at macroscopic distances. In our case, holes are dense in the Fermi sea and local, soft mode shufflings are both very important for collecting operator weight, and very numerous at any filling profile. A new algorithm is structured to naturally group microscopically defined states together that share a macroscopic density profile, which is more expedient for finite-temperature calculations. Although we can only access modest particle numbers, the obtained observables already strongly resemble the exact TDL. Little about the algorithm is specific to this observable. It is readily modified to scour out any thermal operator described in terms of a basis isomorphic to the free-fermionic, such as the repulsive Bose gas.

The structure of the paper is as follows: in the next section, 2, we describe briefly a possible physical setup that directly leads to the series (1). It arises when we consider the well-known Aharonov-Bohm effect [83], but as a many-body problem. After this, in section 3, we motivate why the $\tau$-function is surprisingly non-trivial to calculate, as the number of terms needed to achieve an appreciable portion of its sum scales dramatically in the TDL. This phenomenon is called the *Orthogonality Catastrophe* (OC), a term coined by Anderson [84]. Then we move to the first main result of the present work: in subsection 4.1 an exact, analytic resummation of the $\tau$-function in terms of a regular determinant for finite size states, then augmented to a Fredholm determinant for a thermodynamically large state. In subsection 4.2 we generalize to a Fredholm with infinite support in order to describe statistical ensembles of states. Section 5 makes a slight detour in order to verify the specific case of zero temperature in an alternative fashion: we discover in this case we can sum over the *soft mode* excitations around the Fermi surface explicitly using bosonization. Section 6 introduces one of the other main innovations: an efficient approximation to the $\tau$-function, termed the quasi-$\tau$-function, that in the large time limit can be found as a simple integral of elementary functions, elucidating the scaling behavior to be an exponential times a power law in $t$. Section 7 changes gear entirely, and describes the aforementioned tailor-made algorithm used to numerically calculate the $\tau$-

function. This algorithm is used to corroborate the results preceding it, at a finite system size. The conclusion and outlook are found in section 8.

## 2 Aharonov-Bohm Quench

Before proceeding to the computation of the $\tau$-function defined in equation (1), we illustrate a simple physical setup where it appears naturally. It is a magnetic quench in the system that exhibits the Aharonov-Bohm effect [85]. Namely, consider a continuous one-dimensional circular loop of length $L$ in the horizontal plane with noninteracting fermions (for instance electrons, all in a spin-up state) living on it. There is no spin-flip mechanism in this model, so they are essentially spinless. Through the loop there is a constant vertical magnetic field of strength $\mathcal{B}$, perhaps produced by a solenoid coil. The field only extends to radius $r_c < \frac{L}{2\pi}$, as in figure 1.

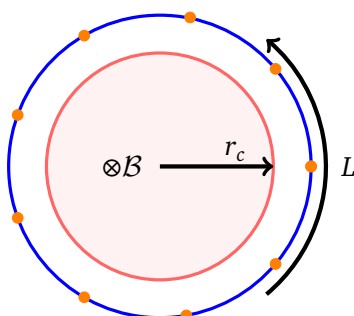

Figure 1: Setup for the Aharonov-Bohm effect. A cylindrical region in the center has a constant magnetic field $\mathcal{B}$ going into the page. Its circular intersection is shaded red. Strictly outside it, our loop of circumference $L$ lies in in the horizontal plane, in the page. On the loop, a number of fermions are schematically indicated in orange.

The resulting vector potential, in cylindrical coordinates $(r, z, \varphi)$ is as follows:

$$\mathcal{A}_r = \mathcal{A}_z = 0, \quad \mathcal{A}_\varphi = \begin{cases} \frac{\mathcal{B}r}{2}, & r \le r_c \\ \frac{\mathcal{B}r_c^2}{2r}, & r > r_c. \end{cases} \tag{6}$$

The fermions couple to the field with coupling strength $e_c$. We change coordinates to the position on the loop $x = \frac{L}{2\pi}\varphi$. Due to the magnetic flux $\Phi = \pi r_c^2 \mathcal{B}$, the fermions experience the following Hamiltonian:

$$H = \frac{1}{2m_e}\left(-i\hbar\frac{\partial}{\partial x} - \frac{e_c \Phi}{L}\right)^2. \tag{7}$$

The single particle eigenstates of this system are plane waves, with quantized momentum $k$. The quantization follows from the periodic boundary condition $e^{ikL} = e^{-i2\pi \nu}$, the RHS of which is the phase picked up by a particle traveling once around the loop.

$$\phi(k; x) = \frac{1}{\sqrt{L}}e^{ikx}, \quad k = \frac{2\pi}{L}(n - \nu), \quad n \in \mathbb{Z}. \tag{8}$$

We have defined $\nu := \frac{2\pi\Phi e_c}{\hbar}$, and posit it to lie on the interval[1] $\nu \in [0, \frac{1}{2})$. This constant, non-integer momentum shift will play a central part in this paper. The energy of such a single fermion is $E(k) = \frac{\hbar^2}{2m_e}k^2$, and in the following we work in units in which $\hbar^2 = m_e$, so $E = k^2/2$.

---

[1] Any other $\nu \in \mathbb{R}$ can be found by translation and reflection symmetry.

Conventionally, in such setups, the field strength $\mathcal{B}$ is fixed and we are not interested in many body effects. We change this perspective. We consider $N$ of these free fermions together. Due to the Pauli exclusion principle, they have distinct momenta collected in a vector $\mathbf{k} = \{k_1 < \ldots < k_N\}$ and their many body wavefunction is a Slater determinant

$$|\mathbf{k}\rangle = \frac{1}{\sqrt{N!}} \det_{1 \le a,b \le N} \phi(k_a; x_b). \tag{9}$$

The multi-particle energy and momentum are the sums over their single particle values.

Furthermore, the system is prepared with zero magnetic field, such that also $\nu = 0$, and at the start of our virtual experiment, $\mathcal{B} > 0$ is switched on, quenching the system to a new eigenbasis with $\nu > 0$. In other words, dynamics suddenly begin according to the new Hamiltonian, while the system finds itself in a thermal ensemble (described by a distribution $\rho(q)$, see section 3) of eigenstates of the original Hamiltonian. Note that this ensemble, for $T = 0$, has all its weight in the unique ground state. When possible, symbols $p, q, g$ and derivates such as $\mathbf{p}, p_a$ refer to the basis with $\nu = 0$ and $k, h$ to that with $\nu > 0$.

The focal point of the present paper is the $\tau$-function. It is defined as a kind of generalized *Loschmidt echo*, or alternatively as an expectation value of a translation operator. Such an expectation value is signified by brackets $\langle \ldots \rangle$, and refers to the ensemble, given by context. The $\tau$-function depends explicitly on space and time $(x, t)$, and implicitly on system length $L$, shift $\nu$, temperature $T$, and on particle number $N$ explicitly or through the chemical potential, $\mu$.

For appropriate Hamiltonian $H$ and momentum operator $P$, $e^{-iHt}$ translates the state in time by $t$ and $e^{iPx}$ in space by $x$. In terms of these translation operators, the $\tau$-function is the following

$$\tau(x, t) := \left\langle e^{-iH_\nu t + iP_\nu x} e^{iH_0 t - iP_0 x} \right\rangle, \tag{10}$$

where $H_\nu$ is the quenched Hamiltonian and $P_\nu$ the momentum operator, which commute. The subscript indicates that here the field is engaged. The pair $(H_0, P_0)$ are the same for zero field, and are generally diagonalized by the state the system is in. Their addition is a gauge choice, but one which will appear to be natural later on. The modulus of $\tau(0, t)$ in some state $|\mathbf{g}\rangle$ is equal to a Loschmidt echo [86], defined as the RHS of

$$\left| \tau_{\mathbf{g}}(0, t) \right|^2 = \left| \langle \mathbf{g} | e^{-itH_\nu} e^{itH_0} | \mathbf{g} \rangle \right|^2. \tag{11}$$

Thus the $\tau$-function may be thought to measure recurrence after a time $t$, while also rotating the loop onto itself over a distance $x$. Ensemble averages are obtained by convex addition of the constituent states, due to linearity of expectations, we discuss this in details in section 4.2.

Equation (10) can be expanded as a sum by inserting a resolution of unity in the shifted fermion basis. The final ingredient is then the overlap between a shifted and unshifted state. From (9),

$$
\begin{aligned}
\langle \mathbf{g} | \mathbf{k} \rangle &:= \frac{1}{N!} \epsilon_{a_1, \ldots, a_N} \epsilon_{b_1, \ldots, b_N} \int_0^L \prod_m dx_m \, \phi^*(g_{a_m}; x_m) \phi(k_{b_m}; x_m) \\
&= \det_{1 \le a,b \le N} \left( \int_0^L dx \, \phi^*(g_a; x) \phi(k_b; x) \right).
\end{aligned}
\tag{12}
$$

The second equality uses the simultaneous expansion of the determinant over columns and rows, which allows the integral to factorize over $m$. The elements of the final determinant are

single particle overlaps. From (8),

$$\int_0^L dx\, \phi^*(g;x)\phi(k;x) = \frac{1}{L}\int_0^L dx\, e^{i(k-g)x} = \frac{1}{iL(k-g)}\left(e^{-i2\pi\nu}-1\right). \tag{13}$$

Using quantization conditions $e^{igL}=1$ and $e^{ikL}=e^{-i2\pi\nu}$. Then finally, the modulus of the overlap is given by (2).

The expression above illustrates the appeal of this model: in other one-dimensional (integrable) many body theories, many observables of interest have been formulated in terms of a similar determinant. We believe this toy model and operator has enough structure to be interesting as a stepping stone towards, for instance, the Lieb-Liniger Bose gas, while still being exactly solvable. The model was inspired by Anderson's orthogonality catastrophe. In the next subsection, we will consider the scaling of these overlaps for $|\mathbf{g}\rangle$ at (non-)zero temperature. This will inform us why this model poses interesting questions already in the ground state, and why it becomes exponentially more challenging at a finite temperature. This means, numerically, we can never perform a full sum over the infinite set of all $\mathbf{k}$. Our approach involves judiciously choosing the terms that collectively hold as much of the weight, or overlap, as possible. Specifically,

$$\tau_{\mathbf{g}}(0,0) = \sum_{\mathbf{k}}' |\langle \mathbf{g}|\mathbf{k}\rangle|^2 = 1 \tag{14}$$

and truncating the sum results in some measure in $s \in [0,1)$ indicating the quality of our approximation. We sometimes refer to this quantity as the *sum rule*.

As an aside, we mention shortly another realization of the $\tau$-function (1): as a generating function for the density-density correlation function in the Tonks-Girardeau gas, which also closely related to the density *formation probability*. For an elaboration, see [22]. The Tonks-Girardeau is the $c\to\infty$ limit of the Lieb-Liniger model of repulsive $\delta$-interacting bosons. The latter is introduced shortly in section 7.4, where it is used in the algorithm developed for this paper. As can be seen from the Bethe equations (163), for infinite $c$, the Lieb-Liniger eigenstates have free-fermionic momenta and wave-functions.

We may adapt the Bethe equations by also shifting $n_a \mapsto n_a - \nu$, termed the *Shifted Bethe equations*, defining shifted rapidities $\mathbf{k}$. Understanding the symbols to temporarily represent normalized Lieb-Liniger states and momenta, we can re-interpret the $\tau$-function in (1) as an operator of the Lieb-Liniger state[2] with rapidities $\mathbf{g}$.

In the Lieb-Liniger model, a distribution function $\rho(q)$ must satisfy

$$2\pi\rho(q) - \int dp\, \frac{2c\,\rho(p)}{(p-q)^2+c^2} = 1. \tag{15}$$

With such a function, we can identify the form-factor

$$\langle \mathbf{g}| e^{2\pi i\nu \int_0^x dq\rho(q)} |\mathbf{k}\rangle = e^{ix\sum_a(g_a-k_a)}\,\langle \mathbf{g}|\mathbf{k}\rangle. \tag{16}$$

Finally, the density-density correlator from (165), is equal to

$$O_2(x,t) := \left\langle \Psi^\dagger\Psi(0,0)\,\Psi^\dagger\Psi(x,t)\right\rangle = -\frac{1}{8\pi^2}\partial_x^2\partial_\nu^2\tau(x,t)\bigg|_{\nu=0}. \tag{17}$$

For the Tonks-Girardeau gas, the $\tau$ on the RHS is indeed the exact same $\tau$-function as in (1).

---

[2]The only difference is no factor $\frac{1}{2}$ in the energy, and thus this factor missing in front of $t$ in the exponent. In [22], $\tau$ is called $\mathcal{Q}_N^\kappa(x,t)$ and $\nu = -i\beta/(2\pi)$.

# 3 Orthogonality Catastrophe

In this section, we elucidate why the problem of calculating the $\tau$-function from the form-factor series representation is so challenging. It has to do with the number of terms required to reach an appreciable sum rule, or portion of the overlap of the initial state, and how this number scales with system size. Let us first set our sights on the ground state, a Fermi sea, where the integers $n$ in the single particle states in (8) are adjacent and centered on the origin (see also figure 2). It is clear that such a choice of integers results in the lowest energy for a given system size. We have a picture that the overlap between shifted and unshifted states is determined by their similarity of occupation in momentum space. Indeed, empirically and mathematically, the closer the momenta are, the larger the overlap. In the limit $\nu \mapsto 0$, this overlap becomes the familiar Kronecker delta function for a free fermion basis. Then naively, one would expect most of the weight of the unshifted ground state to be found in the shifted ground state. Let now $|\mathbf{g}\rangle$ and $|\mathbf{h}\rangle$ be the $N$-particle ground states of $H_0$ and $H_\nu$, respectively.

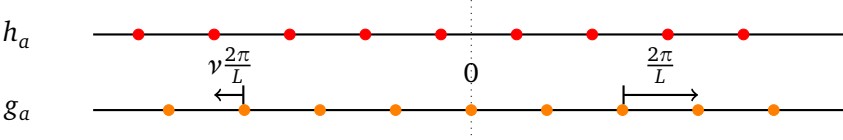

Figure 2: Momentum space picture of unshifted and shifted $N = 9$-particle ground states $|\mathbf{g}\rangle$ and $|\mathbf{h}\rangle$, respectively. $g_a, h_a$ are the single particle momenta.

In some sense, as $N$ increases, the similarity of these ground states can be thought to increase. However, we can show that the ground state overlap vanishes as a power law in the TDL. This was coined by Anderson as the *Orthogonality Catastrophe* [84]. Over the years, there were many attempts to put bounds on the exponent of decay, see e.g. [87]. We compute the scaling and prefactor of the ground state overlap exactly. The vanishing of this overlap is important because any other state $|\mathbf{k}\rangle$ has an even smaller overlap[3] with $|\mathbf{g}\rangle$, so naively, there is no way, even at zero temperature, to efficiently truncate the sum in $\tau$. As a foreshadowing, if we pick a suitable scheme to find microscopic realizations of some excited state with increasing $N$, the diagonal overlap vanishes exponentially in system size, yet more catastrophic.

The ground state overlap (2) is a *Cauchy Determinant*, and thus admits a special factorized identity

$$
\begin{aligned}
|\langle \mathbf{g}|\mathbf{h}\rangle|^2 &= \left(\frac{2}{L}\sin(\pi\nu)\right)^{2N} \frac{\prod_{a<b}(g_b - g_a)^2 \prod_{a<b}(h_b - h_a)^2}{\prod_{b,a}(g_b - h_a)^2} \\
&= \left(\frac{2}{L}\sin(\pi\nu)\right)^{2N} \frac{\prod_{a<b}(\frac{2\pi}{L}b - \frac{2\pi}{L}a)^2 \prod_{a<b}(\frac{2\pi}{L}(b-\nu) - \frac{2\pi}{L}(a-\nu))^2}{\prod_{a,b}(\frac{2\pi}{L}b - \frac{2\pi}{L}(a-\nu))^2} \\
&= \left(\frac{\sin(\pi\nu)}{\pi\nu}\right)^{2N} \prod_{a\neq b}^{N} \left(\frac{b-a}{b-a-\nu}\right)^2 .
\end{aligned}
\tag{18}
$$

These products can be simplified using Barnes G-functions defined by the identity $G(z+1) = \Gamma(z)G(z)$, and *Euler's reflection formula*

$$
\Gamma(1-\nu)\Gamma(1+\nu) = \frac{\pi\nu}{\sin(\pi\nu)},
\tag{19}
$$

---

[3]One can intuit that the diagonal, $\{m_a\} = \{n_a\}$ maximizes the overlap over sets $\{m_a\}$, from expressions such as (2), and this empirical fact has been confirmed numerically on millions of states.

which brings to a concise form

$$|\langle \mathbf{g}|\mathbf{h}\rangle|^2 = \left( \frac{G(1-\nu)G(1+\nu)G^2(N+1)}{G(N+1-\nu)G(N+1+\nu)} \right)^2. \tag{20}$$

The asymptotic behavior of the overlap can be easily deduced from the known asymptotic behaviour of the Barnes function [88].

$$\log G(z+1) = \left( \frac{z^2}{2} - \frac{1}{12} \right) \log z - \frac{3z^2}{4} + \frac{z}{2} \log 2\pi + \frac{1}{12} - \log A_{GK} + \mathcal{O}\left( \frac{1}{z^2} \right), \tag{21}$$

as $z \to \infty$. Here $A_{GK}$ is the *Glaisher-Kinkelin* constant, whose value is immaterial as it drops out neatly. We expand (20) to find

$$|\langle \mathbf{g}|\mathbf{h}\rangle|^2 = \frac{G^2(1-\nu)G^2(1+\nu)}{N^{2\nu^2}} \left( 1 + \mathcal{O}\left( \frac{1}{N^2} \right) \right). \tag{22}$$

This way, the orthogonality catastrophe takes the form of a power law in $N$ vanishing of the ground state overlap, with exponent $-2\nu^2$.

In contrast with the zero temperature overlap, an excited state diagonal overlap vanishes even more rapidly as system size increases. In order to make this claim more robust, we must explain how we compare excited states at various system sizes. Let there be some real distribution function $\rho(q) \in [0, 1]$ that describes the probability that any single particle momentum state, for the momentum $q$ available under quantization, is occupied. Many-body states can be obtained by independently filling the single particle momenta at $q$ with probability $\rho(q)$, thus leaving that state empty with probability $1 - \rho(q)$. This is the philosophy of the grand canonical (Gibbs) ensemble. In practice, $\rho(q)$ will be the Fermi-Dirac distribution

$$\rho(q) = \frac{1}{e^{(q^2/2-\mu)/T} + 1}, \tag{23}$$

with the chemical potential $\mu$ fixed by the demand[4]

$$\int dq \, \rho(q) = 2\pi \frac{N}{L}. \tag{24}$$

In words, in units of the momenta-spacing $\frac{2\pi}{L}$, the area under the distribution must be $N$: by sampling each available momentum independently, we obtain an expected $N$ particles in the many-body state. The function $\rho(q)$ is thus scale independent, and $\mu$ depends only on the temperature $T$ and the density $N/L$. In the TDL, we take $N, L \to \infty$ with density fixed, sampling becomes more dense, and the microscopic state filling resembles the generating distribution more and more under appropriate coarse graining.

This all means that microscopically, we are not in any specific state. In fact, later on we will work in the classical stochastic superposition of all these states, the Gibbs ensemble. For now, any argument about the behaviour of the overlap must necessarily be a probabilistic one.

Again, let the diagonal overlap $\langle \mathbf{g}|\mathbf{h}\rangle$ be that where $\langle \mathbf{g}|$ and $|\mathbf{h}\rangle$ share the same quantum numbers $\{n_a\}$, but the integers in $|\mathbf{h}\rangle$ are shifted by $\nu$ w.r.t $\langle \mathbf{g}|$. Referencing (18), this diagonal overlap is given by restoring $j \mapsto n_a \in \mathbb{Z}$,

$$|\langle \mathbf{g}|\mathbf{h}\rangle| = \left( \frac{\sin(\pi\nu)}{\pi\nu} \right)^N \prod_{b>a}^{N} \left( 1 - \frac{\nu^2}{(n_b - n_a)^2} \right)^{-1}, \tag{25}$$

---

[4]In this work, if the integral domain is omitted, it is always taken to be $\mathbb{R}$

where we have also halved the number of factors by multiplying out those with $a \leftrightarrow b$. The name of the game is to estimate how many factors in the product there are for each possible difference $\Lambda := n_b - n_a$. Such a gap appears for each instance where sampling $\rho(\frac{2\pi}{L}n_a)$ and $\rho(\frac{2\pi}{L}(n_a + \Lambda))$ both yielded an occupied momentum. For a sufficiently large sample, we approximate

$$\#_\Lambda \approx \sum_{n \in \mathbb{Z}} \rho\left(\frac{2\pi}{L}n\right)\rho\left(\frac{2\pi}{L}(n+\Lambda)\right), \tag{26}$$

and moreover, let us assume $\rho(q)$ is sufficiently smooth on the scale indicated by $\frac{2\pi}{L} = dq$, convert to an integral over $q = \frac{2\pi}{L}n$, and invoke the Taylor series,

$$\begin{aligned}
\#_\Lambda &\approx \frac{L}{2\pi} \int dq\, \rho(q)\rho\left(q + \frac{2\pi}{L}\Lambda\right) \\
&= \frac{L}{2\pi} \sum_{m=0}^{\infty} \int dq\, \rho(q)\left(\frac{2\pi}{L}\Lambda\frac{\partial}{\partial q}\right)^m \rho(q).
\end{aligned} \tag{27}$$

The leading order in $N, L \to \infty$ is then simply for $m = 0$. This will prove sufficient to find the scaling behavior. Curtailed at $m \leq 1$, the series would be,

$$\#_\Lambda \approx \frac{L}{2\pi} \int dq\, \rho^2(q) + \Lambda \int dq\, \rho(q)\rho'(q) + \mathcal{O}\left(\frac{1}{L}\right), \tag{28}$$

meaning, to leading order in $L$, this multiplicity $\#_\Lambda$ is independent of $\Lambda$. We call it simply $\#_\Lambda = \alpha N$, for some $\alpha \in [0, 1)$. Considering only the first term of (28), we note that $\rho(q) \in [0, 1] \Rightarrow \rho^2(q) \leq \rho(q)$. For $T = 0$, the distribution is a double step function and $\alpha = 1$, but for $T > 0$ $\alpha < 1$, because together with (24), we observe

$$\#_\Lambda < \frac{L}{2\pi} \int dq\, \rho(q) = N. \tag{29}$$

We also artificially continue the upper bound of $\Lambda$ to infinity, as the extra factors are sufficiently close to unit to not spoil the reasoning. Then the overlap becomes

$$|\langle \mathbf{g}|\mathbf{h}\rangle| \approx \left(\frac{\sin(\pi\nu)}{\pi\nu}\right)^N \prod_{\Lambda=1}^{\infty}\left(1 - \frac{\nu^2}{\Lambda^2}\right)^{-\alpha N}, \tag{30}$$

And finally, another miraculous piece fits into this puzzle in the form of a product identity for the sinc-function,

$$\left(\frac{\sin(\pi\nu)}{\pi\nu}\right) = \prod_{m=1}^{\infty}\left(1 - \frac{\nu^2}{m^2}\right), \tag{31}$$

under which the prefactor is the same as the product,

$$|\langle \mathbf{g}|\mathbf{h}\rangle| \approx \left(\frac{\sin(\pi\nu)}{\pi\nu}\right)^{N(1-\alpha)}; \quad \alpha := \frac{L}{2\pi N} \int dq\, \rho^2(q). \tag{32}$$

Considering $\sin(\pi\nu) < \pi\nu$, this overlap vanishes exponentially in system size $N$. We could add corrections to this scaling by including $m > 1$ terms in the expansion. However, the scaling has been checked numerically for 1000 sampled states at $T = \frac{1}{2}, N = L$ of each of $N \in [11, 21, \dots 191]$. Indeed, the scaling was ostensibly exponential after taking the geometric mean[5] inside the sets of 1000 states and plotting these semi-logarithmically against $N$. The

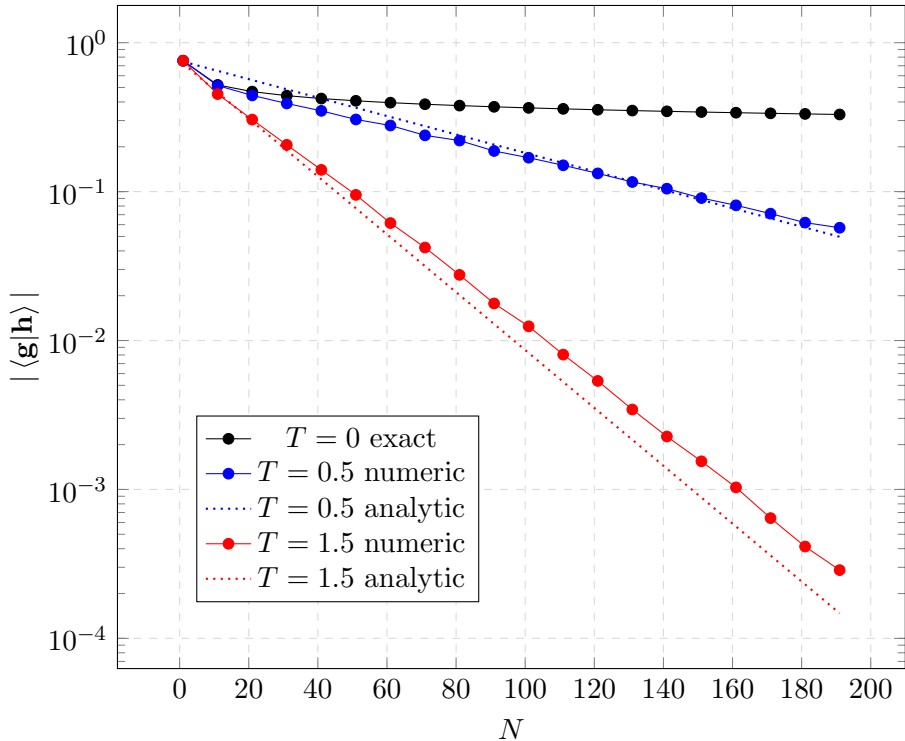

Figure 3: Diagonal overlap scaling plotted against system size on semi-logarithmic axes. Included are the exact values for the ground state, (20), in black. Then for $T = \frac{1}{2}$ in blue and $T = \frac{3}{2}$ in red, the solid disks are the geometric mean of 1000 states sampled numerically from the Gibbs ensemble at that $N$. Compare them to the dotted lines, which are the simple prediction as in (32). It is clear that for finite temperature, scaling is essentially exponential, for zero temperature sub-exponential. $v = 0.4, L = N$.

exponent found just by the analysis in (32) was 7% above the numerical best fit. See figure 3 for an illustration.

Note that this reasoning breaks down exactly at zero temperature because the constant $\alpha = 1$ in that case, the exponential parts cancel, and we must look to higher order terms.[6]

The takeaway is that it is challenging, if not impossible, to resum the thermal $\tau$-function by means of simply identifying the most important terms and neglecting the rest. All terms vanish exponentially in system size. Naive application of a Monte Carlo scheme also does not promise success, as 'uniform' sampling from the infinite Hilbert space is not possible and constrained sampling runs the risk of not collecting enough weight. However, some stochastic methods have been applied to similarly posed thermal observables. See e.g. [89, 90].

## 4 Analytic Calculation of the $\tau$-function

In this section, we present the derivation of the $\tau$-function as a *Fredholm Determinant*. We start with the representation of the $\tau$-function on a given state as a finite determinant and work towards its single state TDL. Then we discuss averaging over the ensemble and argue

---

[5]We are interested in the behavior of the order of magnitude of this statistic. The additive mean would yield the order of magnitude of the largest values. Instead we must average the order of magnitude of our sample: we either take the average of their logarithm, or equivalently, compute the geometric mean.

[6]However, in a completely different language as then also the derivatives of $\rho(q)$ do not exist.

that the modifications can be easily incorporated in the kernel of the Fredholm determinant. Similar derivations were discussed in [61].

## 4.1 Single Eigenstate Case

Our first order of business is to calculate, from microscopics, the $\tau$-function in a a discrete, finite and generic unshifted eigenstate $|\mathbf{g}\rangle$, for which we set $N = L$, and remind the definition of the $\tau_{\mathbf{g}}(x, t)$ given in equation (1)

$$\tau_{\mathbf{g}}(x, t) = \left(\frac{2}{L}\sin(\pi\nu)\right)^{2N} \sum_{\mathbf{k}} \left(\det_{1 \le a, b \le N} \frac{1}{k_a - g_b}\right)^2 e^{\sum_a \left(ix(g_a - k_a) - \frac{i}{2}t(g_a^2 - k_a^2)\right)} \tag{33}$$

with summation over all basis states $|\mathbf{k}\rangle$ of $N$ shifted fermions. By definition, $k_1 < k_2 < \ldots < k_N$, but we may permute them as we see fit: the determinant would change sign under exchange of rows, but its square is invariant. Then we may also augment to an independent summation over the single particle momenta $k_a \in \frac{2\pi}{L}(\mathbb{Z} - \nu)$, at the price of normalizing by the number of sectors $N!$, each sector (ordering) contributes the same amount. The only new terms are collisions when $k_a = k_b, a \ne b$, but at these values the determinant vanishes due to its identical rows. Additionally, we absorb pre- and postfactors by multiplying specific rows and columns in the determinant, as is common with Cauchy-like matrices.

$$\tau_{\mathbf{g}}(x, t) = \frac{1}{N!} \sum_{k_1} \sum_{k_2} \cdots \sum_{k_N} \left(\det_{1 \le a, b \le N} \frac{2\sin(\pi\nu)e^{\frac{i}{2}x(g_b - k_a) - \frac{i}{4}t(g_b^2 - k_a^2)}}{L(k_a - g_b)}\right)^2. \tag{34}$$

Let us define a function $B$ to temporarily simplify notation,

$$B(k, g) := \frac{2\sin(\pi\nu)e^{\frac{i}{2}x(g-k) - \frac{i}{4}t(g^2 - k^2)}}{L(k - g)}, \tag{35}$$

in terms of which we expand both instances of the determinants in (34) using $N$-dimensional Levi-Civita symbols.

$$\begin{aligned}
\tau_{\mathbf{g}}(x, t) &= \frac{1}{N!} \sum_{k_1} \sum_{k_2} \cdots \sum_{k_N} \varepsilon_{a_1, \ldots a_N} \varepsilon_{b_1, \ldots b_N} B(k_1, g_{a_1}) B(k_1, g_{b_1}) \ldots B(k_N, g_{a_N}) B(k_N, g_{b_N}) \\
&= \frac{1}{N!} \varepsilon_{a_1, \ldots a_N} \varepsilon_{b_1, \ldots b_N} \prod_{m=1}^{N} \left(\sum_{k_m} B(k_m, g_{a_m}) B(k_m, g_{b_m})\right) \\
&= \det_{1 \le a, b \le N} \left(\sum_k B(k, g_a) B(k, g_b)\right).
\end{aligned} \tag{36}$$

We have used the observation that the sum factorizes in the second equality, allowing us to drop the subscript $m$. Next we recognized the simultaneous row and column expansion of the determinant in the third equality. This entire derivation is a kind of infinite-dimensional *Cauchy-Binet identity*.

We now direct our attention to the matrix found at the end of (36),

$$A(g_a, g_b) := \sum_k B(k, g_a) B(k, g_b) = \frac{4\sin^2(\pi\nu)}{L^2} \sum_k \frac{e^{\frac{i}{2}x(g_a + g_b - 2k) - \frac{i}{4}t(g_a^2 + g_b^2 - 2k^2)}}{(k - g_a)(k - g_b)}. \tag{37}$$

We wish to take the TDL and turn the sum into an integral. However, due to the singularities, we must move away from the real line. The sum can instead be presented as a contour integral,

of an integrand with simple poles at the summation points $k \in \frac{2\pi}{L}(n - \nu), n \in \mathbb{Z}$. It is useful to choose these points to be the zeros of the function $\cot(kL/2) + \cot(\pi\nu)$, the first reason is that the residue exactly absorbs part of the numerator

$$\lim_{k \to \frac{2\pi}{L}(n-\nu)} \left(k + \nu\frac{2\pi}{L}\right)\frac{1}{\cot(kL/2) + \cot(\pi\nu)} = -\frac{2}{L}\sin^2(\pi\nu), \tag{38}$$

and let the variable $k$ be integrated along a contour $\gamma$ encircling all the poles. This way the matrix elements can be presented as

$$A(g_a, g_b) = i\frac{e^{\frac{i}{2}x(g_a+g_b)-\frac{i}{4}t(g_a^2+g_b^2)}}{\pi L}\oint_\gamma \frac{dk}{\cot(kL/2) + \cot(\pi\nu)}\frac{e^{-ixk+\frac{i}{2}tk^2}}{(k-g_a)(k-g_b)}. \tag{39}$$

The second exceptional property stemming from the choice of the pole generating function concerns the other singularities of expression (39). Ostensibly, it also has poles when $k$, now continuous, equals $g_a$ or $g_b$. The contour $\gamma$ would have to avoid these points artificially. However, when $a \neq b$ these are separate, and $\cot(g_a L/2) = \infty$, as for $g_b$, so these first order poles are cancelled. Thus we allow $\gamma$ to encircle $g_a$ and $g_b$, as long as they are distinct. See figure 4 for an illustration of $\gamma$.

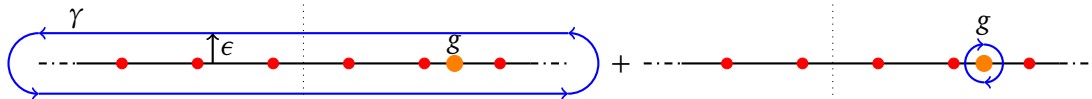

Figure 4: Contour $\gamma$ from (39). It may be taken counter-clockwise, a distance $\epsilon$ around the whole real line. However, to compensate for picking up the pole at $k \to g = g_a = g_b \in \frac{2\pi}{L}\mathbb{Z}$, we add the reverse contribution there. Red dots symbolize poles $k \to \frac{2\pi}{L}(n - \nu), n \in \mathbb{Z}$.

The diagonal contribution: $a = b$, where the pole is second order to begin with, must be considered separately, and the contour $\gamma$ must avoid these double poles. Indeed, let $g := g_a = g_b \in \frac{2\pi}{L}\mathbb{Z}$, then there is a net first order pole at $k \to g$, with residue,

$$\lim_{k \to g} \frac{(k-g) \cdot -2e^{ix(g-k)-\frac{i}{2}t(g^2-k^2)}}{L\left(\cot(kL/2) + \cot(\pi\nu)\right)(k-g)^2} = \lim_{k \to g} \frac{-\frac{2}{L}\tan(kL/2)}{\left(1 + \tan(kL/2)\cot(\pi\nu)\right)(k-g)} = -1, \tag{40}$$

using l'Hôpital's rule, and observing that $\cos(gL/2) = 1$ for any $g$. Thus, the contour integral over $\gamma$ equivalent to an integral encircling the entire real axis at a distance $\epsilon$, as long as we add the negative contribution $\delta_{a,b}$ of the poles at diagonal indices[7]. In cavalier notation, for the integral $A$ in (39),

$$\oint_\gamma (\dots) \mapsto \int_{-\infty-i\epsilon}^{\infty-i\epsilon} (\dots) + \int_{\infty+i\epsilon}^{-\infty+i\epsilon} (\dots) + \delta_{a,b}. \tag{41}$$

This contribution being exactly a delta function is important for the identification of the Fredholm determinant further down, and it is the motivation for the gauge choice of momentum and energy being relative to the outer state, as well as a third reason for the choice of (38).

---

[7]If we are evaluating $A$ at some $g_a = g_b$, while $a \neq b$, obviously the delta is also present. However, this case can be safely ignored because the elements of **g** are distinct. Later we will augment to consider more general sets **g** with repeated elements. This incurs no penalty as the determinant of the matrix $A$ vanishes due to repeated rows/columns. But one could technically keep in mind that the $\delta_{a,b}$ is actually a Kronecker $\delta(g_a, g_b)$.

The next step is to perform the integrals in (39), over two infinite lines: $\epsilon$ above and below the real axis in $k$, neglecting the caps. We make a simplification that is exact in the large L limit: replacing the fast oscillation of the cotangent by its period average. The logic is the following,

$$\int_{-\infty\pm i\epsilon}^{\infty\pm i\epsilon} \frac{y(k)dk}{\cot(kL/2)+\cot(\pi\nu)} \approx \frac{L}{2\pi} \int_{\pm i\epsilon}^{\frac{2\pi}{L}\pm i\epsilon} \frac{dk}{\cot(kL/2)+\cot(\pi\nu)} \cdot \int_{-\infty\pm i\epsilon}^{\infty\pm i\epsilon} y(k)dk, \qquad (42)$$

as long as the function $y(k)$ varies slowly on the scale of $\frac{2\pi}{L}$. In our case, sans prefactors, the magnitude $|\partial_k y(k)| \propto e^{\pm\epsilon(x-kt)}\epsilon^{-3}$ on the diagonal[8] in the limit of $\epsilon \to 0^+$. As for the oscillating part in that limit,

$$\frac{L}{2\pi} \int_{\pm i\epsilon}^{\frac{2\pi}{L}\pm i\epsilon} \frac{dk}{\cot(kL/2)+\cot(\pi\nu)} = \frac{1}{2\pi} \int_0^{2\pi} \frac{(e^{iw\mp\epsilon L}-1)dw}{i(e^{iw\mp\epsilon L}+1)+(e^{iw\mp\epsilon L}-1)\cot(\pi\nu)}$$

$$= \frac{1}{\cot(\pi\nu)\mp i} + \mathcal{O}\left(e^{-\epsilon L}\right) \qquad (43)$$

$$\approx \pm\frac{i}{2}\left(1-e^{\pm 2\pi i\nu}\right)$$

found by expressing the sine and cosine as exponents and substituting $kL \mapsto w$. Depending on the sign in front of $\epsilon$, in the second equality, we neglect either the oscillating or non-oscillating terms, in both cases obtaining a constant integrand. We consider a specific hierarchy of limits: first taking $L \to \infty$ before considering finite (but large) $x$ and $t$. This way, it is clear that the desired behavior is obtained for e.g. $\epsilon \propto 1/\sqrt{L}$. This vanishes, while also allowing the error in (43) and crucially the product of the error and $|\partial_k y(k)|$ to vanish as well, making our approximation indeed exact in the TDL. So each infinite line integral has its own prefactor, related by negative[9] complex conjugation. We collect these findings into the result[10]

$$A(g_a, g_b) = \delta_{a,b} + \frac{e^{\frac{i}{2}x(g_a+g_b)-\frac{i}{4}t(g_a^2+g_b^2)}}{2\pi L} \times$$

$$\int dk \left[ \frac{\left(1-e^{2\pi i\nu}\right)e^{-ixk+\frac{i}{2}tk^2}}{(k+i\epsilon-g_a)(k+i\epsilon-g_b)} + \frac{\left(1-e^{-2\pi i\nu}\right)e^{-ixk+\frac{i}{2}tk^2}}{(k-i\epsilon-g_a)(k-i\epsilon-g_b)} \right]. \qquad (44)$$

Summarizing, we have the $\tau$-function in state $|\mathbf{g}\rangle$ equaling the determinant of the $N \times N$ matrix $A(g_a, g_b)$,

$$\tau_{\mathbf{g}}(x,t) = \det_{1\le a,b\le N}\left(A(g_a, g_b)\right) = \det_{1\le a,b\le N}\left(\delta_{a,b} + \frac{2\pi}{L}\frac{e_+(g_a)e_-(g_b)-e_-(g_a)e_+(g_b)}{g_a - g_b}\right), \quad (45)$$

where diagonal terms are understood as the limit $g_a \to g_b$. In keeping with tradition [2,78,91], we have massaged the quotient into the two auxiliary functions

$$e_-(g) := e^{\frac{i}{2}xg-\frac{i}{4}tg^2},$$

$$e_+(g) := \frac{e_-(g)}{4\pi^2} \int dk \left[ \frac{1-e^{2\pi i\nu}}{k+i\epsilon-g} + \frac{1-e^{-2\pi i\nu}}{k-i\epsilon-g} \right] e^{-ixk+\frac{i}{2}tk^2}. \qquad (46)$$

---

[8]for $a \neq b$ it's $e^{\pm\epsilon(x-kt)}\epsilon^{-2}$.

[9]The top line goes from $\infty$ to $-\infty$.

[10]Technically, the exponents inside the brackets should carry factors $e^{\pm\epsilon(x-tk)-\frac{i}{2}t\epsilon}$, however the exponent is regular and in the implied limit $\epsilon \to 0^+$ these factors have no effect.

Note that these functions do not depend on the system size.

In order to find the TDL of such a determinant, we must have a description of how to select the quantum numbers feeding **g** as $N$ increases. In general, they follow some distribution $\rho(g)$. At this point, one case is obvious: for the ground state, we may already specify $|\mathbf{g}\rangle$ to be the $N$-particle set $g_a = \frac{2\pi}{L}(a - N/2)$. Once $e_+$ is known, (45) suffices to approximate the $\tau$-function. Formally, we may choose to go to the TDL, and recognizing $\frac{2\pi}{L} \to dg$ we obtain the Fredholm determinant of an integral kernel $K(p, q)$ in dummy variables $p$ and $q$ [55]. Such a kernel is understood to act as a convolution, the natural smooth infinitesimal limit of matrix multiplication. For zero temperature,

$$\tau_0(x, t) = \det_{[-k_F, k_F]^2}\left(\mathbb{1} + \hat{K}\right); \quad K(p, q) := \frac{e_+(p)e_-(q) - e_-(p)e_+(q)}{p - q}, \tag{47}$$

where $k_F := \pi N/L$ is the Fermi surface, meaning integration is over the whole sea, and $\mathbb{1}$ the identity operator.

Thermal states are more subtle, as discussed in section 3. If we generate a sequence of many-body states for increasing $N$ such that probability of finding single particle quantum number $g_a$ occupied is always $\rho(g_a)$, we can formally take the limit of $N \to \infty$ of a 'thermal state' $|\mathbf{g}\rangle_{\text{th}}$. Then we can consider the sequence of $\tau$-functions, evaluated in each of these states. We posit that, regardless of the random microscopic realization of the state at each finite $N$, the resulting sequence of determinants converges to the correct Fredholm, which will be calculated exactly, later in equation (62).

But first, we return our attention to the correct evaluation of $e_+(g)$. The integrals in the kernels can be expressed by means of some special functions. To this end, we first consider the auxiliary definition

$$I_\pm(x, t) := \frac{i}{2\pi} \lim_{\epsilon \to 0^+} \int dk \frac{e^{-ixk + \frac{i}{2}tk^2}}{k \pm i\epsilon}. \tag{48}$$

It is directly related to the integrals in the kernel in (46), since

$$\frac{1}{2\pi i} \lim_{\epsilon \to 0^+} \int dk \frac{e^{-ixk + \frac{i}{2}tk^2}}{k \pm i\epsilon - g} = -e^{-ixg + \frac{i}{2}tg^2} I_\pm(x - gt, t). \tag{49}$$

Applying the *Sokhotski-Plemelj* theorem, we identify

$$I_\pm(x, t) = \pm\frac{1}{2} + \frac{i}{2\pi}\mathcal{P}\int \frac{e^{-ixk + \frac{i}{2}tk^2}dk}{k}, \tag{50}$$

with $\mathcal{P}$ indicating the Cauchy principal value of the integral. This identity allows us to set the value for $x = 0$. Namely, using the parity of the integrand we obtain $I_\pm(0, t) = \pm 1/2$. Differentiating over $x$ to remove the singularity, we obtain

$$\frac{\partial}{\partial x} I_\pm(x, t) = \frac{1}{2\pi}\int e^{-ixk + \frac{i}{2}tk^2}dk = \frac{i+1}{2\sqrt{\pi t}}e^{-\frac{i}{2}x^2/t}. \tag{51}$$

Integrating back, and using the initial condition, we arrive at

$$I_\pm(x, t) = \frac{i+1}{2\sqrt{\pi t}}\int_0^x e^{-\frac{i}{2}y^2/t}dy \pm \frac{1}{2} = \frac{1}{2}\left(\text{erf}\left(\frac{i+1}{2\sqrt{t}}x\right) \pm 1\right), \tag{52}$$

which, after we substitute (52) into (49) into (46) gives the following expression for the function $e_+(g)$ in the kernel:

$$
\begin{aligned}
e_+(g) &= -\frac{e^{-\frac{i}{2}xg+\frac{i}{4}tg^2}}{2\pi}\left[\frac{1+\mathrm{erf}\left(\frac{(i+1)(x-gt)}{2\sqrt{t}}\right)}{\cot(\pi v)-i}+\frac{1-\mathrm{erf}\left(\frac{(i+1)(x-gt)}{2\sqrt{t}}\right)}{\cot(\pi v)+i}\right] \\
&= -\frac{\sin^2(\pi v)e^{-\frac{i}{2}xg+\frac{i}{4}tg^2}}{\pi}\left[\cot(\pi v)+i\,\mathrm{erf}\left(\frac{(i+1)(x-gt)}{2\sqrt{t}}\right)\right].
\end{aligned}
\tag{53}
$$

The limit of the erf at late times is a step function. For a visualization see figure 5.

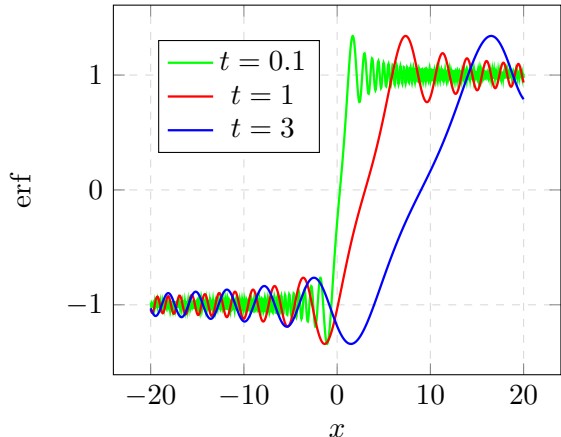

Figure 5: Plots of the real part of object $\mathrm{erf}\left(\frac{(i+1)(x-gt)}{2\sqrt{t}}\right)$, at $g=3$, for different times $t$. The function approximates $2\theta(x-gt)-1$. As $t\to 0$, the oscillations condense and the limit becomes exact. The imaginary part (not shown) oscillates around zero.

In an analogous fashion we can derive that for the static case $t=0$,

$$
I_\pm(x,0)=\theta(x)-\frac{1}{2}\pm\frac{1}{2}.
\tag{54}
$$

Therefore, putting (54) into (49) into (46), we find for the static case,

$$
e_+(g)=-\frac{e^{-\frac{i}{2}xg}}{\pi}\left[\frac{\theta(x)}{\cot(\pi v)-i}+\frac{1-\theta(x)}{\cot(\pi v)+i}\right],
\tag{55}
$$

confirming that taking $x\mapsto -x$ is equivalent to complex conjugation of $\tau(x,0)$, as was clear by construction in (1). Specializing to $x>0$, the kernel becomes

$$
K(p,q)=\frac{e^{2i\pi v}-1}{\pi}\frac{\sin\left(\frac{x}{2}(p-q)\right)}{p-q},
\tag{56}
$$

which we recognize as the renowned *Sine Kernel*, having applications in Random Matrix Theory, Painleve equations and other branches of mathematics (see for instance [92]). More relevantly, its Fredholm determinant is used to express other correlation functions in quantum systems such as the *Tonks Girardeau* impenetrable Bose gas [91], as discussed in 2. For an illustration of the single state $\tau$-functions, see figure 6. Although not technically exact for finite $N$ due to (43), the figure is instructive. It reveals that in the static case, $\tau_g$ is periodic in $x$ with period $L$. Also, already for 7 particles, the ground state approximates the TDL quite well.

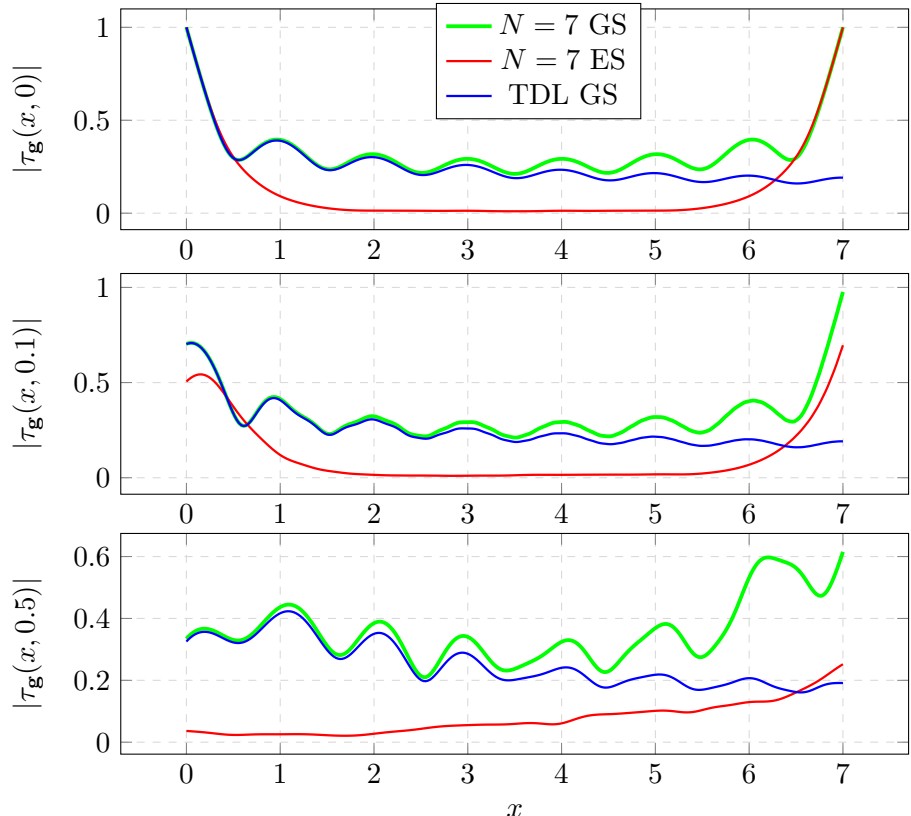

Figure 6: Absolute values of the single state $|\tau_{\mathbf{g}}(x, t)|$ plotted in space for various times, from top to bottom, $t = 0$; $t = 0.1$; $t = 0.5$. Parameters $\nu = 0.4$, $N = L$. GS is the ground state, with ES an arbitrary excited state $|\mathbf{g}\rangle$ of $N = 7$ particles, having quantum numbers $n_a = (-5, -3, 0, 1, 4, 5, 8)$ in $g_a = \frac{2\pi}{L}n_a$. The TDL indicates the Fredholm determinant as in (47).

## 4.2 Gibbs Ensemble of States

In this section, we generalize the $\tau$-function to the more complex case of finite temperature, understood as the statistical average of single eigenstates, weighted by the grand canonical ensemble, or Gibbs ensemble. We do not need the distribution of single particle occupations to be the Fermi-Dirac distribution (23), any $\rho(q)$ with an interpretation as a local probability will work equally well.

Formally we have

$$\tau(x, t) = \sum_N \sum_{\mathbf{q}} P(\mathbf{q})\tau_{\mathbf{q}}(x, t), \tag{57}$$

where $P(\mathbf{q})$ is the normalized probability of finding a state $|\mathbf{q}\rangle$ with unshifted quantum numbers $\mathbf{q} = (q_1 < \ldots < q_N)$ in the ensemble. Observe that this is also a sum over all system sizes $N$. $P(\mathbf{q})$ is given by the product of the probability $\rho(q)$ for each of the $N$ momenta $q$ to be occupied if $q \in \mathbf{q}$, times infinite product of the probability $(1 - \rho(q))$ for $q$ to be unoccupied, if $q \notin \mathbf{q}$. Equivalently,

$$P(\mathbf{q}) = D \cdot \prod_{k \in \mathbf{q}} \frac{\rho(q)}{1 - \rho(q)}, \tag{58}$$

where we have separated a state independent prefactor

$$D := \prod_{q \in \frac{2\pi}{L}\mathbb{Z}} \left(1 - \rho(q)\right) = \det_{(a,b) \in \mathbb{Z}^2} \left(\delta_{a,b}[1 - \rho(q_a)]\right), \tag{59}$$

already suggestively presented as a determinant[11]. Dividing out this prefactor yields a finite expression for the probability of $|\mathbf{q}\rangle$. In the case of the Fermi-Dirac distribution, $P(\mathbf{q}) = De^{\sum_a -(q_a^2-\mu)/T}$ follows directly from (58), in keeping with the principles of statistical physics.

Continuing, we may absorb the statistical weight into the determinant expression by multiplying the square root of the factors into the rows and another into the columns[12] of $A(q_a, q_b)$ in (45), and we employ the familiar trick of summing over all $q_a$ freely, not just the strictly ordered sector[13].

$$\tau(x,t) = D \sum_N \frac{1}{N!} \left[ \sum_{q_1} \cdots \sum_{q_N} \det_{1 \le a,b \le N} \left( \sqrt{\frac{\rho(q_a)}{1-\rho(q_a)}} A(q_a, q_b) \sqrt{\frac{\rho(q_b)}{1-\rho(q_b)}} \right) \right]. \tag{60}$$

It is clear that this is a sum over all the principal minors of an infinite-dimensional matrix, and by careful consideration of the Leibniz expansion with Levi-Civita symbols, one can obtain the identity known as the *Von Koch formula* [55].

$$\tau(x,t) = D \det_{(a,b) \in \mathbb{Z}^2} \left[ \delta_{a,b} + \sqrt{\frac{\rho(q_a)}{1-\rho(q_a)}} A(q_a, q_b) \sqrt{\frac{\rho(q_b)}{1-\rho(q_b)}} \right]. \tag{61}$$

The idea of this equivalence is to expand the product over terms $(\delta_{a,b} + (\ldots))$ in the Leibniz representation of (61). The term corresponding to a given $N$ in (60) results from $N$ factors of $(\ldots)$ and all other factors $\delta_{a,b}$, which in turn reduce the infinite-dimensional Levi-Civita symbol to the $N$-dimensional one. See also an equivalent calculation in the appendix of [61]. Then the upshot is a new Fredholm determinant, but with infinite support. After multiplying the matrix with that of $\sqrt{D}$ from the left and right, for $D$ in (59), and recalling for integral operators $A = \mathbb{1} + K$,

$$\tau(x,t) = \det_{\mathbb{R}^2} \left[ \mathbb{1} + \hat{K}_\rho \right], \quad K_\rho(p,q) := \sqrt{\rho(p)} K(p,q) \sqrt{\rho(q)}. \tag{62}$$

Referring back to their construction in (46), this is akin to absorbing a factor of $\sqrt{\rho(q)}$ into the function $e_-(q)$ for finite temperature ensembles.

## 5 Zero Temperature Soft Mode Summation

Besides considering exponentially decaying overlaps, which we have seen for finite temperature in section 3, let us discuss how the zero temperature orthogonality catastrophe in (22) can be remedied with an explicit form-factor summation. As we have demonstrated in the previous section, the complete summation leads to a finite answer in the TDL, given by equation (47). In this section we show that there exists a partial summation of the so-called soft modes, which is also finite in the TDL, and in fact gives the same large time and space behavior of the Fredholm determinant $\tau$-function (47). This perspective is somewhat disconnected from the main (finite-temperature) theme of the rest of the paper. We mainly follow the lines of

---

[11]For infinite determinants such as this, the formal prescription is to start with a finite set $\{(a,b)\}$ and increase the bounds to $\infty$ as a limit. See appendix B for the details.

[12]Starting from (60), it would have been possible to absorb all the factors onto only the rows or columns of $A$, and we would have $\rho(p)$ instead of $\sqrt{\rho(p)\rho(q)}$ in (62). The current choice is made for symmetry, and is necessary for identification of an approximation scheme in section 6.

[13]We don't have to correct for the sign of reordering the elements, because we are simultaneously swapping rows and columns, and both operations carry a minus sign.

the reference found in [23]. This summation is in essence a microscopic justification of the bosonization technique of [24], and this section relies heavily on a technical understanding of bosonization.

We start by considering $j$-particle excitation over the Fermi sea ground state for shifted fermions $|\mathbf{k}\rangle$. Namely, we specify the set of holes[14]

$$H_j = \{h_1, h_2, \dots h_j\}, \quad 1 \le h_a \le N \tag{63}$$

and the set of particles

$$P_j = \{p_1, p_2, \dots p_j\}, \quad p_a \le 0 \text{ or } p_a > N. \tag{64}$$

Meaning that $\left|\mathbf{k}_{P_j, H_j}\right\rangle$ is found by removing the set $H_j$ from the Fermi sea $|\mathbf{k}\rangle$ and adding the set $P_j$. As before, $|\mathbf{g}\rangle$ is also the ground state, in the unshifted basis. From the product presentation of the Cauchy determinant one immediately deduces,

$$\frac{|\langle \mathbf{k}_{P_j, H_j} | \mathbf{g}\rangle^2}{|\langle \mathbf{k} | \mathbf{g}\rangle|^2} = \left[ \det_{1 \le a, b \le j} \left( \frac{1}{p_a - h_b} \right) \right]^2 \prod_{p \in P_j} F(p) \prod_{h \in H_j} \tilde{F}(h), \tag{65}$$

where

$$F(p) = \prod_{a=1}^{N} \left( \frac{p - a}{p - a - v} \right)^2 = \left( \frac{\Gamma(p)\Gamma(p - N - v)}{\Gamma(p - N)\Gamma(p - v)} \right)^2,$$

$$\tilde{F}(h) = v^2 \prod_{a=1, a \neq h}^{N} \left( \frac{h - a - v}{h - a} \right)^2 = \left( \frac{\sin \pi v}{\pi} \frac{\Gamma(h - v)}{\Gamma(h)} \frac{\Gamma(N - h + v + 1)}{\Gamma(N - h + 1)} \right)^2. \tag{66}$$

Note that $F(h)$ automatically vanishes for $h$ lying outside the interval $[1, N]$, and $F(p)$ for $p < 0$ is understood as a limit

$$\lim_{\epsilon \to 0} \frac{\Gamma(p + \epsilon)\Gamma(p - N - v + \epsilon)}{\Gamma(p - N - \epsilon)\Gamma(p - v - \epsilon)} = \frac{\Gamma(1 - p + N)\Gamma(1 - p + v)}{\Gamma(1 - p)\Gamma(1 - p + N + v)}. \tag{67}$$

Now let us consider the contributions of the so-called soft modes – the particle-hole excitations over microscopic distance within some neighborhood of the Fermi surface. Let us shift the indexing, counting the positions of the excitations from the right edge as

$$p \mapsto p^+ + N, \quad p^+ = 1, 2 \dots, \quad h \mapsto N - h^+ + 1, \; h^+ = 1, 2, \dots N/2, \tag{68}$$

and for the excitations from the left edge

$$p \mapsto 1 - p^-, \quad p^- = 1, 2 \dots, \quad h \mapsto h^-, \; h^- = 1, 2, \dots N/2. \tag{69}$$

Moreover, let us specify that we are in a state with $N_+$ particle-hole-pairs excited around the right edge and $N_-$ pairs excited around the left edge, such that $j = N_+ + N_-$. We construct:

$$\frac{|\langle \mathbf{k}_{P_j, H_j} | \mathbf{g}\rangle^2}{|\langle \mathbf{k} | \mathbf{g}\rangle|^2} = Q(\{p^+, p^-\}, \{h^+, h^-\}, v) R(\{p_+\}, \{h_+\}, v) R(\{p^-\}, \{h^-\}, -v), \tag{70}$$

with the definitions

$$Q(\{p^+, p^-\}, \{h^+, h^-\}) := \prod_{i=1}^{N_+} \prod_{j=1}^{N_-} \frac{(N + 1 - h_i^+ - h_j^-)^2 (N - 1 + p_i^+ + p_j^-)^2}{(N + p_i^+ - h_j^-)^2 (N + p_j^- - h_i^+)^2}$$

$$\times \prod_{i=1}^{N_+} \frac{\Gamma^2(p_i^+ + N)}{\Gamma^2(p_i^+ + N - v)} \frac{\Gamma^2(N - h_i^+ + 1 - v)}{\Gamma^2(N - h_i^+ + 1)} \prod_{j=1}^{N_-} \frac{\Gamma^2(p_j^- + N)}{\Gamma^2(p_j^- + N + v)} \frac{\Gamma^2(N - h_j^- + 1 + v)}{\Gamma^2(N - h_j^- + 1)}, \tag{71}$$

---

[14]In contrast to previous sections, now $h$ and $p$ are integers, not scaled to momenta by $\frac{2\pi}{L}$.

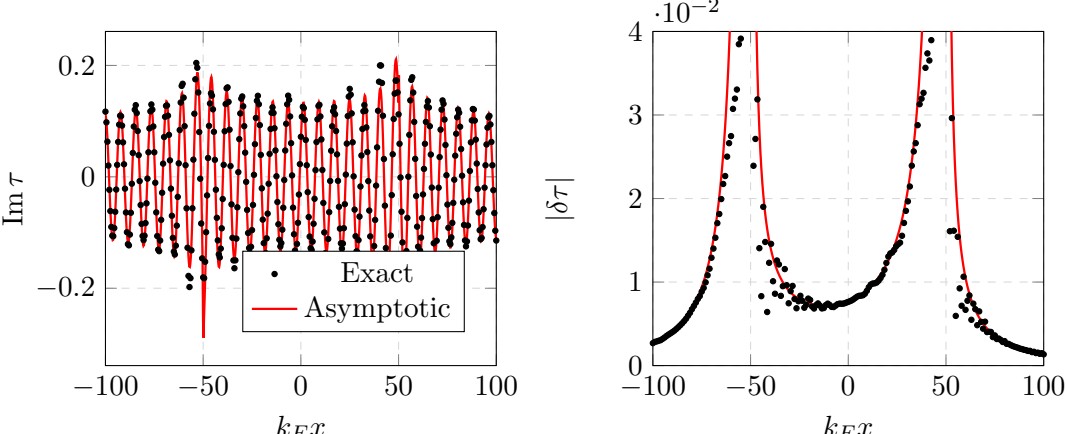

Figure 7: A comparison of the exact expression for the zero temperature $\tau$-function $\tau(x, 100E_F)$, computed by means of the Fredholm determinant (47) (black dots) alongside the asymptotics (red solid lines) for $\nu = 0.4$ and $k_F = 1$. The left panel shows the leading asymptotic (95). The right panel compares the exact answer (102) from which the leading asympotics has been subtracted, against the saddle point contributions (101). This singles out the artificial divergence of asymptotic expansion on the light cone. The time is fixed to $t = 100$ and $E_F = 50k_F^2$, which means that for $k_F = 1$ the singularities are located at $x = \pm 50$. Close to the cone, the approximation is already acceptable.

mediating inter-edge influence, and

$$R(\{p_+\}, \{h_+\}, \nu) := \left(\frac{\sin \pi \nu}{\pi}\right)^{2N_+} \frac{\prod_{i>j}(p_i^+ - p_j^+)^2(h_i^+ - h_j^+)^2}{\prod_{i,j}(p_i^+ + h_j^+ - 1)^2} \prod_{j=1}^{N_+} \frac{\Gamma^2(p_j^+ - \nu)}{\Gamma^2(p_j^+)} \prod_{j=1}^{N_+} \frac{\Gamma^2(h_j^+ + \nu)}{\Gamma^2(h_j^+)},$$

(72)

for intra-edge effects. Taking $\{k_a\}$ to be the excited single particle momenta, the total momentum and energy, as featured in the definition of the $\tau$-function (1), are

$$\delta P_{\mathbf{k}} \equiv \sum_a g_a - k_a = 2k_F \nu - \frac{2\pi}{L} \sum_{a=1}^{N_+} (p_a^+ + h_a^+ - 1) + \frac{2\pi}{L} \sum_{a=1}^{N_-} (p_a^- + h_a^- - 1),$$

(73)

$$\delta E_{\mathbf{k}} \equiv \sum_a \frac{g_a^2}{2} - \frac{k_a^2}{2} = -\frac{2\pi}{L} k_F \sum_{a=1}^{N_+} (p_a^+ + h_a^+ - 1) - \frac{2\pi}{L} k_F \sum_{a=1}^{N_-} (p_a^- + h_a^- - 1).$$

(74)

Here we have reminded ourselves of the Fermi momentum $k_F = \frac{\pi N}{L}$. This momentum basically defines the domain on which the Fredholm operator acts (see equation (47)). It can be divided out by a proper rescaling of $x$ and $t$, but we prefer to keep it explicitly.

Now let us turn our attention to the soft modes satisfying $p^\pm \ll N$ and $h^\pm \ll N$. In this case,

$$Q(\{p^+, p^-\}, \{h^+, h^-\}) \approx 1.$$

(75)

So the summations over left and right edges are virtually independent.

With this simplification, the remaining summations over soft modes can be carried out after making the observation that the ratio of overlaps is essentially an average of vertex operators in free boson theory. This approach is nothing but an instance of bosonization in the style

of the Kyoto school [93–96]. To be more specific, we consider the theory of *auxiliary* Dirac fermions $\psi(z)$ and $\bar{\psi}(z)$, $z \in \mathbb{C}$ [24,97]. The mode expansion is defined as

$$\bar{\psi}(z) = \sum_{n \in \mathbb{Z}} z^n \psi_n^+, \quad \psi(z) = \sum_{n \in \mathbb{Z}} z^{-n} \psi_n \tag{76}$$

with modes satisfying fermionic commutation relations

$$\{\psi_n, \psi_m^+\} = \delta_{n,m}, \qquad \{\psi_n, \psi_m\} = \{\psi_n^+, \psi_m^+\} = 0. \tag{77}$$

The vacuum of the Dirac fermions, $|0\rangle$, is chosen as the state with all non-positive momenta filled by fermions

$$\psi_n |0\rangle = 0, \ \ \text{if } n > 0, \quad \psi_n^+ |0\rangle = 0, \ \ \text{if } n \le 0. \tag{78}$$

The vacuum expectation value of these fields is equal to

$$\langle 0|\bar{\psi}(z)\psi(w)|0\rangle = \frac{1}{1-w/z}, \tag{79}$$

with the assumption that $|w/z| < 1$. We define the fermion density as the normal ordered product with respect to the vacuum (78), namely

$$J(z) = : \psi^+(z)\psi(z) : = \sum_{a \in \mathbb{Z}} \frac{\sum_{b \in \mathbb{Z}} : \psi_b^+ \psi_{b+a} :}{z^a} \equiv \sum_{a \in \mathbb{Z}} \frac{J_a}{z^a}. \tag{80}$$

Using commutation relations (77), one can easily obtain for newfound currents $J_a$,

$$[J_a, \psi_m] = \psi_{m-a}, \quad [J_a, \psi_m^+] = -\psi_{m+a}^+, \quad [J_a, J_b] = a\delta_{a+b,0}. \tag{81}$$

We see that the currents form a Heisenberg algebra of free-bosons. More formally, we unite the positive and negative components into bosonic fields in the following way:

$$\varphi_+(z) = \sum_{a>0} \frac{J_a}{az^a}, \quad \varphi_-(z) = \sum_{a>0} z^a \frac{J_{-a}}{a}. \tag{82}$$

Additionally, we can define the field $\varphi(z)$

$$\varphi(z) := \varphi_-(z) - \varphi_+(z) + J_0 \log z + P, \tag{83}$$

where $P$ is an additional operator, conjugated with $J_0$ by the relation

$$[J_0, P] = 1. \tag{84}$$

Therefore, formally, we can write

$$z\frac{\partial \varphi(z)}{\partial z} = J(z), \tag{85}$$

which resembles the way bosonic fields are introduced in traditional condensed matter physics [4,98,99]. By vertex operator, we understand the normally ordered exponent

$$e^{\varphi(z)} :\equiv e^{\varphi_-(z)} e^P z^{J_0} e^{-\varphi_+(z)}. \tag{86}$$

The key observation is that $R$ from (72) can be presented as a matrix element of the vertex operator, between the vacuum $\langle 0|$ and the excited state

$$|P_j, H_j\rangle = \psi_{p_1}^+ \ldots \psi_{p_j}^+ \psi_{1-h_1} \ldots \psi_{1-h_j} |0\rangle. \tag{87}$$

This state describes soft modes. Note, however, that contrary to the original formulation, the range for $p_a$ and $h_a$ is unlimited. Using Wick's theorem, we can present

$$\det_{1\le a,b\le j} S_{p_a h_b} := \langle 0|e^{\alpha\varphi_+(z)}|P_j, H_j\rangle = \det_{1\le a,b\le j} \langle 0|e^{\alpha\varphi_+(z)}\psi^+_{p_a}\psi_{1-h_b}|0\rangle. \tag{88}$$

In order to compute this matrix element, we follow the approach in reference [96]. Namely, first we present each mode as a contour integral encircling the origin,

$$S_{ph}(z) = \frac{1}{(2\pi i)^2} \oint \frac{du}{u^{p+1}} \oint \frac{dv}{v^h} \langle 0|e^{\alpha\varphi_+(z)}\psi^+(u)\psi(v)|0\rangle =$$

$$\frac{1}{(2\pi i)^2} \oint \frac{du}{u^{p+1}} \oint \frac{dv}{v^h} \langle 0|e^{\alpha\varphi_+(z)}\psi^+(u)e^{-\alpha\varphi_+(z)}e^{\alpha\varphi_+(z)}\psi(v)e^{-\alpha\varphi_+(z)}|0\rangle =$$

$$= \frac{1}{(2\pi i)^2} \oint \frac{du}{u^p} \oint \frac{dv}{v^h}\frac{1}{u-v}\left(\frac{z-v}{z-u}\right)^\alpha. \tag{89}$$

Here we have assumed that $|v/u| < 1$. After taking a derivative over $z$ the integrals decouple. Indeed, taking into account that

$$\partial_z \frac{1}{u-v}\left(\frac{z-u}{z-v}\right)^\alpha = -\frac{\alpha}{(z-u)(z-v)}\left(\frac{z-u}{z-v}\right)^\alpha, \tag{90}$$

we obtain

$$\partial_z S_{ph}(z) = -\alpha \oint \frac{du}{2\pi i}\frac{(z-u)^{-\alpha-1}}{u^p} \oint \frac{dv}{2\pi i}\frac{(z-v)^{\alpha-1}}{v^h} =$$

$$-\alpha z^{-\alpha-p}\frac{\Gamma(p+\alpha)}{\Gamma(p)\Gamma(1+\alpha)}z^{\alpha-h}\frac{\Gamma(h-\alpha)}{\Gamma(h)\Gamma(1-\alpha)}. \tag{91}$$

Integrating back, the answer reads

$$S_{ph}(z) = \frac{z^{1-p-h}}{p+h-1}\frac{\sin\pi\alpha}{\pi}\frac{\Gamma(p+\alpha)}{\Gamma(p)}\frac{\Gamma(h-\alpha)}{\Gamma(h)}. \tag{92}$$

Similar results can be obtained for the matrix elements of $e^{i\alpha\varphi_-(z)}$. Finally, dividing the particles and holes per side, we can present

$$\frac{|\langle \mathbf{k}_{P_j,H_j}|\mathbf{g}\rangle|^2}{|\langle \mathbf{k}|\mathbf{g}\rangle|^2}e^{ix\delta P_{\mathbf{k}}-it\delta E_{\mathbf{k}}} = e^{2i\nu k_F x}\langle 0|e^{\nu\varphi_+(1)}|P_j^+, H_j^+\rangle\langle P_j^+, H_j^+|e^{\nu\varphi_-(z^+)}|0\rangle$$

$$\times \langle 0|e^{\nu\varphi_+(1)}|P_j^-, H_j^-\rangle\langle P_j^-, H_j^-|e^{\nu\varphi_-(z^-)}|0\rangle. \tag{93}$$

Here $z^\pm = e^{2\pi i(t\pm x+i\epsilon)/L}$. We have added $i\epsilon$ as a regulator, which will later be taken to $\epsilon \to 0$, in order to have convergent sums. Now performing summation is straightforward, as the sums simply form a resolution of unity,

$$\sum_{P,H}\frac{|\langle \mathbf{k}_{P_j,H_j}|\mathbf{g}\rangle|^2}{|\langle \mathbf{k}|\mathbf{g}\rangle|^2}e^{ix\delta P_{\mathbf{k}}-it\delta E_{\mathbf{k}}} =$$

$$e^{2i\nu k_F x}\langle 0|e^{\nu\varphi_+(1)}e^{\nu\varphi_-(z^+)}|0\rangle\langle 0|e^{\nu\varphi_+(1)}e^{\nu\varphi_-(z^-)}|0\rangle = \frac{e^{2i\nu k_F x}}{(1-z^+)^{\nu^2}(1-z^-)^{\nu^2}}. \tag{94}$$

In the last step we have computed the average of the vertex operators, in free Gaussian theory.

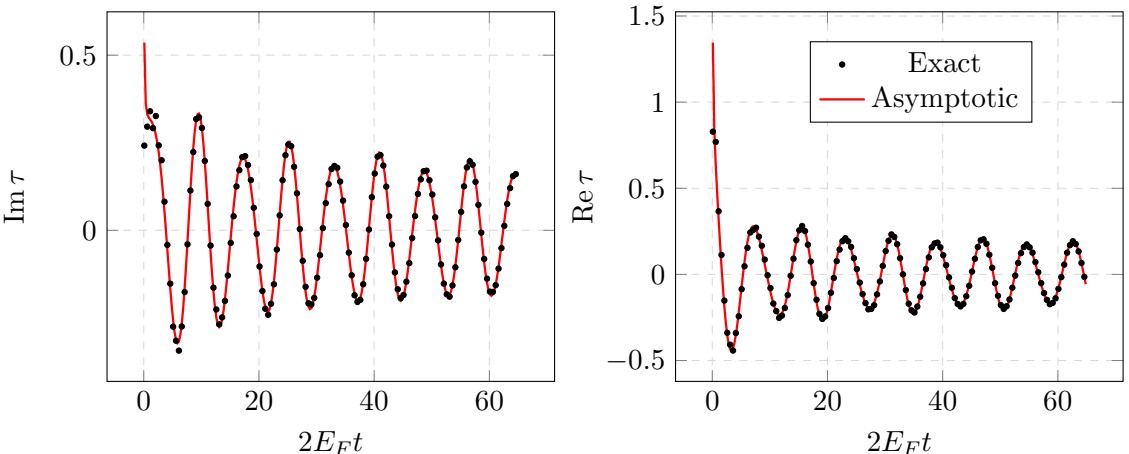

Figure 8: A comparison of the exact expression for the zero temperature tau function $\tau(k_F t, t)$ computed by the Fredholm determinant (47) at the light cone (black dots), over the asymptotics (red solid lines) for $\nu = 0.4$ and $k_F = 1$.

Using the orthogonality catastrophe (22) for the overlap $|\langle \mathbf{k}|\mathbf{g}\rangle|^2$ we obtain an expression that has a finite TDL,

$$\tau_\nu^{\mathrm{IR}}(x,t) \equiv \sum_{\mathbf{k} \in \mathrm{IR}} e^{ix\delta P_{\mathbf{k}} - it\delta E_{\mathbf{k}}} |\langle \mathbf{k}|\mathbf{g}\rangle|^2 =$$

$$\frac{G(1-\nu)^2 G(1+\nu)^2}{N^{2\nu^2}} \frac{e^{2i\nu k_F x}}{(1-e^{2\pi i(k_F t - x + i\epsilon)/L})^{\nu^2}(1-e^{2\pi i(k_F t + x - i\epsilon)/L})^{\nu^2}} =$$

$$= \frac{G^2(1-\nu)G^2(1+\nu)e^{2i\nu k_F x}}{(-2ik_F(k_F t - x + i\epsilon))^{\nu^2}(-2ik_F(k_F t + x + i\epsilon))^{\nu^2}}. \quad (95)$$

Where we have taken $L \to \infty$, $N \to \infty$, expanded the exponent to first order, and used the definition of the Fermi momentum. The IR notation signifies that we are summing the simplified expressions of overlaps and extended the summation to infinity. The comparison with the exact form of the $\tau$-function is shown in figure 7. Notice that the original expression (47) was invariant under integer shifts $\nu \mapsto \nu + m$, $m \in \mathbb{Z}$. These shifts are still present in the asymptotic. They physically correspond to the *umklapp* effect, when $m$ fermionic quantum numbers are moved from the left edge of the Fermi sea to the right (or vice versa, depending on the sign of $m$). In most cases this contributes to subdominant orders and is suppressed for large $x$. Another contribution comes from the saddle point and can be described as follows. Let us consider one hard (i.e. macroscopically far from the Fermi surface) excitation, this means that the soft mode condition $p^\pm \ll N$ is violated for the largest $p_{N_+}^+ \equiv p \sim N$. Notice that in this case, the condition on $Q$ from (75) transforms into

$$Q \approx \left(\frac{p+N}{N}\right)^{2\nu}, \quad (96)$$

and expression (72) into

$$R \approx p^{-2(\nu+1)} \left( \frac{\sin \pi \nu}{\pi} \right)^{2N_+} \frac{\prod_{a>b}^{N_+-1}(p_a^+ - p_b^+)^2 \prod_{a>b}^{N_+}(h_a^+ - h_b^+)^2}{\prod_{a=1}^{N_+-1} \prod_{b=1}^{N_+}(p_a^+ + h_b^+ - 1)^2} \times$$

$$\prod_{b=1}^{N_+-1} \frac{\Gamma^2(p_b^+ - \nu)}{\Gamma^2(p_b^+)} \prod_{b=1}^{N_+} \frac{\Gamma^2(h_b^+ + \nu)}{\Gamma^2(h_b^+)}. \tag{97}$$

Now imagine that one of the $h_b^+$ equals 1, say $h_{N_+}^+ = 1$. Then the rest of excitations can be considered as particle-hole pairs over the Fermi sea with the last particle removed. Effectively this can be achieved by the redefinitions of $h_+ \mapsto \tilde{h}_+ = h_+ - 1$ and $p_+ \mapsto \tilde{p}_+ = p_+ + 1$. In these notations,

$$R \approx \frac{p^{-2\tilde{\nu}}}{\Gamma(-\nu)^2} \left( \frac{\sin \pi \nu}{\pi} \right)^{2\tilde{N}_+} \frac{\prod_{a>b}^{\tilde{N}_+}(\tilde{p}_a^+ - \tilde{p}_b^+)^2 \prod_{a>b}^{\tilde{N}_+}(\tilde{h}_a^+ - \tilde{h}_b^+)^2}{\prod_{a=1}^{\tilde{N}_+} \prod_{b=1}^{\tilde{N}_+}(\tilde{p}_a^+ + \tilde{h}_b^+ - 1)^2} \prod_{b=1}^{\tilde{N}_+} \frac{\Gamma^2(\tilde{p}_b^+ - \tilde{\nu})}{\Gamma^2(\tilde{p}_b^+)} \prod_{b=1}^{\tilde{N}_+} \frac{\Gamma^2(\tilde{h}_b^+ + \tilde{\nu})}{\Gamma^2(\tilde{h}_b^+)}, \tag{98}$$

where $\tilde{N}_+ = N_+ - 1$ and $\tilde{\nu} = \nu + 1$. Since none of the remaining $h_b^+$s equals 1 (1 is taken by $h_{N_+}^+$) the set of $\tilde{h}^+$ runs over positive integers and the set $\tilde{p}^+$ runs over positive integers greater than one. Relaxing our initial assumption of $h_{N_+}^+ = 1$ allows us to cover also the $\tilde{p}_b^+ = 1$ case, without changing the overall structure (98). Up to a modified $\nu$, this structure is identical to the soft modes that we had before, so we can perform the soft mode summation again to obtain

$$\tau_\nu^{\text{SP}}(x,t) \equiv \sum_{\mathbf{k} \in \{\text{IR} \cup p\}} e^{ix\delta P_\mathbf{k} - it\delta E_\mathbf{k}} |\langle \mathbf{k}|\mathbf{g}\rangle|^2 =$$

$$\frac{G(-\nu)^2 G(1+\nu)^2 e^{2i\nu k_F x - ixk_F + itk_F^2/2}}{N^{2\nu^2}(1 - e^{2\pi i(t-x+i\epsilon)/L})^{(1+\nu)^2}(1 - e^{2\pi i(t+x-i\epsilon)/L})^{\nu^2}} \sum_p \frac{e^{ixk_p - itk_p^2/2}}{p^{2\nu+2}} \left( \frac{p+N}{N} \right)^{2\nu}. \tag{99}$$

Here we have introduced $k_p := \frac{2\pi}{L}(N/2 + p)$. In the TDL this asymptotic expression transforms into the integral

$$\tau_\nu^{\text{SP}}(x,t) \approx \frac{2k_F G^2(-\nu) G^2(1+\nu) e^{2i\nu k_F x - ixk_F + itk_F^2/2}}{(-2ik_F(k_F t - x + i\epsilon))^{(1+\nu)^2}(-2ik_F(k_F t + x + i\epsilon))^{\nu^2}} \times$$

$$\int_{k_F}^{\infty} dk_p \frac{(k_p + k_F)^{2\nu} e^{ixk_p - itk_p^2/2}}{(k_p - k_F)^{2\nu+2}}. \tag{100}$$

The asymptotic value of this integral is dominated by its saddle point, which is present only if $x/t > k_F$. Otherwise, the integral gives exponentially small corrections (the divergence at $k_p \approx k_F$ is superficial, since in that region we return to the soft mode regime). Overall we obtain

$$\tau_\nu^{\text{SP}}(x,t) \approx \sqrt{\frac{2\pi}{it}} \frac{2k_F \theta(x/t - k_F) G^2(-\nu) G^2(1+\nu) e^{2i\nu k_F x + i(x - k_F t)^2/(2t)}}{(-2ik_F(k_F t - x + i\epsilon))^{(1+\nu)^2}(-2ik_F(k_F t + x + i\epsilon))^{\nu^2}} \frac{(x/t + k_F)^{2\nu}}{(x/t - k_F)^{2\nu+2}}. \tag{101}$$

Similarly, we can compute the saddle point contribution in the case $-k_F < x/t < k_F$. To do so, we have to consider one hard hole excitation. That is, we form the new Fermi sea by removing one particle from the position $x/t$ and putting it to the right (or left) edge of the Fermi surface, and resum all soft excitations over this new Fermi sea. The result turns out to be

$$\tau_\nu^{\text{SH}}(x,t) \approx \sqrt{\frac{2\pi}{-it}} \frac{2k_F \theta(x/t - k_F) G^2(1-\nu)G^2(\nu) e^{2i\nu k_F x - i(x-k_F t)^2/(2t)}}{(-2ik_F(k_F t - x + i\epsilon))^{(1-\nu)^2}(-2ik_F(k_F t + x + i\epsilon))^{\nu^2}} \frac{(k_F - x/t)^{2\nu-2}}{(k_F + x/t)^{2\nu}}. \tag{102}$$

By the same token as with the soft modes, the saddle point contribution must be weighted by the multiplicity stemming from all possible integer shifts $\nu \mapsto \nu + m$, $m \in \mathbb{Z}$. Considering more than one hard excitation is possible, but this will produce only subleading corrections. We compare the subleading terms $\delta\tau := \tau - \tau_\nu^{\text{IR}} - \tau_{\nu-1}^{\text{IR}}$ with $\tau_{\nu-1}^{\text{SP}} + \tau_\nu^{\text{SH}}$ in Fig. 7. This figure clearly demonstrates the applicability of our method, and illustrates the regime where our asymptotic is valid: almost immediately outside the light cone, $x = \pm k_F t$. Inside the cone, the asymptotic diverges, something that in fact can be seen on both panels of the figure. All the while, the true Fredholm determinant remains finite. In reality, these singularities are absent.

It is reasonable to assume that the asymptotic behavior in the cone can also be deduced from a soft mode summation. We cannot perform it explicitly here, as the saddle point is located exactly at the Fermi momentum. Nevertheless, keeping in mind that the soft modes should cure the orthogonality catastrophe and invoking dimensional analysis we conjecture the following asymptotic, intended for $x = k_F t$

$$\tau(k_F t, t) \approx \frac{G^2(1-\nu)G^2(1+\nu) e^{2it k_F \nu}}{(C\sqrt{t})^{\nu^2}(4k_F t)^{\nu^2}}. \tag{103}$$

We compare this prediction with numeric results in Fig. (8). We observe that $C$ is approximately 0.8 and varies by a factor of $\sim 2$. We can easily generalize our arguments to the momentum dependent phase shift $\nu(q)$, and reproduce Riemann-Hilbert results for the generalized time-dependent sine kernel as in [77].

# 6 The Quasi-Kernel

As expounded in section 4, the most general expression of the $\tau$-function of a thermal ensemble is in terms of an infinite-dimensional determinant: the Fredholm determinant of the kernel (62). If we can artificially construct some other matrix of *effective* form-factors that has the same determinant, we have obviously found an alternative expression for our observable. Our goal in this section is to find a determinant for which the large $x$ and $t$ asymptotics are described in more elementary functions. We make an argument for a form that comes close to the true kernel in a specific domain, and argue that its error scales as $\mathcal{O}\left(1/\sqrt{t}\right)$ at late times.

In the following, we call the exact result of section (4) the true $\tau$-function, and the proposed approximation the *quasi-$\tau$-function*, and related objects such as the kernel will mirror the notation of the true function, with a tilde. The *quasi-kernel* is given in terms of a function $\eta(k)$ that replaces $\nu$, as if the shift in the fermions' momentum were itself momentum dependent. The advantage is that instead of summing over the infinite set of length-$N$ vectors $\mathbf{k}$, we only need a *single* term with one infinitely-sized vector $\mathbf{k}$. This way, the quasi-$\tau$-function, is the square of an infinite Cauchy-like matrix. Infinite expressions involving $\mathbb{Z}$ here are always understood as first taking $a, b, m \in \{-M, -M+1, \ldots M\}$, for some $M \in \mathbb{N}$ and then sending $M \to \infty$. With that in mind, consider

$$\tilde{\tau} := \prod_{m \in \mathbb{Z}} \frac{f(k_m)}{f(q_m)} \frac{4\sin^2(\pi\eta(k_m))}{L^2 + 2\pi L\eta'(k_m)} \left(\det_{(a,b)\in\mathbb{Z}^2} \frac{1}{k_a - q_b}\right)^2, \tag{104}$$

where for the time being, $\eta(k)$ and $f(k)$ are some smooth, $L$-independent $\mathbb{C} \to \mathbb{C}$ functions[15] without singularities near the real line. $q_a := \frac{2\pi}{L}a$ as before, and the vector $\mathbf{k} = \{k_a\}$ consists of all solutions of the equation

$$e^{ikL} = e^{-2\pi i \eta(k)}, \tag{105}$$

within a distance of $\epsilon = 1/\sqrt{L}$ of the real line. Observe that for the constant choice $\eta(k) = \nu$, the solutions reduce to the momenta with constant shift of the previous section. Formally $k_a$ satisfies $k_a - \frac{2\pi}{L}a = -\frac{2\pi}{L}\eta(k_a)$. We expect $\eta(k) = \mathcal{O}(1)$, therefore as $L \to \infty$, the RHS vanishes, and as a zeroth order approximation we also recover $k_a = \frac{2\pi}{L}a$. The first order approximation would be to enter that solution into $\eta(k_a)$ and solve:

$$k_a = \frac{2\pi}{L}\left(a - \eta\left(\frac{2\pi}{L}a\right)\right) + \mathcal{O}\left(\frac{1}{L^2}\right), \tag{106}$$

which suffices for our purposes. So as $L \to \infty$, we are assured to have the same structure of solutions as in the constant $\eta(k) = \nu$ case, but slightly perturbed into the complex plane. For an illustration in case $\eta(k)$ were $\mathbb{R} \to \mathbb{R}$, see figure 9.

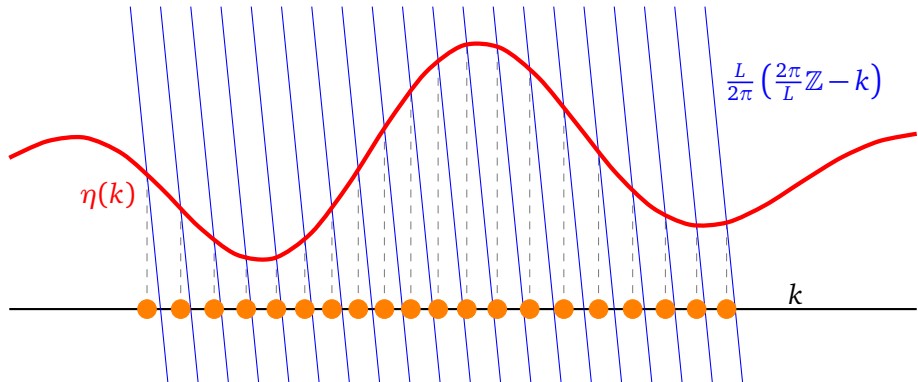

Figure 9: Cartoon of equation $\frac{L}{2\pi}\left(\frac{2\pi}{L}\mathbb{Z} - k\right) = \eta(k)$ for $\eta : \mathbb{R} \to \mathbb{R}$. While the LHS (blue) are uniformly spaced, parallel lines, the solutions (orange dots) are not uniformly spaced. However, as $L \to \infty$, the blue lines become denser and steeper, and the solutions become more uniform and less sensitive to the shape of $\eta(k)$.

Our goal is to equate, as best we can, $\tilde{\tau} = \tau(x, t)$, using this demand to fix $f(k)$ and $\eta(k)$. To do this we first present $\tilde{\tau}$ as a Fredholm determinant. Analogous to the arguments around expression (34), we absorb the prefactors into the determinant and view the squared determinant as the determinant of a squared matrix,

$$\tilde{\tau} = \det_{(a,b)\in\mathbb{Z}^2}\left(\sum_{k\in\mathbf{k}}\frac{f(k)}{\sqrt{f(q_a)f(q_b)}}\frac{4\sin^2(\pi\eta(k))}{(L^2 + 2\pi L\eta'(k))(k - q_a)(k - q_b)}\right) = \det_{(a,b)\in\mathbb{Z}^2}\left(\tilde{A}(q_a, q_b)\right). \tag{107}$$

As before, we tackle the elements of the matrix with complex analysis. The sum over the solutions of (105), is also over the roots of $\cot(kL/2) + \cot(\pi\eta(k)) = 0$. So we divide by this expression and integrate over a contour $\gamma$ for $k$ now complex and continuous, that wraps around all these roots, wherever they may lie. In that case, analogous to way (38) was handled, we must divide out $2\pi i$ times the residue at $h = \frac{2\pi}{L}\eta(h)$,

$$\lim_{k\to h}\frac{k - h}{\cot(kL/2) + \cot(\pi\eta(k))} = -\frac{2\sin^2(\pi\eta(k))}{L + 2\pi\eta'(k)}, \tag{108}$$

---

[15]In related works such as [77], $f(k)$ would be given as some exponent $e^{g(k)}$, here we choose a simpler notation.

where we restored $h \mapsto \frac{2\pi}{L}\eta(k)$ as the function only needs to agree with the residue at the poles. Substituting $2\pi i$ times (108) for the aforementioned quotient allows the presentation of the contour integral,

$$\tilde{A}(q_a, q_b) = \oint_\gamma dk \frac{f(k)}{\sqrt{f(q_a)f(q_b)}} \frac{i}{\pi L \Big(\cot(kL/2) + \cot(\pi\eta(k))\Big)(k - q_a)(k - q_b)}, \tag{109}$$

where, by the same reasoning as before, $\gamma$ need only avoid the poles on the $a, b$-diagonal when $k \to q_a = q_b$, as this results in a net first order singularity. We picture a situation similar to figure 4, except the contour loops around points that may be displaced $\mathcal{O}(1/L)$ from the positions demarcated precisely on the real line. Furthermore, at such a pole, all factors involving $f(k), f(q)$ cancel neatly in precisely the way treated in (40), allowing us to encircle the real line indiscriminately at a distance $\epsilon \propto 1/\sqrt{L}$, for the cost of adding a Kronecker delta, just as in (41). Finally, the last approximation of averaging over the oscillation of the cotangent, described originally in (43), still holds as we assumed $\eta(k)$ is independent of $L$ and may be taken constant inside one period of $\cot(kL/2)$. This allows us to replace division by $\cot(kL/2) + \cot(\pi\eta(k))$ with multiplication by $\pm\frac{i}{2}(1 - e^{\pm 2i\pi\eta(k)})$ for the line displaced from the real by $\pm i\epsilon$. Then for $\epsilon \to 0^+$,

$$\tilde{A}(q_a, q_b) = \delta_{a,b} + \frac{1}{2\pi L \sqrt{f(q_a)f(q_b)}} \times$$
$$\int dk \left[ \frac{\left(1 - e^{2\pi i\eta(k)}\right)f(k)}{(k + i\epsilon - q_a)(k + i\epsilon - q_b)} + \frac{\left(1 - e^{-2\pi i\eta(k)}\right)f(k)}{(k - i\epsilon - q_a)(k - i\epsilon - q_b)} \right]. \tag{110}$$

We learn that the quasi-$\tau$-function and quasi-kernel as in $\tilde{A}(j_a, j_b) = \delta_{a,b} + \frac{2\pi}{L}\tilde{K}(j_a, j_b)$, are again given by the form

$$\tilde{\tau} = \det_{\mathbb{R}^2}\left[\mathbb{1} + \hat{\tilde{K}}\right], \quad \tilde{K}(p, q) = \frac{\tilde{e}_+(p)\tilde{e}_-(q) - \tilde{e}_+(q)\tilde{e}_-(p)}{p - q} \tag{111}$$

by defining the auxiliary functions

$$\tilde{e}_-(q) = \frac{1}{\sqrt{f(q)}}$$
$$\tilde{e}_+(q) = \frac{\tilde{e}_-(q)}{4\pi^2} \int dk \left[ \frac{1 - e^{2\pi i\eta(k)}}{k + i\epsilon - q} + \frac{1 - e^{-2\pi i\eta(k)}}{k - i\epsilon - q} \right] f(k). \tag{112}$$

We are now in a position to identify a candidate for $f(k)$, by judicious comparison to (46). We use this freedom to get the correct dependence on $q, p$ on the factors outside the integral over $k$, in this case absorbing them into $\tilde{e}_-$ instead of leaving them outside $K$. That means

$$f(q) := \frac{e^{-ixq + \frac{i}{2}tq^2}}{\rho(q)}, \tag{113}$$

which incidentally injects the integral over $k$ with the same phase as in (46), however now divided by the density. We see $\sqrt{\rho(q)}e_-(q) = \tilde{e}_-(q)$ exactly, and it also means we must demand $\tilde{e}_+(q) = e_+(q)$, if we wish to have the kernels agree.

This last observation will yield our condition on $\eta(k)$. As an aside, we reformulate the expressions (46) and (112) to remove the dependence on $x$ in the exponents. Note that the total exponent in the kernel is

$$e^{ix\left(\frac{q}{2} + \frac{p}{2} - k\right) - it\left(\frac{q^2}{4} + \frac{p^2}{4} - \frac{k^2}{2}\right)} = e^{-it\left(\frac{1}{4}(q - x/t)^2 + \frac{1}{4}(p - x/t)^2 - \frac{1}{2}(k - x/t)^2\right)}, \tag{114}$$

so we propose to shift all momenta $q - x/t \mapsto q$, $p - x/t \mapsto p$, $k - x/t \mapsto k$. Ultimately, all are integrated over the whole $\mathbb{R}$, so we need not change any limits of integration, and as the denominators depend on the difference $q - k$, the only remnant is a shift in the distribution $\rho(q) \mapsto \rho(q + x/t)$ and $\eta(k) \mapsto \eta(k + x/t)$. It is then natural to think of the $\tau$-function and its quasi-cousin, for a given speed $x/t$ of a ray, as being solely dependent on time $t$. Hence,

$$\tilde{e}_-(q) = e^{-\frac{i}{4}tq^2} \sqrt{\rho(q + x/t)}$$
$$\tilde{e}_+(q) = \frac{\tilde{e}_-(q)}{4\pi^2} \int \left[ \frac{1 - e^{2\pi i \eta(k + x/t)}}{k + i\epsilon - q} + \frac{1 - e^{-2\pi i \eta(k + x/t)}}{k - i\epsilon - q} \right] \frac{e^{\frac{i}{2}tk^2} dk}{\rho(k + x/t)}. \tag{115}$$

We endeavor to have this kernel equal[16] that in (46), where the challenge lies specifically in the identification of $\tilde{e}_+ = e_+(q)$, where we may apply the same shift by $x/t$ in the latter, equivalent to setting $x = 0$. At present the authors do not know of an explicit function $\eta(k + x/t)$ that satisfies this equation in all regimes. We instead make use of some approximations valid at large times. First we use the approximation valid for a smooth function $y(k) : \mathbb{C} \to \mathbb{C}$,

$$\int dk \frac{e^{\frac{i}{2}tk^2}}{k \pm i\epsilon - q} y(k) = e^{\frac{i}{2}q^2 t} \int dw \frac{e^{\frac{i}{2}w(w/t + 2q)}}{w \pm i\epsilon t} y(w/t + q) \approx$$
$$e^{\frac{i}{2}q^2 t} y(q) \int dw \frac{e^{iwq}}{w \pm i\epsilon} = \mp 2\pi i e^{\frac{i}{2}q^2 t} y(q) \theta(\mp q) \tag{116}$$

after substituting $t(k - q) \mapsto w$ in the first equality. The second near equality is valid when we consider the hierarchy of limits $1 \gg \frac{1}{t} \gg \epsilon$ such that $\epsilon t \to 0$. The final equality is found by the Cauchy residue theorem. Applying this formula to $y(k) = \left(1 - e^{\pm 2\pi i \eta(k + x/t)}\right)/\rho(k + x/t)$ from (115), and to $y(k) = 1 - e^{\pm 2\pi i v}$ from (46), we obtain the apparent solution

$$\eta(q + x/t) \approx \begin{cases} -\frac{i}{2\pi} \log\left[1 + \left(e^{2\pi i v} - 1\right)\rho(q + x/t)\right] & \text{if } q < 0 \\ \frac{i}{2\pi} \log\left[1 + \left(e^{-2\pi i v} - 1\right)\rho(q + x/t)\right] & \text{if } q > 0 \end{cases}, \tag{117}$$

which also tells us the limiting behavior $\eta(\pm\infty) = 0$. Unfortunately, the obtained function is problematic around the domain boundary. At face value, it is discontinuous, which violates the assumptions made at the beginning of this section. The size of the discontinuity can be estimated as follows

$$\Delta = \frac{i}{2\pi} \log\left(1 + 2\left(\cos(2\pi v) - 1\right)\left(\rho - \rho^2\right)\right), \tag{118}$$

where $\rho = \rho(x/t)$. At the physically allowed boundaries, $\rho \in \{0, 1\}$, the discontinuity is zero. Those are excluded, and we solve for the maximum over $\rho$ differentially,

$$\frac{\partial \Delta}{\partial \rho} = 0 \Leftrightarrow \frac{\partial}{\partial \rho}\left(\rho - \rho^2\right) = 0 \Rightarrow \rho = \frac{1}{2}. \tag{119}$$

We can use this to put bounds on the magnitude of the discontinuity. Typical distributions have a solution $\rho(k) = \frac{1}{2}$ for some momentum $k$, so we can assume, for certain speeds, the bound is saturated.

$$|\Delta| \leq -\frac{1}{\pi} \log\left(\cos(\pi v)\right). \tag{120}$$

For an illustration of this relation, see figure 10. As $v \to \frac{1}{2}$, the bound diverges to infinity.

---

[16]This is in fact a more strict demand than necessary. We would only need the determinants of the kernels to agree. However, our chosen demand yields an apparent solution.

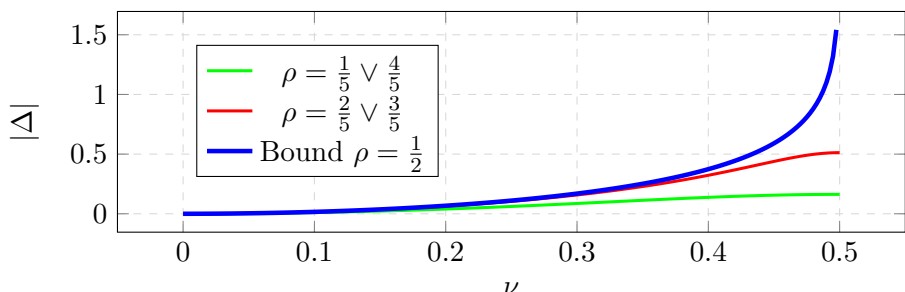

Figure 10: Scaling of the magnitude of the discontinuity $\Delta$ as it depends on the shift $\nu$. Two graphs are produced by equation (118) for non-extremal $\rho$, the value of the probability distribution at the point of inversion $k = x/t$. The final graph is the bound, when $\rho = \frac{1}{2}$, printed in equation (120).

The discontinuity, if left untreated, leads to a faulty expression for $\tilde{\tau}$, due to what is in our interpretation, the remnants of the orthogonality catastrophe. Indeed, let us evaluate the quasi-$\tau$, as defined in (104). It can be split into two parts, the overlap portion, which we term $\mathcal{F}$, and the prefactors, which we call $\mathcal{G}$.

$$\tilde{\tau} = \mathcal{G}\mathcal{F}. \tag{121}$$

The first part,

$$\mathcal{F} := \prod_{m \in \mathbb{Z}} \frac{4\sin^2(\pi\eta(k_m))}{L^2} \left( \det_{(a,b)\in\mathbb{Z}^2} \frac{1}{k_a - q_b} \right)^2. \tag{122}$$

The rest is collected in

$$\mathcal{G} := \prod_{m \in \mathbb{Z}} \frac{f(k_m)}{f(q_m)} \frac{1}{1 + \frac{2\pi}{L}\eta'(k_m)}. \tag{123}$$

We can perform an elaborate set of summations and approximations in order to ply expressions (122) and (123) into the form of a functional on $\eta(k)$, with possibly an explicit dependence on a discontinuity of size $\Delta$ at the obvious point $k = x/t$. The former calculation is performed in appendix B, and the result is expression (214). The latter, we perform now[17]. The same premise as in the appendix, we assume $\eta(k)$ features a single discontinuity.

Let us print the prefactors and expand $f(k_m)$ around $q_m$ to first order,

$$\mathcal{G} \approx \prod_{m \in \mathbb{Z}} \frac{f\left(\frac{2\pi}{L}m\right) - f'\left(\frac{2\pi}{L}m\right)\frac{2\pi}{L}\eta\left(\frac{2\pi}{L}m\right)}{f\left(\frac{2\pi}{L}m\right)\left(1 + \frac{2\pi}{L}\eta'\left(\frac{2\pi}{L}m\right)\right)}. \tag{124}$$

As in the appendix, we consider the log of $\mathcal{G}$, and expand it to first order. There is only a single sum, so higher order vanish in the TDL by the same reasoning as found in (176). We find

$$\log\mathcal{G} = \sum_{m \in \mathbb{Z}} \log\left( \frac{1 - (f'/f)\left(\frac{2\pi}{L}m\right)\frac{2\pi}{L}\eta\left(\frac{2\pi}{L}m\right)}{1 + \frac{2\pi}{L}\eta'\left(\frac{2\pi}{L}m\right)} \right) \approx -\int dq\left[ \frac{f'}{f}(q)\eta(q) + \dot{\eta}(q) \right]$$
$$= \eta(-\infty) - \eta(\infty) + \Delta - \int dk\, \eta(k)\frac{\partial}{\partial k}\log(f(k)) \tag{125}$$

by replacing $\frac{2\pi}{L}m \mapsto k$ and $\sum_m \frac{2\pi}{L} \mapsto \int dk$. We demanded earlier that $\eta(\pm\infty) = 0$, so the total derivative vanishes as well, save for the optional discontinuity $\Delta = \int dk\eta'(k) - \int dk\dot{\eta}(k)$,

---

[17]With $\mathcal{G}$, there are fewer convergence concerns than with $\mathcal{F}$ so we forego the treatment with an intermediate size product of $2M$ terms, as is done in appendix B, and move straight to shorthand $M = \infty$.

which arises because we observe that the discrete sampling of the derivative in (124) is not sensitive to any jump, so its infinitesimal limit is $\dot{\eta}(k)$. Here we retrieve from (216) the definition of the piece-wise derivative (which omits Dirac-$\delta$ peaks).

Above, we made a choice for $f(q)$ in (113), resulting in the identity

$$f'(q) = e^{-ixq + \frac{i}{2}tq^2} \left( \frac{-ix + itq}{\rho(q)} - \frac{\rho'(q)}{\rho^2(q)} \right) = f(q) \cdot \left( i(tq - x) - \frac{\rho'(q)}{\rho(q)} \right). \tag{126}$$

Summarizing,

$$\mathcal{G} \approx \exp\left( \int dk\, \eta(k) \left( i(x - tk) + \frac{\rho'(k)}{\rho(k)} \right) + \Delta \right). \tag{127}$$

Assuming the conventional Fermi-Dirac distribution (23), we note additionally

$$\frac{\rho'(k)}{\rho(k)} = -\frac{k}{T\left( e^{(\mu - k^2/2)/T} + 1 \right)}. \tag{128}$$

In (121), we substitute the explicit forms of $\mathcal{F}$ from (214) and $\mathcal{G}$ for completeness:

$$
\begin{aligned}
\tilde{\tau} \approx & \left( \frac{2\pi}{L} \right)^{\Delta^2} (2\pi)^{\Delta}\, G^2 (1 - \Delta) \cdot \exp\left\{ \int dk\, \eta(k) \left( i(x - kt) + \frac{\rho'(k)}{\rho(k)} \right) \right. \\
& \left. + \int dq \int dp\, \eta'(p)\eta'(q) \log|p - q| + 2\Delta \int dk\, \dot{\eta}(k) \log|k| \right\}.
\end{aligned}
\tag{129}
$$

However, this formula is not useful to us as it does not have a well-behaved $L \to \infty$ limit. This divergence[18], an orthogonality catastrophe in its own right, stems directly from the discontinuity. At this point we might proceed similarly to section 5 and try to perform a soft mode summation of the effective form-factors to compensate this catastrophe. This path appears daunting as it would require redefinition of the effective form-factors to make some 'room', unoccupied momenta, for the soft modes to inhabit. Furthermore, the summation itself can hardly be performed, as we would find ourselves in the exact situation of section 5 when the critical point coincides with the Fermi momentum. Our best guess, similar to the previous, is that performing this resummation would effectively replace $L \mapsto \sqrt{t}$. Below we demonstrate more rigorous way to arrive at this result.

We start by re-examining of our pivotal approximation in (116). We must take care to specify its domain of applicability. If $q$ is close to zero in equation (116), the ignored term $w/t$ in the exponent may begin to dominate the scaling as the contour is expanded. For this reason, the approximation is only valid sufficiently far away from the origin. In order to get a sense of how far, we solve it asymptotically for large $t$ in another way, hearkening back to section 4.1. We use the tautological $y(k) = y(q) + y(k) - y(q)$,

$$
\begin{aligned}
\int dk\, \frac{e^{\frac{i}{2}tk^2}}{k \pm i\epsilon - q} y(k) & = y(q) \int dk\, \frac{e^{\frac{i}{2}tk^2}}{k \pm i\epsilon - q} + \int dk\, \frac{y(k) - y(q)}{k - q} e^{\frac{i}{2}tk^2} \\
& = -2\pi i\, y(q) e^{\frac{i}{2}tq^2} I_{\pm}(-qt, t) + \frac{y(q) - y(0)}{q} e^{\frac{i}{4}\pi} \sqrt{\frac{2\pi}{t}} + \mathcal{O}\left( \frac{1}{t} \right) \\
& = -\pi i e^{\frac{i}{2}tq^2} y(q) \left( \mathrm{erf}\left( -\frac{i+1}{2} q\sqrt{t} \right) \pm 1 \right) + \mathcal{O}\left( \frac{1}{\sqrt{t}} \right),
\end{aligned}
\tag{130}
$$

which confirms (116), when we remember that we have seen in figure 5 that the erf function approaches a step function: similar integrals were treated when we first considered the kernel

---

[18] $\Delta$ is imaginary

of the $\tau$-function. In the second equality of (130), we have used (49) with $x = 0$ and the *stationary phase approximation* at $t = 0$ simultaneously. The latter is allowed because the integrand is regularized. Then we identify (52) in the third equality. The combination $q\sqrt{t}$ as the argument everywhere except in $y(q)$, teaches us that whatever domain we find acceptable for application of (116), its boundaries scale as $|q| \propto 1/\sqrt{t}$.

This way, it is natural to assume the existence of the regularization on some scale $\varepsilon = O(1/\sqrt{t})$, that the correct solution $\eta_\varepsilon(k)$ is a smooth function with the imaginary part[19] shifting from negative to positive via some sigmoid function $S(k/\varepsilon)$. That is, a smooth function such that $\lim_{k \to \pm\infty} S(k) = \pm 1$. Examples are the hyperbolic tangent or error function[20]. Introducing an auxiliary function,

$$\chi(k) := 1 + \left(e^{2\pi i \nu} - 1\right)\rho(k), \tag{131}$$

and shifting back the momentum by $-x/t$, we infer from condition (117),

$$\eta_\varepsilon(k) \approx \frac{1}{2\pi}\left[iS\left(\frac{k - x/t}{\varepsilon}\right)\log\left|\chi(k)\right| + \arg\left(\chi(k)\right)\right]. \tag{132}$$

For small finite values of $\varepsilon$, we appear not to be numerically sensitive to the exact choice. It would be desirable to find a condition that produces a strictly fitting-parameter-free $\eta(k)$, rather than expand the analysis to discontinuous $\eta(k)$. The authors invite any astute reader to propose a candidate.

Now the time has arrived to illustrate the resulting $\eta_\varepsilon(k)$ for some different $T$. The speed $x/t$ only determines the point of inversion in the imaginary part. See figure 11.

From figure 11 comes the interpretation that 'amongst the occupied states', there is a shift of $\nu$, and outside there is none, and on the boundary around the Fermi surface, there is a smooth transition.

Let us discuss now how the choice of the unknown sigmoid $S(k)$ affects the asymptotic. By the reasoning surrounding equality (114), it is enough to consider $x = 0$. In shortened notations, let us define

$$\eta_\varepsilon(k) = u(k) + v(k)S(k/\varepsilon), \tag{133}$$

where we identify from equation (132),

$$u(k) = \frac{1}{2\pi}\arg\left(\chi(k)\right), \quad v(k) = \frac{i}{2\pi}\log\left|\chi(k)\right|. \tag{134}$$

As the scale $\varepsilon$ is $L$-independent, we can safely use results of equation (129), valid in the TDL. With $\varepsilon > 0$, $\eta_\epsilon$ is a continuous function and (129) reduces to

$$\tilde{\tau} \approx \exp\left\{\int dk\, \eta_\varepsilon(k)\left(i(x - kt) + \frac{\rho'(k)}{\rho(k)}\right) + \int dq \int dp\, \eta'_\varepsilon(p)\eta'_\varepsilon(q)\log|p - q|\right\}. \tag{135}$$

In the single integral, from $\mathcal{G}$, we can safely put $\varepsilon \to 0$, while the double integral requires special treatment. We denote it as $\log\mathcal{F}_\varepsilon$,

$$\log\mathcal{F}_\varepsilon = \int dq \int dp\, \eta'_\varepsilon(p)\eta'_\varepsilon(q)\log|p - q|. \tag{136}$$

---

[19]Incidentally for this model, the left and right parts of $\eta(k)$ are complex conjugates, so the real part is connected automatically.

[20]Some $S(k)$ veering out into the complex plane suffices as well, as long as its asymptotes are correct. In fact, equating the traces $\mathrm{Tr}\,K = \mathrm{Tr}\,\tilde{K}$ is an interesting demand on $\eta(k)$, which agrees with ours to first order. It results in a quadratic equation in $\eta(k)$ which is solved, for some $T, \mu, \nu$ by functions that oscillate around the origin but settle down to zero at both infinities.

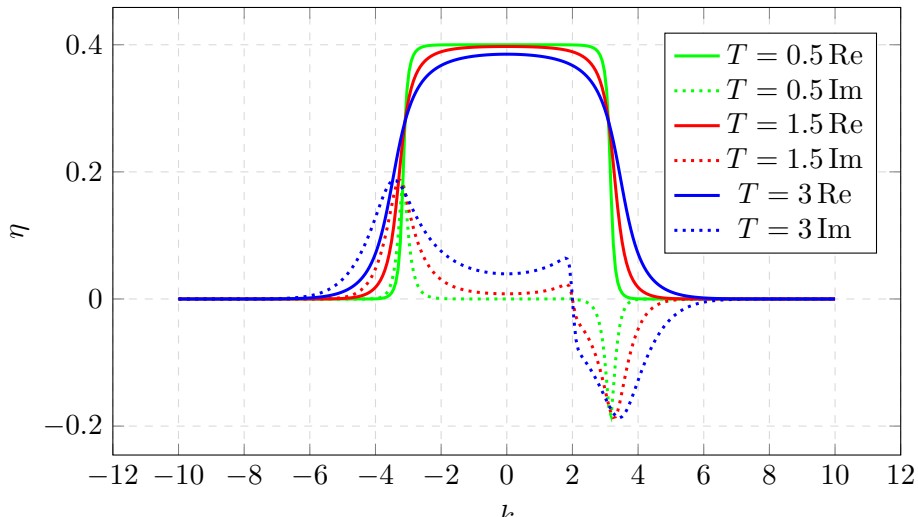

Figure 11: Solutions for $\eta_\varepsilon(k)$ from (132), at varying temperatures in the Fermi-Dirac distribution (23). The left and right limits of $\eta(k)$ happen to be each other's complex conjugate, for this quasi-kernel. The plots were generated with shift $v = 0.4$, speed $x/t = 2$, density $N/L = 1$, and $\epsilon = 0.1$. The sigmoid used is $S(k) = \operatorname{erf} k$. $v$ modulates the amplitude while $x/t$ marks the point of inversion and $\varepsilon$ scales the slope of inversion in the imaginary part.

We wish to study this expression in $\varepsilon \to 0^+$ limit. Care must be taken to understand the divergences in the derivatives of $\eta_\varepsilon(k)$. The dependence on $\varepsilon \propto 1/\sqrt{t}$ will eventually teach us about the exact dependence of $\tau$ on time. To this end, we present it as a sum of three terms

$$\log \mathcal{F}_\varepsilon \approx \underbrace{\int dp \int dq\, \dot{\eta}_\varepsilon(q)\dot{\eta}_\varepsilon(p) \log|q-p|}_{J_1} + \underbrace{2\varepsilon^{-1}\int dp \int dq\, \dot{\eta}_\varepsilon(q)v(p)S'(p/\varepsilon)\log|q-p|}_{J_2}$$

$$+ \underbrace{\varepsilon^{-2}\int dp \int dq\, v(p)v(q)S'(p/\varepsilon)S'(q/\varepsilon)\log|q-p|}_{J_3}. \quad (137)$$

Valid in the limit $\varepsilon \to 0^+$, we can again use piece-wise derivatives,

$$\dot{\eta}_\varepsilon(k) = u'(k) + v'(k)S(k/\varepsilon). \quad (138)$$

We evaluate the limits of $J_1$, $J_2$ and $J_3$, using integration by parts repeatedly. In $J_1$ we can easily take the discontinuous $\eta(k) := \lim_{\varepsilon \to 0^+} \eta_\varepsilon(k)$ limit and obtain

$$\lim_{\varepsilon \to 0^+} J_1 = \int dp \int dq\, \dot{\eta}(q)\dot{\eta}(p)\log|q-p|, \quad (139)$$

which can be equivalently rewritten as

$$\lim_{\varepsilon \to 0^+} J_1 = -\frac{1}{2}\int dq \int dp\, \frac{\eta(q)\dot{\eta}(p) - \dot{\eta}(q)\eta(p)}{q-p} - \Delta \int dk\, \dot{\eta}(k)\log|k|, \quad (140)$$

where an explicit dependence appears on the size of the discontinuity, (172). In order to compute $J_2$ we perform a change of variable $p \mapsto p\varepsilon$,

$$J_2 = 2\int dp \int dq\, \dot{\eta}_\varepsilon(q)v(p\varepsilon)S'(p)\log|q-p\varepsilon|, \quad (141)$$

which in the limit reduces to

$$
\begin{aligned}
\lim_{\varepsilon \to 0^+} J_2 &= 2 \int dp\, S'(p) \int dq\, \dot{\eta}_\varepsilon(q) v(0) \log|q| \\
&= 4v(0) \int dk\, \dot{\eta}(k) \log|k| = 2\Delta \int dk\, \dot{\eta}(k) \log|k|.
\end{aligned}
\tag{142}
$$

Here we have used that by definition of a sigmoid

$$
\int dk\, S'(k) = 2.
\tag{143}
$$

Similarly for $J_3$,

$$
\begin{aligned}
\lim_{\varepsilon \to 0^+} J_3 &= v(0)^2 \int dp \int dq\, S'(p) S'(q) \log|(p-q)\varepsilon| \\
&= \Delta^2 \log\varepsilon + \frac{\Delta^2}{4} \int dp \int dq\, S'(p) S'(q) \log|p-q|.
\end{aligned}
\tag{144}
$$

In (142) and (144), we have substituted the observation that $\Delta := 2v(0)$ by construction. Thus we see how the discontinuity re-appears. Overall we obtain from (137)

$$
\begin{aligned}
\log \mathcal{F}_\varepsilon \approx {}& \Delta^2 \log\varepsilon + \frac{\Delta^2}{4} \int dp \int dq\, S'(p) S'(q) \log|p-q| \\
&- \frac{1}{2} \int dp \int dq\, \frac{\eta(q)\dot{\eta}(p) - \dot{\eta}(q)\eta(p)}{q-p} + \Delta \int dk\, \dot{\eta}(k) \log|k|.
\end{aligned}
\tag{145}
$$

The second term in this expression is unknown, it depends on the original shape of the sigmoid $S(k)$. However, assuming the shape of $S(k)$ is stable in the large time limit, this only scales the eventual quasi-$\tau$ by a constant factor.

$$
C_0 := \frac{\Delta^2}{4} \int dp \int dq\, S'(p) S'(q) \log|p-q|.
\tag{146}
$$

We can illustrate its size. If we take, for example, $S(k) = \operatorname{erf}(k)$, the constant evaluates to

$$
C_0 = -\frac{\Delta^2}{2}(\gamma + \log 2) \approx -0.635 \cdot \Delta^2,
\tag{147}
$$

which is a small constant indeed, for a discontinuity of $\Delta = \mathcal{O}(1)$, as long as $v < 0.48$, approximately, which can be seen from figure 10. We leave the derivation, involving polar coordinates, to the reader.

Continuing our evaluation of $\tau$ as $\eta(k)$ moves towards the discontinuous limit, we must include $\mathcal{G}$ and finally identify[21] $\varepsilon \approx 1/\sqrt{t}$, to find an approximation of the quasi-$\tau$-function.

$$
\begin{aligned}
\log \tilde{\tau} \approx {}& C_0 - \frac{1}{2}\Delta^2 \log t + \int dk\, \eta(k)\left(i(x - tk) + \frac{\rho'(k)}{\rho(k)}\right) \\
&- \frac{1}{2} \int dp \int dq\, \frac{\eta(q)\dot{\eta}(p) - \dot{\eta}(q)\eta(p)}{q-p} + \Delta \int dk\, \dot{\eta}(k) \log|k|.
\end{aligned}
\tag{148}
$$

---

[21]Were we to generalize $\varepsilon \approx c/\sqrt{t}$, this would be tantamount to shifting $C_0 \mapsto C_0 + \Delta^2 \log c$.

This result[22] shows us how $\tilde{\tau}[\eta]$ is a functional of the momentum-dependent shift function $\eta(k)$. For our specific case of interest, $\eta(k)$ is taken from (132),

$$\eta(k) = \frac{i}{2\pi}\Big[ \log(\chi^*(k))\,\theta(k - x/t) - \log(\chi(k))\,\theta(x/t - k)\Big],$$
$$\dot{\eta}(k) = \frac{i}{2\pi}\left[ \frac{(1 - e^{-2\pi i\nu})\rho'(k)}{\chi^*(k)}\theta(k - x/t) - \frac{(1 - e^{2\pi i\nu})\rho'(k)}{\chi(k)}\theta(x/t - k), \right].$$

(149)

With these choices, we have plotted the true exact $\tau$-function against the speed $x/t$, along-side the exponentiation of (148) in figure 12, for various temperatures and densities. It appears $C_0$ is insignificantly small. Ignoring it, the agreement is still very convincing. Where they disagree most, close to the Fermi-surface, is incidentally where $\Delta$, and thus $C_0$ are largest, which can be seen later from figure 13. This means we have a good clue to where this error originates.

Along with $\Delta$, which equation (118) tells us is a function of $\nu$ and $\rho(x/t)$, and therefore implicitly of $T$, $\mu$, and $x/t$, we recognize two other constants required in order to understand the scaling of $\tilde{\tau}$ with time. Defining

$$C_1 := i \int dk\, \eta(k)\Big(x/t - k\Big)$$
$$C_2 := \int dk\, \eta(k)\frac{\rho'(k)}{\rho(k)} - \frac{1}{2}\int dp \int dq\, \frac{\eta(q)\dot{\eta}(p) - \dot{\eta}(q)\eta(p)}{q - p} + \Delta \int dk\, \dot{\eta}(k)\log|k|,$$

(150)

we can rewrite (148) to the concise

$$\tilde{\tau}(x,t) \approx t^{\Delta^2/2}e^{C_1 t + C_2}.$$

(151)

An illustration of the shape of these constants, depending on $x/t$, $T$ and $N/L$ for the thermal Fermi-Dirac distribution, can be found in figure 13. $C_1$ is the dominant constant. From the figure, we learn that the speed of oscillation of the $\tau$-function is not very strongly dependent on temperature, but is on the speed of the ray, conversely the amplitude of oscillation is strongly dependent on temperature, shrinking several orders of magnitude as $T$ increases from $\frac{1}{2}$ to 3.

A few notes on the choices of parameters. We have taken $\nu = 0.4$ in virtually all figures. This is a large, yet still in a sense lopsided shift. There is a symmetry $\nu \mapsto -\nu$ as well as $\nu \mapsto 1 - \nu$, so $\nu = 0.5$ is the largest value of interest, however that being the point of symmetry might obfuscate some of the physics going on. For smaller $\nu$, most signals will simply be of smaller amplitude but qualitatively similar[23]. We have also often set $t = 10$. This is empirically large enough to be in the 'large time limit', as is evinced by i.e. figure 12. At density $N/L = 1$, we can compute the Fredholm determinants faithfully up to times of about $t = 20$. $\tau(x, 20)$ will oscillate more rapidly at a smaller amplitude than $\tau(x, 10)$, in a way predictable thanks to figure 13. Above this rough bound, the interference between the oscillating parts of the kernel must be sampled too finely to allow for efficient numerical computation of the Fredholm. At lower densities, $N/L = 0.3$, the chemical potential $\mu$ is much smaller, and there is less weight in the rapidly oscillating part of the spectrum. In that case, we can push to around $t = 60$. We hope to give a sense of what part of parameter space is readily accessible, and to what scope we have checked our claims. In order to show the agreement in another way, we have plotted

---

[22]When we compare (148) to the alternatively defined (129) we recognize some of the scaling in $\Delta$. We alluded to this correspondence earlier, though their discrepancies are no contradiction. The former is the scaling limit of the overlap of $\tilde{\tau}$ at late times, with increasingly serrated $\eta(k)$. The latter is in the TDL with with a discontinuous $\eta(k)$.

[23]At $\nu = 0$, we are expanding free fermions in their own eigenbasis, and the problem becomes trivial. Only one term contributes to the $\tau$-function, which will then be a phasor in time and space.

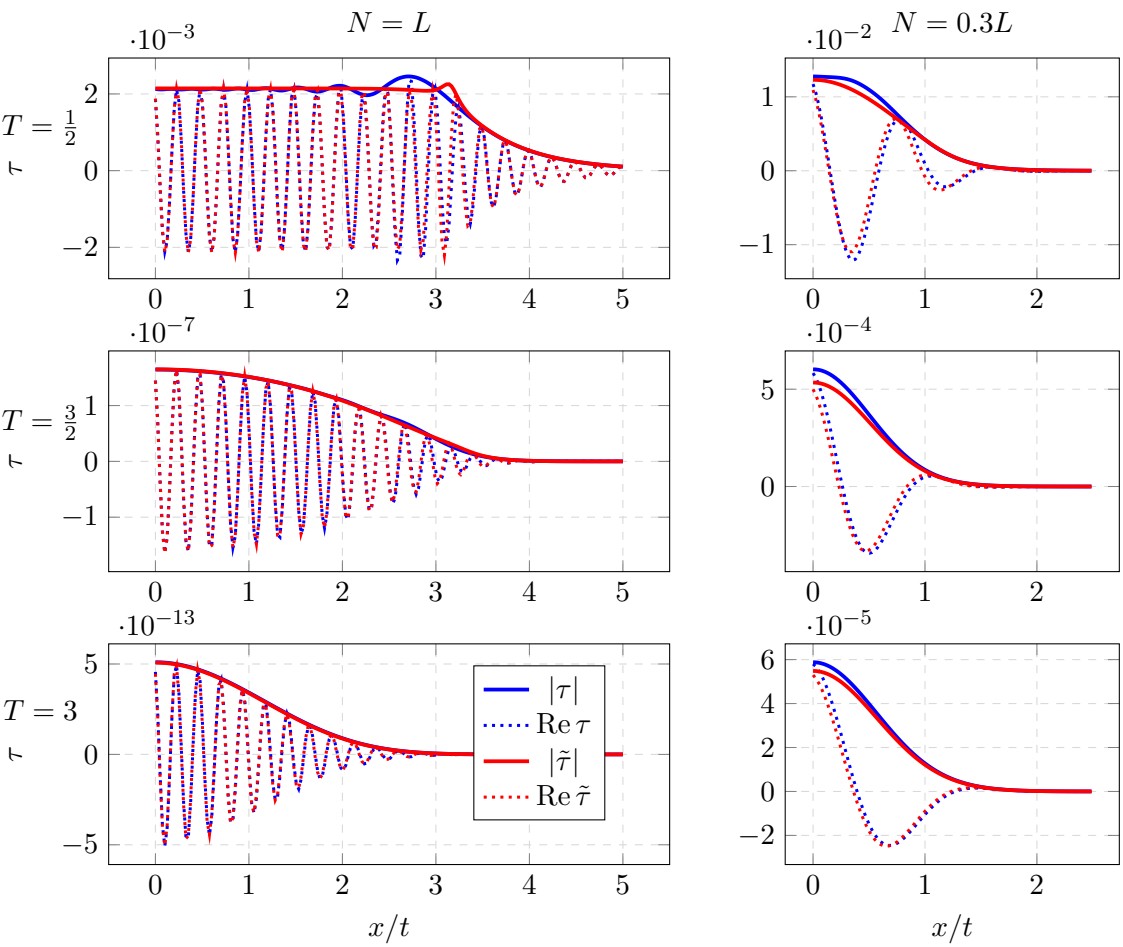

Figure 12: The Fredholm determinant $\tau$-function from (62) and quasi-$\tau$-function $\tilde{\tau}$ from (148) are plotted against speed $x/t$ side by side. The dotted lines are the real part of these complex functions and the solid lines are their moduli. From top to bottom, at temperatures $T \in \{0.5, 1.5, 3\}$, from left to right we have densities $N/L = 1$ and $N/L = 0.3$. We use $\eta(k)$ from (149). The time is $t = 10$, at higher times, the amplitude will be attenuated more. The distribution is Fermi-Dirac, (23), shift $v = 0.4$. Finally, the unknown constant $C_0$ from (147) is set to zero, which ostensibly does not result in a large error.

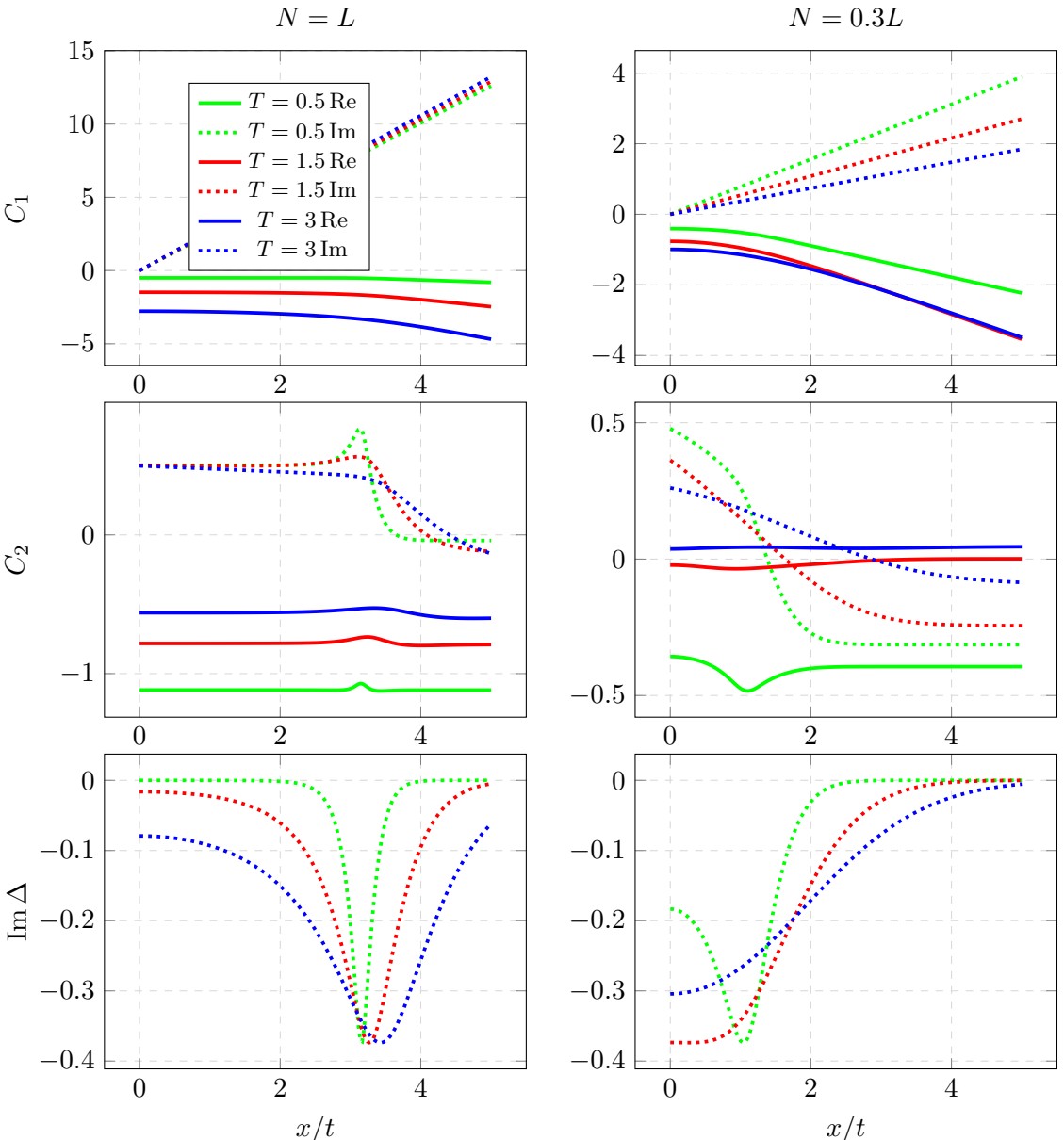

Figure 13: The Constants $C_1$ and $C_2$, as defined in (150), as well as discontinuity size $\Delta$ from (118) are useful to estimate the $\tau$-function, according to (151). These quantities are plotted against speed $x/t$, for the thermal distribution (23) at different temperatures $T \in \{0.5, 1.5, 3\}$. The left column has unit density, the right, $N/L = 0.3$. The shift $\nu = 0.4$. $\operatorname{Re}\Delta = 0$, which is not plotted.

the logs of $\tilde{\tau}$ sans $C_0$ vs. $\tau$ for the accessible range. See figure 14. It would appear to be a pure exponential in time, however, the moduli are off by 11% to 10% along the whole plotted domain and we see the error shrink with time if we include the power law of $t$. By contrast, without the power law, the moduli are off by 30% to 46%, and this error grows with time.

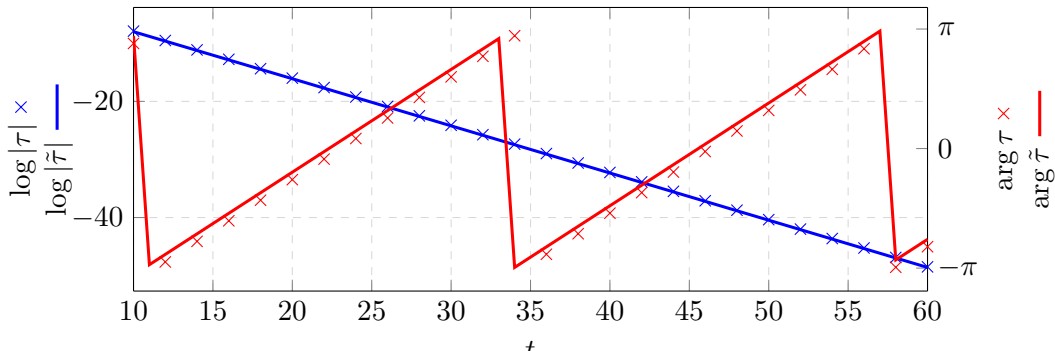

Figure 14: The log and phase of both Fredholm determinant $\tau$-function from (62) and quasi-$\tau$-function $\tilde{\tau}$ from (148) are plotted against time. Solid lines are the approximate $\tilde{\tau}$, marks are the exact $\tau$. Blue is the modulus, red the phase. The distribution is Fermi-Dirac, $T = 1.5$, $x/t = 0.5$, $N/L = 0.3$ and $v = 0.4$. In this case (see figure 13), $\Delta$ is non-negligible, so it is one of the more challenging sets of parameters for our model. The unknown constant $C_0$ from (147) is nonetheless set to zero. The exponent out-competes the power law, hence the line appears straight.

# 7 Hilbert Space Search Algorithm

In other sections of the present work, summations over a multi-fermionic energy eigenbasis were carried out in full, yielding exact expressions of thermal operators. This basis is infinite. In practice, by algorithmically summing a finite subset of terms an appreciable portion of these operators can already be found. In fact, the error stemming from the curtailed sum can be made arbitrarily small in finite time. That is the perspective of this section: we corroborate the exact results with numerical summations that approximate operators such as the $\tau$-function, directly from expressions (1) and (2). This is an entirely independent path and therefore serves as a strong confirmation of those results. Moreover, these numerical techniques can be applied to other systems with fewer exact identities available. For an example, see subsection 7.4 where we apply them to the Bose gas.

## 7.1 Partitioning of Hilbert Space

The first thing that must be understood is the structure of Hilbert space, for we intend to navigate it. In short, the eigenbasis we use is parametrized by sets of disjunct integers, as described in subsection 2. However, 'uniformly' sampling from this set is both unwise and impossible, so we must partition the basis into regimes, which we will later use to carve smart paths that will efficiently produce most of the *weight* of the operators. We will establish nomenclature along the way. The first step is to consider *boxes*, and their *filling*. On the number line, a box is a small finite number of consecutive available single particle quantum numbers, or integers. We give each box an index $a$. Global states can be classified by the amount of quantum numbers, or fillings $\{\phi_a\}$, occupying each of these boxes. When we are denoting these box fillings, it is conventional to assume the omitted boxes away from the origin are empty. If we take box size $\beta = 5$, and for a state with $N = 9$, and fill the boxes starting at quantum number $n = -10$ with consecutively 1,3,3 and 2 quantum numbers, a *microshuffling* corresponding to a multi-fermion state, consists of a choice of *local shuffling* for each box individually. A possible example, which serves as a reference bra state in this section, is

$$\langle \mathbf{g}| = \left| \begin{array}{c} [-10,-6] \\ \circ\,\circ\,\circ\,\bullet\,\circ \end{array} \right| \begin{array}{c} [-5,-1] \\ \bullet\,\bullet\,\circ\,\circ\,\bullet \end{array} \left| \begin{array}{c} [0,4] \\ \bullet\,\bullet\,\circ\,\circ\,\bullet \end{array} \right| \begin{array}{c} [5,9] \\ \bullet\,\circ\,\circ\,\circ\,\bullet, \end{array} \right| \tag{152}$$

corresponding to exact quantum numbers $\{-7 < -5 < -4 < -1 < 0 < 1 < 4 < 5 < 9\}$. The ranges indicate the integers per box.

Other states with the same box filling would be, e.g.

$$
\begin{aligned}
|\mathbf{k}\rangle &= |\circ\circ\circ\bullet\bullet\circ|\bullet\bullet\circ\circ\bullet|\bullet\bullet\circ\circ\bullet|\bullet\circ\circ\circ\bullet| \\
|\mathbf{k}\rangle &= |\bullet\circ\circ\circ\circ|\bullet\circ\bullet\circ\bullet|\circ\circ\bullet\bullet\bullet|\circ\circ\bullet\circ\bullet| \\
|\mathbf{k}\rangle &= |\circ\circ\bullet\circ\circ|\bullet\bullet\bullet\circ\circ|\circ\bullet\bullet\bullet\circ|\circ\circ\bullet\bullet\circ|
\end{aligned}
\tag{153}
$$

$$\cdots$$

The utility of this language is that it allows us to jointly consider ket states that have a similar distribution of quantum numbers, which in turn strongly predicts the size of an overlap with the reference bra state. Concretely, expressions such as (2) and its product identity as a Cauchy determinant show us that overlaps are largest when the distance between the quantum numbers in bra and ket are small, culminating in a maximum overlap, for a given $N$, which is diagonal. Thus if we know the box filling of the bra state, ket states with the same or similar fillings are likely to have a large overlap with this bra state.

We immediately introduce a somewhat unintuitive notation, that nonetheless avoids much ambiguity. We index the boxes, starting with index $a = 0$ on the box containing $n = 0$, and then adding boxes alternating from the left and right of those added before[24]. The indices of the boxes in (152) are, from left to right, $[3, 1, 0, 2]$. If more boxes were added to the left, their indices would count up using odd numbers, and to the right using even numbers. If one wishes to know what the index $a$ of a box is, containing quantum number $n$, the formula is formally,

$$
a = \begin{cases} 2|\lfloor n/\beta\rfloor| & \text{if } n \geq 0 \\ 2|\lfloor n/\beta\rfloor| - 1 & \text{if } n < 0 \end{cases}.
\tag{154}
$$

Conversely, the smallest quantum number of each box, in order of the box index, is $0, -\beta, \beta, -2\beta, 2\beta, \ldots$ When describing box fillings, we list them according to their indices, not their numerical ordering. The main advantages are that there is no ambiguity where the counting begins, all boxes will eventually feature at a predictable point, and all trailing omitted numbers in $\{\phi_a\}$ may be assumed zero, reflect the grouping of quantum numbers near the origin, in turn stemming from the energetic suppression of high momenta. Then the box filling of (152) is $\{\phi_0, \phi_1, \phi_2, \phi_3\}_\beta = \{3, 3, 2, 1\}_5$, subscript holding the box size.

We can straightforwardly generate all states with a given box filling as the Kronecker product of the spaces of local shufflings over each box. With the origin just to the right of the middle $\otimes$,

$$
\{3, 3, 2, 1\}_5 \ni \ldots \circ\circ \left| \otimes \begin{vmatrix} \circ\circ\circ\circ\bullet \\ \circ\circ\circ\bullet\circ \\ \circ\circ\bullet\circ\circ \\ \circ\bullet\circ\circ\circ \\ \bullet\circ\circ\circ\circ \end{vmatrix} \otimes \begin{vmatrix} \circ\circ\bullet\bullet\bullet \\ \circ\bullet\circ\bullet\bullet \\ \circ\bullet\bullet\circ\bullet \\ \circ\bullet\bullet\bullet\circ \\ \bullet\circ\circ\bullet\bullet \\ \bullet\circ\bullet\circ\bullet \\ \bullet\circ\bullet\bullet\circ \\ \bullet\bullet\circ\circ\bullet \\ \bullet\bullet\circ\bullet\circ \\ \bullet\bullet\bullet\circ\circ \end{vmatrix} \otimes \begin{vmatrix} \circ\circ\bullet\bullet\bullet \\ \circ\bullet\circ\bullet\bullet \\ \circ\bullet\bullet\circ\bullet \\ \circ\bullet\bullet\bullet\circ \\ \bullet\circ\circ\bullet\bullet \\ \bullet\circ\bullet\circ\bullet \\ \bullet\circ\bullet\bullet\circ \\ \bullet\bullet\circ\circ\bullet \\ \bullet\bullet\circ\bullet\circ \\ \bullet\bullet\bullet\circ\circ \end{vmatrix} \otimes \begin{vmatrix} \circ\circ\circ\bullet\bullet \\ \circ\circ\bullet\circ\bullet \\ \circ\circ\bullet\bullet\circ \\ \circ\bullet\circ\circ\bullet \\ \circ\bullet\circ\bullet\circ \\ \circ\bullet\bullet\circ\circ \\ \bullet\circ\circ\circ\bullet \\ \bullet\circ\circ\bullet\circ \\ \bullet\circ\bullet\circ\circ \\ \bullet\bullet\circ\circ\circ \end{vmatrix} \otimes \right| \circ\circ\ldots
\tag{155}
$$

Although descriptive, the actual box filling of a state is often less interesting than its *excitation profile* or particle-hole-profile as compared to a reference state. In the algorithms used,

---

[24]This is evocative of a well-known bijection between the integers and natural numbers.

a box filling is extracted from a bra state, and ket states are found by first choosing an excitation profile over this filling. Any excitation profile has a *level*, $\lambda$, which is the number of particles ($+1$) as well as the number of holes ($-1$) created. So, with the state in (152) as a reference, any state with box filling $\{4, 3, 1, 1\}_5$ has a level $\lambda = 1$ excitation with excitation profile $\{+1, 0, -1, 0\}_5$, while a filling $\{4, 3, 0, 2\}_5$ has a level $\lambda = 2$ excitation with respect to the reference, with profile $\{+1, 0, -2, +1\}_5$. Note that holes or particles may be in the same box, but may not result in a negative filling or one that exceeds the box size.

Moving on, consider the placement of the particles in these excitation profiles. If they are inside boxes that were not previously empty in the reference state, we need not increase the length of the sequence in order to describe the excitation. If, however, they are moved farther out on the number line, we must add a *pad* to the excitation, which tracks how many boxes must be added to either side in order to describe the excitation. We always add positive and negative boxes simultaneously, and thus always add an equal number of indices to the sequence. On top of (152), (156) has a pad 1, level 1 excitation, and results in the excitation profile

$$
\left| \begin{array}{c} [-15, -11] \\ \circ\circ\circ\circ\circ \end{array} \right| \begin{array}{c} [-10, -6] \\ \circ\circ\circ\bullet\circ \end{array} \left| \begin{array}{c} [-5, -1] \\ \bullet\bullet\circ\circ\circ \end{array} \right| \begin{array}{c} [0, 4] \\ \bullet\circ\circ\circ\bullet \end{array} \left| \begin{array}{c} [5, 9] \\ \bullet\circ\circ\circ\bullet \end{array} \right| \begin{array}{c} [10, 15] \\ \circ\bullet\circ\circ\circ \end{array} \right| \mapsto \{-1, 0, 0, 0, +1, 0\}_5,
$$
(156)

with penultimate and ultimate elements of this sequence corresponding to the rightmost and leftmost box of the LHS, respectively. The pad of more complicated excitations is the maximum pad among all particles. It is clear that two excitation profiles with different pads cannot contain the same state, so this forms a partition of Hilbert space.

A final refinement is the concept of a *tier*. Empirically, varying the exact placement of quantum numbers inside a single box in the ket state can change the overlap with a reference bra state by an order of magnitude. As more boxes are shuffled locally, this effect compounds, such that even two states with the same box filling can have a virtually vanishing overlap with each other, if their local shufflings are mismatched. Conversely, two states with matching local shuffling on most boxes but a single particle-hole excitation with a high pad might have a larger overlap. In order to account for this hierarchy, we classify local box shufflings, per box, into tiers, starting at tier 0. These tiers are specific to a certain box, box filling, and reference state. Roughly, the local shufflings in a tier all result in an overlap that is at least[25] some factor $\epsilon < 1$ smaller than those in the next tier down, i.e. the lower the tier, the larger the overlap. The box at index 1, with filling 3, $\phi_1 = 3$, might have the following tiers, increasing to the right,

$$
\text{tiers}_1^3 = \left\{ \left| \begin{array}{c} 0 \\ \bullet\bullet\circ\circ\bullet \end{array} \right|, \left| \begin{array}{c} 1 \\ \circ\bullet\bullet\circ\bullet \\ \bullet\circ\circ\bullet\bullet \\ \bullet\circ\bullet\circ\bullet \\ \bullet\bullet\circ\bullet\circ \end{array} \right|, \left| \begin{array}{c} 2 \\ \circ\bullet\circ\bullet\bullet \\ \bullet\bullet\bullet\circ\circ \end{array} \right|, \left| \begin{array}{c} 3 \\ \circ\circ\bullet\bullet\bullet \\ \circ\bullet\bullet\bullet\circ \\ \circ\bullet\bullet\circ\bullet \end{array} \right| \right\},
$$
(157)

while the tiers for the box at index 2 with $\phi_2 = 3$ might be completely differently divided. Also, when box 1 is excited, and has $\leq 2$ or $\geq 4$ particles, a new set of tiers must be constructed. Tier 0 always exists, and may only contain the trivial shuffling if the filling is 0 or $\beta$. Higher tiers may or may not exist, but the union of tiers is always all possible local shufflings. In the case $\beta = 5$, $\phi_a = 3$, there are $\binom{\beta}{\phi_a} = 10$.

It is worth stressing that, contrary to level and pad, the population of the tiers is not a priori obvious and is a computational choice made on-the-go, based on incomplete information. The exact method is explained in the next subsection. It will depend strongly on the exact quantum numbers of the reference state. Moving forward, we assume for each box and local filling of

---

[25]We choose not to have empty tiers, although the factor between tiers might be $\mathcal{O}(\epsilon^2)$ or smaller.

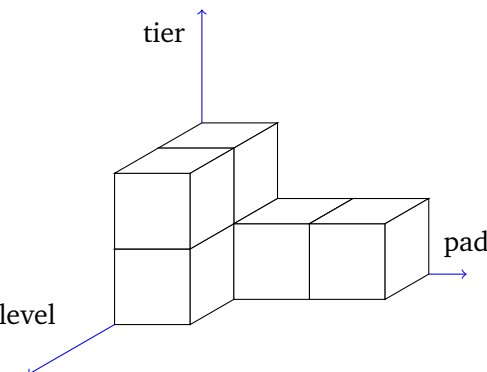

Figure 15: Visualization of the three-dimensional Young diagram partitioning of free-fermionic Hilbert space. We use such diagrams to visualize exploration of the space. Each block is defined by a level, pad and tier, and represents a set of states, disjunct from other blocks.

that box, tiers have been constructed. Then we can augment the definition to *global tiers*. For a given excitation profile, we may ask for all global shufflings at a certain global tier. This simply means the sum of local tiers: the sum over all boxes of the tier to which the local shuffling in that box belongs. Box 1 might be filled with a local shuffling from its tier 0, box 2 from its tier 1, box 3 from its tier 1, and box 4 from its tier 2: then the global tier is 4. There may be multiple states with this exact distribution of local tiers, because any local tier might contain more than one shuffling as in (157), and their tensor product is contained in the space of that global tier.

This is the reason for starting tiers at 0: the highest weight (tier 0) local shufflings do not contribute to the global tier for any box. Notably, in our notation, we suppress the infinite sequence of empty boxes to the left and right of the origin. These all have the trivial filling, and thus only tier 0, so their inclusion in the sum does not affect the global tier, allowing it to be well-defined. Thus we may think of the tier as a kind of particle in its own right: for the microshufflings at global tier 3, we may distribute the local tiers in 3 different boxes, or 2 in one box and 1 in another box, or all in the same box, as long as the boxes indeed have that many tiers to offer. There is a maximum tier, which is the sum of the maximal tiers of all boxes at a given excitation profile.

Hilbert space is said to be partitioned into *blocks*: a block is a set of microstates corresponding to a particular level, pad and global tier. The level and pad comprise many excitation profiles, and when given a tier, each profile represents a number, or *salvo* of states. The union of all these states forms the block. The three characteristics or dimensions aid in the visualisation of it being a cube. When exploring Hilbert space, they neighbor each other incrementally, like a 3-dimensional Young diagram. This is visualized in diagrams such as figure 15.

In conclusion, navigation of free-fermionic Hilbert space is done as follows. The preparation:

- Choose a reference state with specific quantum numbers.

- Divide the number line into boxes of size $\beta$ and determine their reference filling.

- Construct local tiers for each box $a$, for each filling $\phi_a \in \{0, 1, \dots, \beta\}$ of that box.

- Then, repeatedly, a set or salvo of states in Hilbert space is found by the steps:

  - Choose a block: an excitation level, a pad for the box of the outer excited particle, and a global tier.

- Choose an excitation profile corresponding to this pad and level, if available[26].

- If the global tier chosen does not exceed the maximum global tier of this excitation profile, consider all microshufflings at this excitation profile and global tier in unison.

The set provided at the end of this protocol is unique and disjunct from the sets found for any other block or excitation profile in this block. We adopt the view that we expect all states corresponding to the microshufflings of this set to have mutually similar overlaps with the reference state. The algorithmic procedure of making all the choices in the above checklist, is described in subsection 7.2. The goal is an efficient approximation of the $\tau$-function.

On that note: our guiding ansatz is a modification of (14). If we only sum a subset $\mathcal{S}$ of the available $|\mathbf{k}\rangle$, as the weight is nonnegative, the result is some number $s$,

$$s := \sum_{\mathbf{k} \in \mathcal{S}} |\langle \mathbf{g} | \mathbf{k} \rangle|^2 \in [0, 1), \tag{158}$$

termed the sum rule. The more weight $|\langle \mathbf{g} | \mathbf{k} \rangle|^2$ can be added, the better. From (1), the $\tau$-function is a weighted average of phasors, each with unit modulus. In the worst case scenario, where they all constructively interfere, the error of the $\tau$-function is exactly the missing weight $1-s$. However, this scenario is only true (modulo periodicity) at $(x, t) = (0, 0)$, and at all other points in space-time, the true error is considerably smaller, as the omitted phasors would also destructively interfere. The error satisfies

$$1 - s = \sum_{\mathbf{k} \notin \mathcal{S}} |\langle \mathbf{g} | \mathbf{k} \rangle|^2 \geq \left| \sum_{\mathbf{k} \notin \mathcal{S}} |\langle \mathbf{g} | \mathbf{k} \rangle|^2 \, e^{\sum_a \left( i x (g_a - k_a) - i t (g_a^2 - k_a^2) \right)} \right|, \tag{159}$$

by the triangle inequality. With judicious choices of the search direction, this error can be expediently minimized.

## 7.2 Algorithm Search

Now that the structure and nomenclature of the Hilbert space partitioning has been established in the previous subsection, we explain at a high level how our algorithm goes about navigating these partitions to efficiently calculate a large part of the weight of the desired operators.

It is evident that even inside $\mathcal{S}$, not all $|\mathbf{k}\rangle$ are created equal, but contribute different weight. The aim of the algorithm is to incrementally explore more of Hilbert space, adding pads and levels as needed, constructing and scoring excitation profiles and local tiers progressively. When these blocks are established, they can be used to generate more states as needed. Prepared blocks are not immediately depleted. In fact, also the scoping out of the tiers is done by generating states and evaluating their weight. These states need not be revisited later. When a satisfactory sum rule $s$ is reached, the algorithm terminates.

From here, we describe the various objects and routines used in the Python code. Functions and important variables are typeset in **bold**. A summary:

| Name: | **FFstate** |
|---|---|
| Purpose: | Class object that approximates the $\tau$-function efficiently. |
| Inputs: | The quantum numbers of the bra state **qn**, the momentum shift **nu**, $\beta$ the box size, **B** and the system length, **L**. |
| Outputs: | None, however stores many important results as class variables. These are called with the prefix **self.** |
| Source: | https://github.com/DMChernowitz/Free-Fermionic-Hilbert-Search |

---

[26]If the level is larger than the number of particles, or possible holes at this pad, there are no excitation profiles available.

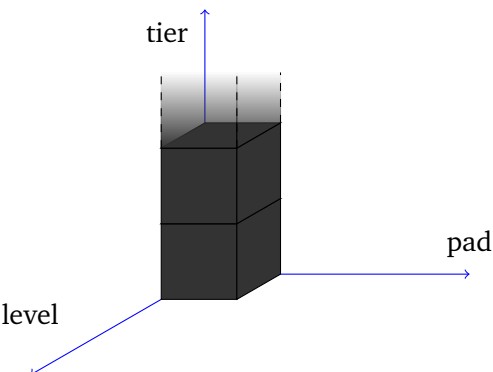

Figure 16: Visualization of the progress of the algorithm, when **ground_shuffle** has been executed. The dark column above the origin is level zero, pad zero, and all states, and thus all tiers for zero pad and zero level are treated by this method. None of these need revisiting, so we may think of all these blocks as being 'inactive'.

To experiment with this routine, it and all dependencies can be found in the 'FFstate.py' file at the URL in the above table. A typical use of the code could be:

```
1   state = FFstate([-7, -4, -3, -2, -1, 0, 1, 2, 3, 4, 6], 0.4, 4, 11)
2   state.hilbert_search(0.95)
3   plt.plot(state.tau)
```

This code plots (the real part of) the 11-fermion $\tau$-function, $\nu = 0.4$, with a 0.05 error margin, over $x \in [0, 10]$, for rays $t/x \in \{0, \frac{1}{4}, \frac{1}{2}, \frac{3}{4}, 1\}$.

As an initialization, a reference state object is created, an instance of the class **FFstate**, in the spirit of (152). It has predetermined quantum numbers **qn**, those defining $\langle \mathbf{g} |$. This object is central to this section and its subroutines or methods comprise the algorithm. Its box size $\beta$ is the variable **B**. Its filling is stored for the *inner* boxes, those non-empty in $\langle \mathbf{g} |$ (in pairs on the left and right together), as well as **self.U**: a nested list, per box, of all possible local shufflings, at present filling. This is provided by a routine called **ind_draw**, described in appendix A.2, and will be used by other internal subroutines. The $\tau$-function for a discretized mesh of $(x, t)$-coordinates, is initialized as a complex **Numpy** array with zeros for elements, as no weight has been added yet.

At this point, we are interested in constructing the tiers for the boxes at present filling $\{\phi_a\}$, without particle-hole excitations. To this end, the first states are produced[27], which comprise all possible microshufflings with the original filling. See figure 16 for a diagram of the progress.

This is performed by the **FFstate** method **ground_shuffle** which in turn calls the recursive **in_box_shuffle**, both methods of the **FFstate** class, and the latter described in appendix A.4. In short, it recursively glues local shufflings together in the spirit of equation (155). For each state $| \mathbf{k} \rangle$, **ground_shuffle** calls the method **update_tau**, from A.10, where the weight $|\langle \mathbf{g} | \mathbf{k} \rangle|^2$ is calculated, and the contribution to $\tau_\mathbf{g}(x, t)$ is stored. **in_box_shuffle** also indexes each local shuffling. This means that each state is returned accompanied by a unique list of indices $\{j_0, j_1, \ldots, j_{l-1}\}$, one index for each of $l$ inner boxes. The utility of this is to allow **ground_shuffle** to fill a high-dimensional tensor termed the **weightrix** with the weights. The

---

[27]There is a check, if the *complexity* (calculated in **self.complexity** as the number of unexcited microshufflings) of the bra state is too high, the code will not begin lest it spend too long in preparation. Often, the maximum allowed complexity will be around 400K states.

dimension of the weightrix is equal to the number of filled boxes, and the size of the $a^{\text{th}}$ dimension is the number of local shufflings $\binom{\beta}{\phi_a}$ in box $a$. Then each element of the **weightrix** corresponds to exactly one state and holds the weight $|\langle \mathbf{g}|\mathbf{k}\rangle|^2$. When all weights have been entered, we can construct the first local tiers. We find a normalized proxy for the importance of a local shuffling of a box by integrating out the shuffling of all other boxes. If the weightrix is denoted $W$, then the proxy for box $a$, shuffling $j_a$ is given by

$$w_{j_a}(a) = \frac{\sum_{j_0,...,j_{a-1},j_{a+1},...j_{l-1}} W_{j_0,...,j_{l-1}}}{\sum_{j_0,...,j_{l-1}} W_{j_0,...,j_{l-1}}}, \tag{160}$$

calculated in Python by the routine **disect**, discussed in appendix A.5. With this proxy, the tiers for these inner boxes are constructed. The algorithm uses a preset constant $\epsilon = 0.3$ to separate the local shufflings into tiers. This choice of $\epsilon$ is found empirically to produce the quickest search in combination with a box size of $\beta = 4$, and between 10 and 100 particles. The exact division into tiers is performed by subroutine **sortout** from A.6. In short: we attempt to bin their proxy weight $w_{j_a}(a)$ on a log scale into bins of size $\log \epsilon$. Each bin is a tier.

The tiers are stored in the variable **self.BT** of the **FFstate** class, created by **ground_shuffle**. This is a highly nested list. The first index is the level $\lambda$, the second is the box index $a$, and the third is the tier. At this depth, one finds a list of local shufflings (themselves lists) of the quantum numbers in that box: these shufflings comprise the tier. **self.BT** will be used continuously in the rest of the search algorithm. At the same time, we create an variable of the **FFstate** class, **self.p_h_profiles**, with a similar nested structure. The purpose of this variable is to hold any and all admissable excitation profiles. The first index of **self.p_h_profiles** denotes the level, the second denotes the pad, and the third enumerates the possible excitation profiles at this level and pad. For each such profile, an entry consists of a list of the following three objects: the profile, described as a list of integers in the style of (156), a list of the *extremal* indices of this profile, and a *walking*[28] variable for this profile. An extremal index is a box index such that the box is filled by a number of particles not found at a lower pad or level. If the level is $\lambda$, only boxes with $\lambda$ particles or holes are extremal. However, if besides level $\lambda > 1$, the pad is nonzero, then only the outer two boxes (or last two indices) can be extremal, and only if their box has $\lambda$ particles, because putting $\lambda$ particles in lower-indexed boxes occurs at a lower pad already. It should be clear that an excitation profiles has zero, one or at most two extremal indices, one for particles and one for holes. At any level, if the particles or holes are divided over more than one box, they don't contribute an extremal index. The utility of the extremal indices is directly tied to the creation of local tiers at nonzero level, and will feature in the following stage of the algorithm: exploration.

Finally, **ground_shuffle** constructs the **self.blocks** and **self.cart** variables for the **FFstate** instance.

The **self.blocks** is a nested list and has first, second and third indices for level, pad and tier and holds an instance of **self.deepen_gen** as each element, a generator for salvos (small sets) of states in that block. This generator is explained in A.7, but it is similar to the recursive generation of **ground_shuffle**. Recall that just because we have prepared the generator, it does not mean we have prompted it to produce all states in the blocks, and most have not yet been evaluated.

The related **self.cart** holds a sorted list of entries, each corresponding to a block, an address in Hilbert space. The intention is that the top entry of **self.cart** is the block where to search for states next, i.e. from which generator in **self.blocks** we expect the most new weight per state

---

[28]This **walking** variable holds the sum of unsquared overlaps $\langle \mathbf{g}|\mathbf{k}\rangle$ of all states $|\mathbf{k}\rangle$ produced from this profile, as the algorithm progresses. This is useful information when trying to understand the field theory limit of these states, where many microshufflings correspond to a single field theory state, labeled by the excitation profile.

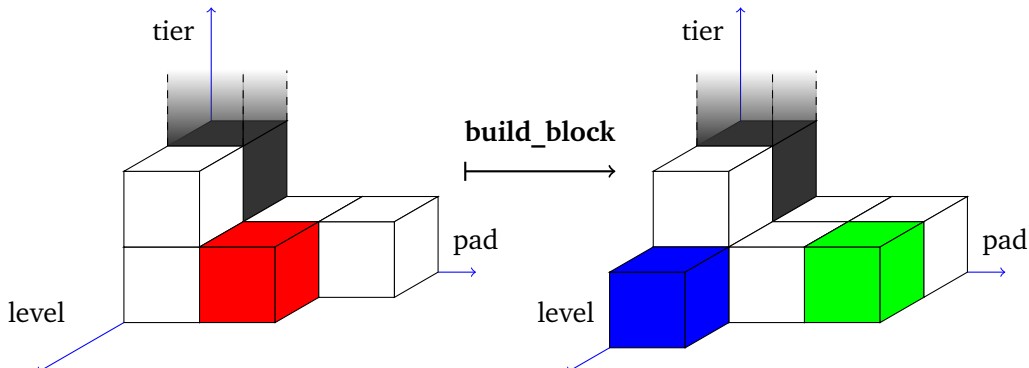

Figure 17: Visualization of the three-dimensional Young diagram partitioning of free-fermionic Hilbert space, at some point during the search. The red block is the current block the algorithm uses to produce states. When **build_block** is called, if still unexplored, it explores the column of the green block, and if **self.peps** $\leq 1$, also the blue, constructing their excitation profiles and the local tiers they use. Then it adds the green (blue) block(s) to **self.cart**.

to lie. Entries in **self.cart** are each lists, whose first entries are the weight per state, produced over the most recent salvo from an instance of **deepen_gen**, and the rest of the list is the level, pad, and tier of that block so it can be found in **self.blocks**.

After **ground_shuffle**, we jump-start exploration with the **build_block** method of the **FFstate** class, which is intended to scout just beyond the boundaries of states already produced. **build_block** is called with the arguments of the current level and pad, at this stage in the algorithm, they are both zero. **build_block** calls another method named **explore**, which prepares, or explores, a new column of blocks: this means to construct the necessary local tiers and possible excitation profiles in the column of blocks. If the current pad is the maximally explored pad at this level, **build_block** calls **explore** at the current level and one pad higher[29]. If the current level is the maximally explored level, and the pad is larger or equal to **self.peps**$\lambda$, **build_block** calls **explore** at one level higher and zero pad[30]. The variable **self.peps** is usually 1 or 2, and the use of this condition prevents the algorithm exploring high levels before putting low numbers of particles at appreciable distance from the origin, which empirically carry more weight. High level exploration is extremely expensive in computation time, for simple combinatoric reasons. For a schematic of the **build_block** rule, see figure 17. Using this procedure, we generally avoid exploring blocks unless we expect them to be used soon after.

The code used in **explore** is printed in A.11. In short, it functions as follows. **explore** produces all states at the new pad and level with a particular characteristic: in extremal boxes, we allow all local shufflings, as the tiers are not yet known. For any excitation profile, most if not all boxes are non-extremal, so their tiers are known, and we only allow tier 0 local shufflings in these. The weights of these states allow us to construct tiers for extremal boxes, again by means of a **weightrix**.

Though carrying the same name as the variable in **ground_shuffle**, the structure of the **weightrix** is quite different at this stage, because we only need to determine tiers for extremal

---

[29]It may occur, for instance at level 1, that we call **explore** on a level and pad such that lower levels at this pad have not been explored in this instance of the **FFstate** object, though the resulting tiers from lower levels are needed. Therefore, **explore** first calls itself on one level lower, at this pad.

[30]It may happen that the level $\lambda$ is too high to relocate all quantum numbers inside the allowed boxes at a certain pad. There is a fail-safe, until there are at least some states available in the generator of **self.deepen_gen**, **build_block** keeps increasing the pad, up to a maximum pad determined beforehand.

boxes.

In case **explore** is called with a pad of zero, the **weightrix** is a nested list of lists. Let the level be $\lambda$. The first index of the **weightrix** takes values 0 or 1 and determines whether we are describing particles or holes, respectively. The second index $a$ runs over the original inner boxes. The third index $j_a$ enumerates the possible local shufflings in box $a$ when it is extremal: with filling $\phi_a + \lambda$, in the case of particles, or $\phi_a - \lambda$, in the case of holes. There may be no such shufflings, if the $\lambda > \phi_a$ for holes or $\lambda > \beta - \phi_a$ for particles, then that $a$ holds an empty list. The entry of the **weightrix** $W_{j_a}^0(a)$, is the sum of the weights $|\langle \mathbf{g}|\mathbf{k}\rangle|^2$ of all states produced by **explore** that have $\lambda$ particles in box $a$, while also having the local shuffling corresponding to $j_a$ in that box. The same is true for $W_{j_a}^1(a)$ but with holes. If their excitation profile has an extremal box, states contribute to a term in the **weightrix**, and a few states, for instance with an excitation profile $\{-\lambda, 0, 0, +\lambda\}$, contribute to both $W_{j_0}^1(0)$ and $W_{j_3}^0(3)$.

If, on the other hand, if **explore** is called with the pad nonzero, then extremal boxes can only be the outer two (those with the highest indices), and only with $\lambda$ particles in them, not holes. It follows the **weightrix** is one dimension smaller, we omit the first index, and the second index $a$ can only run over two values.

The proxy weight for an extremal box is simply the $W$ value, normalized over the available local shufflings, i.e. index $j_a$. Dropping the superscript, which only applies for zero pad,

$$w_{j_a}(a) = \frac{W_{j_a}(a)}{\sum_{j_a} W_{j_a}(a)}. \tag{161}$$

Using these weights, **explore** calls the method **sortout** again, treated in A.6, in order to update the **self.BT** variable with the newly minted tiers for the extremal boxes at this level and pad. It also adds a **deepen_gen** instance at this level and pad, and with input tier zero, corresponding to global tier one, to the **self.blocks** variable of the **FFstate** object. With this generator, the rest of the states in the block, not used during exploration, can be yielded.

All the above is executed upon initialization. After the first calls to explore, there are some generators in **self.blocks** and block addresses in **self.cart**. Now the undetermined stage of the algorithm begins.

On our instance of the **FFstate** class, we call the method **hilbert_search**, with two inputs: the desired sum rule, and the maximum run time of the search. If the latter is not given, it defaults to $10N$ seconds. If either are reached, as checked between exploration or salvos of states, the search is terminated. **hilbert_search** has the following structure: While the desired sum rule and time are not met, and the cart is non-empty, repeat the following steps:

- sort **self.cart** descending by the first entry of each element: the average weight over the last salvo of states from the corresponding block.

- The top block of **self.cart** becomes the current block. With its level, pad and tier:

    - run **build_block** to explore if necessary.
    - Run **stack_block** to add more tiers if necessary.

- If the previous two steps added any blocks tot the **self.cart**, we restart the loop, otherwise, we run the method **harvest**.

If the cart is empty, more blocks on the periphery are explored.

If the current block is atop its column, **stack_block** increases its height by adding the next tier **deepen_gen** to **self.blocks**, and its coordinates to **self.cart**. For a schematic representation, see figure 18.

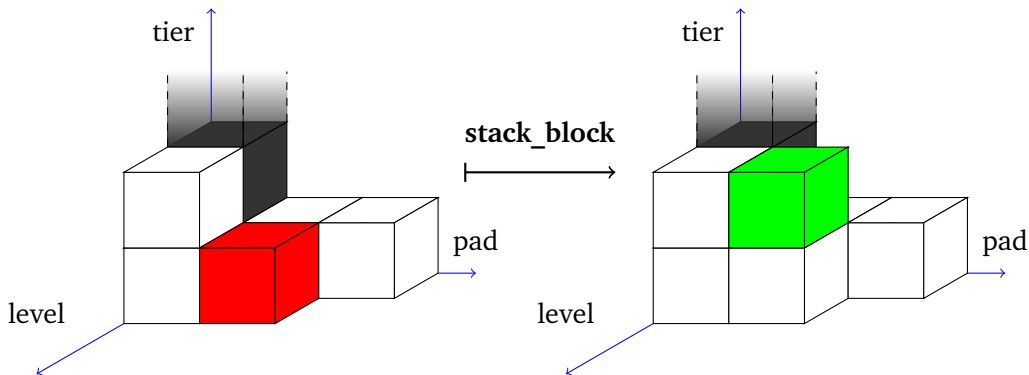

Figure 18: Visualization of the three-dimensional Young diagram partitioning of free-fermionic Hilbert space. The first time **stack_block** is called on the current block (shown in red), it adds the green block to **self.cart** and **self.blocks** in the **FFstate** instance, so higher tier states may be produced at this level and pad.

In turn, **harvest** extracts salvos of states from the **deepen_gen** generators in the current block in **self.blocks**. When the average weight found dips below the second entry of **self.cart**, we return to the main loop in **hilbert_search**, continuously optimizing our search area.

**hilbert_search** remembers its progress through the use of generators, such that once it terminates, it can be called again on the same **FFstate** instance to add more time and continue the search. The main code of the algorithm is printed below. Some of the **FFstate** class methods, as well as general auxiliary functions, are omitted, those whose code is treated in the appendices A. While running, the algorithm prints pairs of numbers indicating the level, followed by the pad, that are currently being explored.

```python
from numpy import zeros, pi, array as arr, sin, exp, prod, linspace, newaxis, kron
from numpy.linalg import det
import matplotlib.pyplot as plt
from time import clock

class FFstate:
    def __init__(self,qn,nu,B,L,eps=0.3,ceps=0.2,maxcomplex=200000):
        self.states_done = 0
        self.clo = clock()
        self.N = len(qn)
        self.L = L
        self.B = B
        if B >=self.N:
            print('box too large')
        self.nu = nu
        self.maxorder = self.L
        self.qn = qn
        self.abra = arr(self.qn)[:,newaxis]
        shiftqn = arr(self.qn)-nu
        self.pbra,self.ebra = -sum(shiftqn),-sum(shiftqn*shiftqn)
        self.presin = (sin(pi*self.nu)/pi)**self.N
        self.eps = eps
        self.carteps = ceps
        self.peps = 1
```

```python
25          sca = 2*pi/self.L
26          xpts = 201
27          place = linspace(0,10,xpts)
28          rpts = 5
29          ray = linspace(0,1,rpts)
30          raytime = kron(place,ray).reshape((xpts,rpts))
31          self.sum_rule = 0
32          self.Place = sca*1j*place[:,newaxis]
33          self.Time = -sca*sca*0.5j*raytime #
34          self.tau = zeros((xpts,rpts),dtype=complex)
35          self.get_box_filling()
36          print('box fill:',self.boxfil,'shuffle states:',self.complexity,end=', ')
37          if self.complexity < maxcomplex and self.complexity>0:
38              self.ground_shuffle()
39              self.build_block(0,0)
40          else:
41              print('This would take too long.')
42
43      def qn2box(self,q):
44          return 2*abs(q//self.B)-int(q<0)
45
46      def box2qn(self,boxj):
47          return (1-2*(boxj%2))*((boxj+1)//2)*self.B
48
49      def get_local_shuffle(self,fillin,s):
50          locshuf = []
51          for bf in ind_draw(s,s+self.B,fillin):
52              p,e = 0,0
53              for n in bf:
54                  p += n
55                  e += n*n
56              locshuf.append([bf,p,e])
57          return locshuf
58
59      def get_box_filling(self):
60          self.innerboxes = 2*max([abs(hu//self.B)+int(hu>=0) for hu in
61              [self.qn[0],self.qn[-1]]]+[(self.N+self.B-1)//(2*self.B)+1])
62          self.boxfil = [0 for _ in range(self.innerboxes)]
63          self.maxpad = (self.N//self.B)*5
64          for q in self.qn:
65              self.boxfil[self.qn2box(q)] += 1
66          self.complexity = prod([ncr(self.B,bb) for bb in self.boxfil])
67          self.boxemp = [self.B-bb for bb in self.boxfil]
68          s = 0
69          self.U = [[]]
70          for j in range(self.innerboxes):
71              self.U[0].append(self.get_local_shuffle(self.boxfil[j],s))
72              if s<0:
73                  s = -s
74              else:
75                  s = -s-self.B
```

```python
    def ground_shuffle(self):
        weightrix = zeros(shape=tuple([ncr(self.B,nnf) for nnf in self.boxfil]))
        walking = 0
        for ket,p,e,multind in self.in_box_shuffle(self.boxfil):
            w,w2 = self.update_tau(ket,p,e)
            weightrix[tuple(multind)] = w2
            walking += w
        odw = disect(weightrix)
        self.BT = [[None for boj in range(self.innerboxes)]]
        self.sortout(odw,0,0)
        self.p_h_profiles = [[[[0 for _ in range(self.innerboxes)],[],walking]]]]
        self.blocks = [[[]]]
        self.cart = []

    def stack_block(self,l,p,t):
        if len(self.blocks[l][p]) == t+1:
            self.blocks[l][p].append(self.deepen_gen(l,p,t+1))
            cm = None
            for rm,cm in self.blocks[l][p][t+1]:
                if cm:
                    self.cart.append([rm/cm,[l,p,t+1]])
                    break

    def harvest(self):
        l,p,t = self.cart[0][1]
        if len(self.cart)>1:
            thresh = self.cart[1][0]
        else:
            thresh=0
        for rm,cm in self.blocks[l][p][t]:
            if rm < thresh*cm:
                self.cart[0][0] = rm/cm
                return
        self.cart.pop(0)

    def build_block(self,l,p):
        if len(self.blocks[l]) == p+1 and p<self.maxpad:
            self.explore(l,p+1)
            print(l,p+1,end=' ... ')
        if len(self.blocks) == l+1 and l<self.maxorder and p >= self.peps*l:
            for p0 in range(self.maxpad+1):
                print(l+1,p0,end=' ... ')
                self.explore(l+1,p0)
                if self.cart and self.cart[-1][1]==[l+1,p0,0]:
                    return
            print('not enough tiers')

    def hilbert_search(self,wish,maxtime = 0):
        if not maxtime:
            self.clo = self.N*3 + clock()
```

```
127          else:
128              self.clo = maxtime + clock()
129          while self.sum_rule < wish and clock()<self.clo:
130              if self.cart:
131                  self.cart.sort(reverse=True)
132                  l,p,t = self.cart[0][1]
133                  lc = len(self.cart)
134                  self.build_block(l,p)
135                  self.stack_block(l,p,t)
136                  if len(self.cart)> lc:
137                      continue
138                  self.harvest()
139              else:
140                  l1 = len(self.blocks)
141                  for l in range(l1):
142                      pma = len(self.blocks[l])
143                      if pma <= self.maxpad:
144                          self.explore(l,pma)
145                          l = -1
146                          break
147                  if l1 <= self.maxorder:
148                      self.explore(l1,0)
149                  elif l>0:
150                      return
151          print('!')
```

## 7.3 Comparison Numerical to Analytic Results

We now show how successful the algorithm is at calculating the $\tau$-function. For thermal expectation values, we must average classically over system states drawn from a thermal ensemble, described in section 4.2. We have used the grand-canonical ensemble, choosing single-particle quantum numbers independently to construct multi-particle states. First, we pick a finite system length $L$. The Fermi-Dirac distribution (23) is then fitted with chemical potential $\mu$ such that the expected filling $N = L$, however, for finite realizations each may be off by some margin[31]. The resulting operator is averaged over states with different particle numbers. We highlight a different domain of the $\tau$-function than in other sections: as a function of position $x$, in the static case ($t = 0$) and at a speed $x/t = 1$. This features larger amplitudes which are slightly easier to handle numerically. We know by construction $\tau(0,0) = 1$. However, this is also the hardest point to correctly compute by summing basis states: all contributing phasors interfere constructively, and we would need to saturate the sum rule $s$ to recover this result. For this reason, around $x = 0$, the algorithm has the largest error, which is $1 - s$. Away from the origin, the omitted phasors are mostly the highly-oscillating ones, and they would cancel each other largely, meaning at most points of space-time, our error is much lower, as evinced by the figure below. We compare the results of the algorithm with the Fredholm determinant from (47) in figure 19. The parameters chosen are $T \in \{0.5, 1.5, 3\}$, shift $v = 0.4$, and $L \in \{15, 25\}$. Although small, these systems are hardly distinguishable from the TDL. Increasing particle number reduces accuracy computationally[32] but does not appear to augment

---

[31]For $L = 25$, the standard deviation is close to 1.6 particles.

[32]All plots shown here can be generated on a single CPU of a standard Intel Core i7 3GHz laptop computer in a matter of hours. The typical number of ket states considered varies wildly per bra state, depending on the entropy

the many-body character of this system. The agreement is especially striking in the static case, likely because the required phasors do not oscillate so rapidly ($\mathcal{O}(k^2 t)$), meaning less finely tuned summation is required to achieve high accuracy.

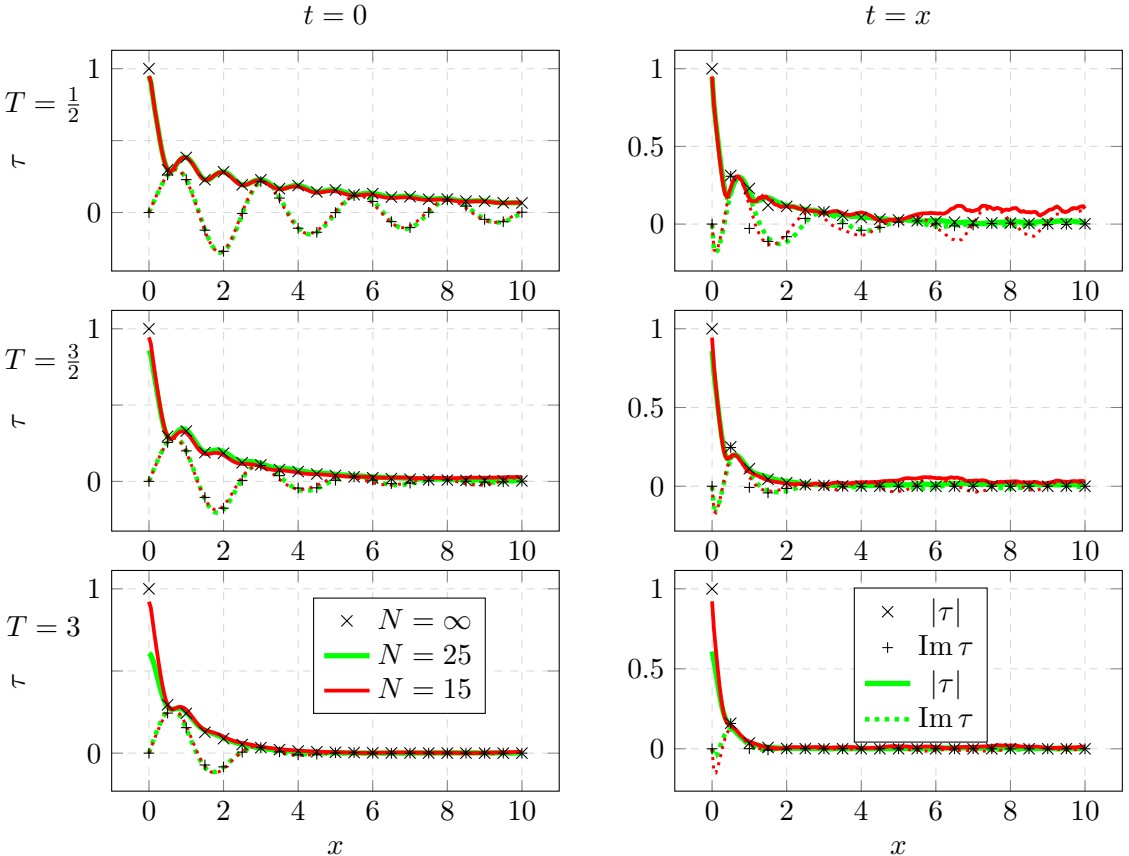

Figure 19: Numerical calculation of $\tau$ via the algorithm, plotted against position $x$. Green lines have $L = 25$ and chemical potential $\mu$ such that the expected filling $N = L$ in the grand-canonical ensemble, red the same with $L = 15$. For comparison, these are plotted alongside the exact result of the Fredholm in black marks. The solid lines (X's) are the envelope $|\tau|$, while the corresponding dotted lines (+'s) are the imaginary part, showing oscillations. From top to bottom, temperature increases through $T \in \{0.5, 1.5, 3\}$. The left column is the static $\tau$-function, right the ray $x/t = 1$. We see near-perfect agreement away from $x = 0$, where the unsaturated sum rule prevents us correctly interfering all the phasors. It is also clear that $L = 15$ already approaches the TDL. The box size used is $\beta = 4$, shift $\nu = 0.4$. Each numerical plot is a flat average over 50 sampled states. We expect adding samples would improve the erratic average at larger $x$.

## 7.4 Lieb Liniger Model

As a proof of concept, we have modified the code from section 7.2 to handle a system with an isomorphic Hilbert space: the repulsive Lieb-Liniger model [100]. This is an integrable gas of $\delta$-interacting bosons on the continuum. If each boson, denoted $a$, has coordinate $x_a$, the

---

of the latter. It could be anywhere from several hundred for the bra's at $T = \frac{1}{2}, L = 15$ to around a million, for the those at $T = 3, L = 25$.

Hamiltonian is printed

$$H \propto -\sum_{a=1}^{N} \frac{\partial^2}{\partial x_a^2} + 2c \sum_{a<b} \delta(x_a - x_b), \tag{162}$$

writing $c \geq 0$ for the interaction strength. The Tonks-Girardeau gas, mentioned in subsection 2, is the $c \to \infty$ limit of this system, and its particles are essentially free fermions again, connecting to the main theme of this paper.

The interaction energy means the particles are in general not free, however, eigenstates are still in bijection to the sets of disjunct integers $\{n_a\}$ for $N$ odd, and half-integers for $N$ even. In lieu of pure single particle momenta, the particles now have sets of *rapidities* $q_a$ that satisfy coupled *Bethe Equations*.

$$Lq_a = 2\pi n_a - 2 \sum_{b=1}^{N} \arctan\left(\frac{q_a - q_b}{c}\right). \tag{163}$$

Total momentum remains unchanged as $\sum_a q_a$ and energy is doubled by convention, $\sum_a q_a^2$. In the Lieb-Liniger model, there are determinant style operators of central interest. If we move to second quantization notation, in terms of a field operator $\Psi$, typical focal points are the field-field correlation function,

$$O_1(x,t) := \left\langle \Psi^\dagger(0,0)\Psi(x,t) \right\rangle \tag{164}$$

and the density-density correlation function

$$O_2(x,t) := \left\langle \Psi^\dagger(0,0)\Psi(0,0)\Psi^\dagger(x,t)\Psi(x,t) \right\rangle. \tag{165}$$

Form-factors are single entries of the matrices $\langle \mathbf{g}|\Psi|\mathbf{k}\rangle$ and $\langle \mathbf{g}|\Psi^\dagger\Psi|\mathbf{k}\rangle$. They can be expressed in terms of a dressed determinant of the rapidities of the energy eigenstates $\langle \mathbf{g}|$ and $|\mathbf{k}\rangle$. For the exact forms, see e.g. appendices C & D of [101]. With this knowledge, on $N$-particle state $\langle \mathbf{g}|$, these two-point functions 164 and 165 can be expanded in an internal basis of eigenstates $|\mathbf{k}\rangle\langle\mathbf{k}|$, where $\mathbf{k}$ is size $N$ for $O_2$ and size $N-1$ for $O_1$. The time and space dependence is then simply found by evolving using the translation operators, i.e. multiplying every term by a $\mathbf{k}$ dependent phase. The single state expectation is

$$\langle \mathbf{g}|O_1|\mathbf{g}\rangle = \sum_{\mathbf{k}} \left| \langle \mathbf{g}|\Psi^\dagger|\mathbf{k}\rangle \right|^2 e^{i\sum_a \left(x(g_a - q_a) - t(g_a^2 - q_a^2)\right)}, \tag{166}$$

and similar for $O_2$. Therefore, numerical calculation of these $O_1$ and $O_2$ is similar in complexity to that of the $\tau$-function. The largest difference is that the Bethe equations must be solved for every eigenstate. This is done with the Newton-Raphson method, which increases computational time. Conversely, empirically the weight $\left| \langle \mathbf{g}|\Psi^\dagger|\mathbf{k}\rangle \right|^2$ is more concentrated in a smaller number of eigenstates, as compared to the shifted fermion basis. In other words, the distribution is more skewed over $|\mathbf{k}\rangle$, reducing the number of states that must be visited to shrink the error to a satisfactory value. Hence for similar system sizes, at the analytically most challenging interaction strength of $c \approx 4$ [102], the Lieb-Liniger calculations are comparable in efficiency those of the free-fermionic $\tau$-function (1).

One computational speed-up is achieved by computing the Jacobian of the rapidities as a function of the original quantum numbers, and applying a linear correction to the original rapidities in order to approximate the rapidities of states with similar integers. Deriving (163) to $\partial n_b$, the Jacobian $\frac{\partial q_a}{\partial n_b}$ is found to satisfy

$$L\frac{\partial q_a}{\partial n_b} + 2\sum_{m=1}^{N} \left( \frac{c}{c^2 + (q_a - q_m)^2} \left( \frac{\partial q_a}{\partial n_b} - \frac{\partial q_m}{\partial n_b} \right) \right) - 2\pi\delta_{a,b} = 0, \tag{167}$$

which is also solved once, upon initialization, with the Newton-Raphson method. The inputs are the rapidities $\{g_a\}$ of the bra $\langle \mathbf{g}|$ state for the density operator $O_2$. Because the basis for expanding $O_1$ has one boson fewer, we excite from a state whose quantum numbers are obtained from those of $\mathbf{g}$ by removing the integer closest to the origin, and moving all positive integers down by $\frac{1}{2}$, and negative integers up by $\frac{1}{2}$. Accordingly, the solutions $\mathbf{k}$ of these inputs to the Bethe equations are used in (167) for the field-field correlator $O_1$.

When each local box shuffling is added recursively in **in_box_shuffle** (see A.4) and **shuffle_profile_tier** (see A.9), we can already linearly add the contributions of integers $\tilde{n}_b$, which are displaced from the original $n_b$, to the predicted rapidities $\tilde{q}_a$, summing over $b$ in bursts in

$$\tilde{q}_a \approx q_a + \sum_b \frac{\partial q_a}{\partial n_b}(\tilde{n}_b - n_b).$$

(168)

One extra step is necessary because of the choice of box index ordering in (154) and the excitations between boxes. In order to match $\tilde{n}_b$ to $n_b$, at the same $b$, we must know, for each box, how many occupied quantum numbers are left of that box. This is calculated beforehand from the box filling $\phi_a$ and the excitation profile, and passed as a list to the modified **in_box_shuffle** and **shuffle_profile_tier**.

Thermally distributed states for infinitely large systems are found by sampling single particle quantum numbers according to the solutions $\rho(k)$ of the *Thermodynamic Bethe Equations* [103], exactly as in the free fermion case. For finite systems, the probability of a quantum number being occupied depends on the other occupied numbers, and technically, they cannot be sampled independently. Nonetheless sampling that way is the norm is this field of research. Tabulating a large enough selection of many-body states, computing their energies, and sampling according to the Gibbs weight is alas computationally prohibitive. For a selection of the results[33] for $O_1(x,0)$ and $O_1(x,x)$, see figure 20. We have chosen the temperature to be twice the corresponding values of the comparable plots for the $\tau$-function, to compensate for the energy being doubled as well. We see again that for very entropic bra states, the algorithm struggles to find the full weight $s = 1$, visible around $x = 0$.

As for $O_2$, we cannot plot in the same way: it is not normal ordered, from which it follows that there is a Dirac-$\delta$-divergence at $(x,t) = (0,0)$. The sum rule is therefore by definition infinite. Instead, we show what is essentially the Fourier-transform of $O_2$: the *Dynamical Structure Factor* (DSF) [102]. We bin the weight per contributing state of the operator, according to its momentum and energy, on a heat map to obtain $O_2(p,E)$. Because the operator is always real, the weight (the Form Factor squared) is already positive, and all states accumulate constructively. In order to aid in discerning the weight differences, which span many orders of magnitude, we have in fact plotted the log of the weight density of $O_2(p,E)$. The units of the axis are in the Fermi momentum and energy: $k_F := q_N$ and $E_F := q_N^2$ for $|\mathbf{k}\rangle$ the $N$-particle ground state. See figure 21. These subfigures also offer an insight into which states the operator scopes out: in the white areas, no states have been considered. The algorithm was run until a 'sum rule' of $s = 12$ was attained, any positive number is possible: more and more states at the periphery will be added[34].

---

[33] these were obtained on a single thread of a laptop computer, with an Intel i7 3Ghz processor, in less than an day.

[34] Multiple runs were performed on the same computer in a matter of hours. The average number of ket states $\langle \# \rangle$ visited per $N = 15$ bra state were, for $T \in \{1, 3, 6\}$ were, respectively, 14K, 128K and 206K.

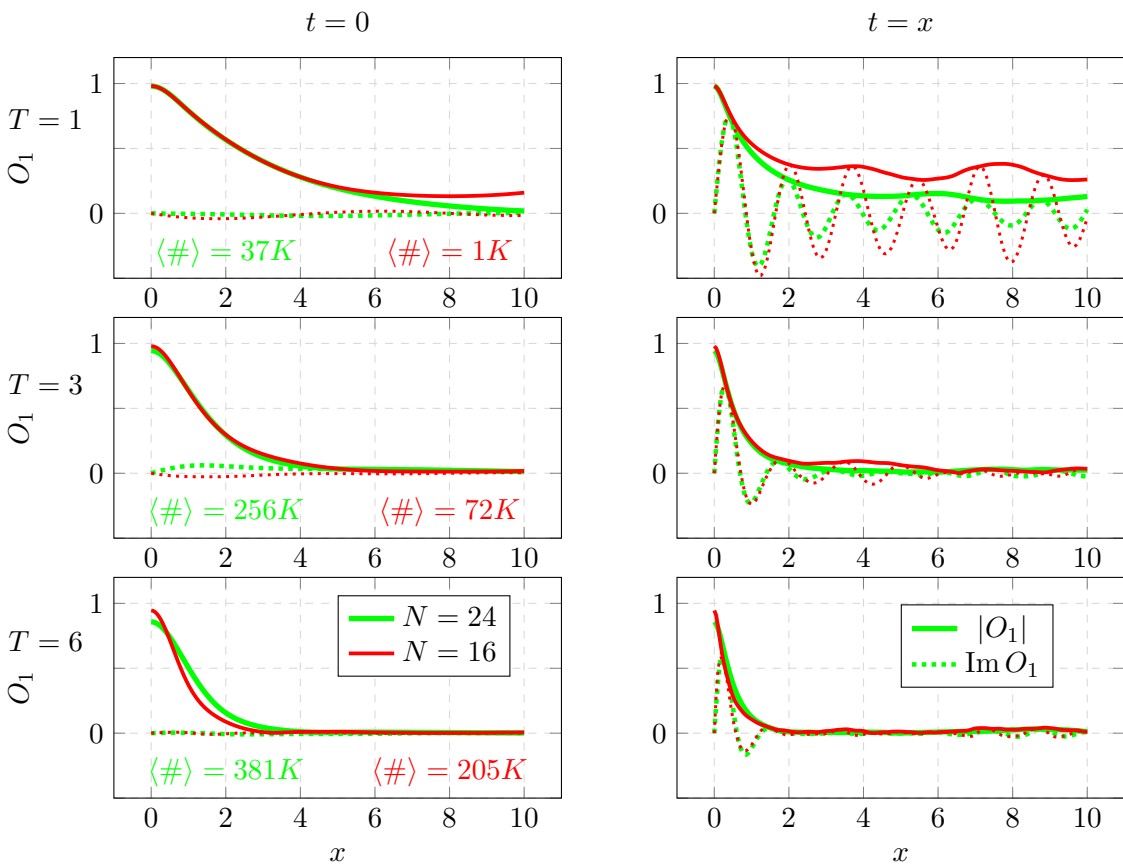

Figure 20: Numerical calculation of $O_1 := \langle \Psi^\dagger(0,0)\Psi(x,t) \rangle$ via the algorithm, plotted against position $x$. Green lines have $N = L = 24$, red $N = L = 16$. From top to bottom, temperature increases through $T \in \{1,3,6\}$. The left column is the static $\psi$-operator expectation value, right the ray $x/t = 1$. The box size used is $\beta = 4$, interaction parameter $c = 4$. Each numerical plot is a flat average over 50 sampled bra states. The average number of ket states $\langle \# \rangle$ visited per bra is printed on the left bottom.

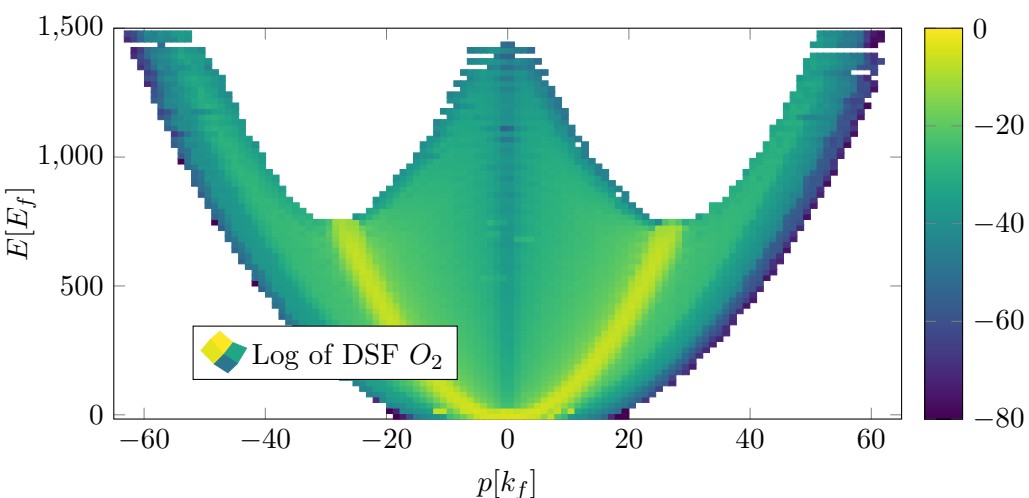

Figure 21: Numerical calculation of the log of the Density Structure Factor $O_2 := \langle \Psi^\dagger \Psi \Psi^\dagger \Psi \rangle$ via the algorithm, as a function of the momentum and energy of the contributing states, in units of $k_F$ and $E_F$. The color corresponds to the log of the weight density in each rectangular bin. $T = 6$, as temperature rises, the figure stretches to higher $E$ and $p$, but remains qualitatively similar. $N = L = 15$. The heat map shown is a flat average, bin-wise, over that of 50 sampled states, before taking the log.

# 8 Conclusion

Looking back, in this paper we have considered numerous ways to evaluate free-fermionic $\tau$-functions in finite-temperature states or ensembles. Analytically, we presented it in section 4 in terms of a determinant for finite sizes, and in the TDL a Fredholm determinant. We analyzed the latter's asymptotics at distant parts of space-time via an application of the effective form-factor method (Ref. [81]).

Additionally, we reviewed in section 5 how the same asymptotic can be obtained at zero temperature, using a completely orthogonal set of techniques. As a stepping stone, we have reproduced the original ground state orthogonality catastrophe: the power law vanishing of the form-factors with system size. Then we remedied this by a well-tuned soft mode summation, which is nothing other than the microscopic administering of bosonisation.

At $T = 0$, we have observed a novel scaling behavior, when the critical momentum is located on the Fermi surface. In this case we were not able to find the overall constant in the asymptotic as the combinatorial techniques of the soft mode summation are based on linearization of the spectrum. This is no longer applicable in the critical case, and neither are other bosonization methods.

Besides providing a manner in which to extract analytically the asymptotic of the Fredholm determinants, the soft mode summation at zero temperature also serves as an example of a partial summation of a form-factor series that is finite in the TDL. For finite temperature, we could not apply the same logic. The orthogonality catastrophe is much more severe in this case, as shown in section 3, and far more kinds of soft modes must be taken into account. To bypass this situation and still obtain some analytic results, in section 6 we constructed effective form-factors, in terms of which we only care about a single overlap, albeit of infinite-dimensional states. One interpretation of this, is that we reformulated the finite temperature finite (but thermodynamically large) system into a zero-tempererature (only involving the ground state), infinite system. This approach proved successful and taught us more about the asymptotics of the $\tau$-function.

Namely, the aforementioned scaling phenomenon also shows itself when we extract the asymptotic late time behavior by means of effective form-factors. We found sufficient demands to solve for the momentum dependent effective phase shift $\eta(k)$ of the fermions, valid as long as $k$ is sufficiently far from the critical point $x/t$. Across this domain, the imaginary part of the phase shifts in some as of yet unknown fashion, and the width of the domain over which it shifts scales as $O(1/\sqrt{t})$. At $t \to \infty$, the effective phase shift must exhibit a discontinuity. Initially this would seem problematic: the corresponding effective overlap has its own orthogonality catastrophe in this case, and there are no soft modes available to compensate. However, we believe the truth is more subtle. We proposed that this discontinuity is regularized by some unknown sigmoid function. Nevertheless, we have demonstrated that the sigmoid's influence only reaches to an overall constant factor in the functional form of the $\tau$-function approximation. We are able to make robust predictions about the scaling behavior of the $\tau$-function, which are supported by numerous figures and simulations showing very promising agreement, and free of any fitting parameters.

In particular, we have discovered that on top of the expected exponential decay there is an additional power law scaling, with an exponent dependent on the discontinuity. As is well known, gapless field theories at zero temperature become scale invariant and their observables behave as power laws. Away from the ground state, they have a characteristic length in an exponent. In this we find, just like in some similar models discussed in the introduction (e.g. [80, 82]), a hybrid scenario: both kinds of scaling in the same regime. Notably, in a parallel work [104], a generalization of a similar free-fermionic $\tau$-function was considered, where the additional power law was also confirmed in the time-like regime.

This feature is universal in the sense that it appears over all nonzero time scales, and with only very light demands on the distribution function $\rho(q)$. We believe this property warrants further investigation. There could be a strategy from microscopic first principles. One might assume a discontinuous shift defining the effective form-factors, designate a neighborhood of the critical point in phase space and compensate the resulting orthogonality catastrophe with a judicious summation of soft modes. The same approach should be applicable to finding the asymptotic at zero temperature, in the case when the critical point coincides with the Fermi momentum. Yet another possibility to tackle this problem is by using recent methods developed in [46].

If one would like to attempt a variation on the technique of effective form-factors, perhaps one may relax the demand that the kernel equal the quasi kernel, and instead only require their determinants to agree. This should allow for far more freedom in the shape of the effective form-factors, if one can manage to find an explicit solution to this demand.

Furthermore, another interesting question is how to adjust the calculations of the effective form-factors at the point of symmetry $\nu = 0.5$. In our case, the approximation breaks down and the error diverges.

Completely orthogonal to the techniques listed above, in section 7 we have created a tailor-made algorithm to sum over bases of Hilbert spaces with a free-fermionic structure, i.e. spanned by sets of disjunct integers. Sharing none of the assumptions of the other results, it simply improves upon a brute force summation by partitioning Hilbert space in an intelligent way, guided by the heuristic that states with locally similar sets of quantum numbers have larger mutual overlaps. With this perspective, states that yield high operator weight at early steps of the algorithm help it to identity other states with high probability of yielding more operator weight. Our algorithm complements existing software such as ABACUS [13] well, as the former is intended for highly entropic expectation values, and the latter for physics close to the ground state. Collecting all weight is an endless task, nonetheless the error can be efficiently minimized. For very modest system sizes we already cover most of the behavior we expect in the TDL. As a proof of principle, the algorithm was also applied to the repulsive Bose gas.

# Acknowledgements

We are grateful to Jean-Sébastien Caux for numerous fruitful discussions and useful suggestions. The authors thank the European Research Council, the National Research Foundation of Ukraine and the Polish National Agency for Academic Exchange for enabling this research.

**Funding information** The authors gratefully acknowledge the support of European Research Council under ERC Advanced grant No 743032. OG acknowledges support by the National Research Foundation of Ukraine grant 2020.02/0296 and from the Polish National Agency for Academic Exchange (NAWA) through the Grant No. PPN/ULM/2020/1/00247.

# A  Subroutines

In this appendix, we briefly explain some of the routines used repeatedly in the overarching algorithm. Because it is a recurring theme, it is worthwhile to comment on how all elements of an arbitrarily large tensor product space, such as (155), are produced algorithmically, without using excessive memory. The application is any kind of permutation or combinatoric set, see for example A.4, A.9, and A.3. In this reseach we have made eager use of recursion by *generator*

objects. In python, one can create an object of the generator class. This object is a function that will sit in memory, with its local variable space saved, without running to completion. Every time it is prompted or called, for instance to produce a local shuffling, it runs its code until it yields another output, then halt again. This is preferable, when many combinatoric values must be found, to storing all such results in a list, as each is often only needed once and only one at a time. When the code of the generator terminates, the object is cleared from memory. Once can loop over generators in a similar fashion as over lists. One difference: though the number of outputs produced is deterministic, it may not be known a priori. Such generators may call themselves with a smaller input, tensoring a simple space to a larger space with an already tensored structure. This process is evocative of mathematical induction.

## A.1 Binomial Coefficient

| Name: | **ncr** |
|---|---|
| Purpose: | Calculate the binomial coefficient. |
| Inputs: | Choose **r** from **n** distinguishable choices. |
| Outputs: | the amount of ways this can be done, $\binom{n}{r}$. |

This is an efficient code for binomial coefficients on integers, using falling factorials.

```python
def ncr(n,r):
    if r < 0:
        return 0
    p,q = 1,1
    for j in range(r):
        p *= n-j
        q *= j+1
    return p//q
```

## A.2 Local Box Shufflings

| Name: | **ind_draw** |
|---|---|
| Purpose: | Provide all local box shufflings for a box allowing the integers $[o, o+1, \ldots, m-1]$, filled with **n** quantum numbers. |
| Inputs: | Lower bounding integer $\geq$ **o**, upper bounding integer $<$ **m**, sample size **n**. |
| Outputs: | Each call to an instance of **ind_draw** yields a unique list of **n** integers between from $[o, m-1]$. Terminates after $\binom{m-o}{n}$ outputs. |

**ind_draw** is a generator that calls itself recursively. All shufflings at size **n** can be obtained by choosing the first element $k$ from $[o, m-n]$, and joining it with the all results of a box shuffling from a smaller box: using $k+1$ as lower bounding integer, and $n-1$ filling. We only take values of $k$ that allow for a legal filling in lower recursion. In order for recursion to terminate, at **n** $= 0$, the trivial empty list must be returned. As a fail safe, the algorithm only yields any results for nonnegative **n**, otherwise recursion would run away to **n** $\to -\infty$.

```python
def ind_draw(o,m,n):
    if n>=0:
        if n == 0:
            yield []
        else:
```

```
6        for k in range(o,m-n+1):
7            for sub in ind_draw(k+1,m,n-1):
8                yield [k]+sub
```

## A.3 Constrained particle placement

| Name: | **boxicles** |
|---|---|
| Purpose: | Provide all ways to place **n** particles divided over a number of positions, where each position has a maximum allowed occupation. |
| Inputs: | Number of particles **n**, and a list of nonnegative integers of allowed occupation per position, **depths**. |
| Outputs: | Each call to an instance of **boxicles** yields a unique list of the same length as the input **depths**, with nonnegative inputs summing to **n**, and never is the $k^{\text{th}}$ element of such an output larger than the $k^{\text{th}}$ element of **depths**. All possibilities are yielded, filling the positions from left to right, then the routine terminates. |

  **boxicles** is a generator that calls itself recursively. All placement profiles with **n** particles can be obtained by adding a single particle to a placement profile of **n** − 1 particles. In order to avoid duplicates, we never add to lower level profiles at a position that is to the right of any pre-placed particle. This means we loop over the position $k$ of the profile from left to right, placing an additional particle there if allowed. When we encounter a non-empty position, we may add one particle, then break out of loop. In this manner we achieve a definite ordering of the placement of particles, meaning a profile cannot be produced in multiple ways.

  At the lowest level of recursion, **n** = 0, we must terminate by yielding a list of zeros.

```
1   def boxicles(n,depths):
2       M = len(depths)
3       if n == 0:
4           yield [0 for _ in range(M)]
5       else:
6           for preput in boxicles(n-1,depths):
7               for k in range(M):
8                   if preput[k]<depths[k]:
9                       yield [preput[a]+int(a==k) for a in range(M)]
10                  if preput[k]:
11                      break
```

## A.4 All Global Microshufflings at Original Box Filling

| Name: | **FFstate.in_box_shuffle** |
|---|---|
| Purpose: | Provide all global microshufflings corresponding to a certain list of box fillings. |
| Inputs: | A list **Bfil** of nonnegative integers describing the occupation of each box. |
| Outputs: | Each call to an instance of **in_box_shuffle** yields 4 variables: <br><br> 1. A unique microshuffling, described by a list of integers. This is chosen such that an amount of particles are placed in the $a^{\text{th}}$ box equal to the $a^{\text{th}}$ entry of **Bfil** <br><br> 2. the scaled momentum of this shuffling (sum of all integers) <br><br> 3. the scaled energy of this shuffling (sum of all squared integers) <br><br> 4. A list, equal in length to **Bfil**, of the indices $\{j\}$, corresponding to an enumeration of the local shuffling of each box. <br><br> After all microshufflings are produced, the routine terminates. |

**in_box_shuffle** is both a method of the **FFstate** class as well as a generator that calls itself recursively. The recursive nature is clear: if we can generate all microshufflings for some number $l$ of boxes, extending each of those with each local box shuffling of an additional box in turn, constitutes all microshufflings of $l+1$ boxes. At the lowest level of recursion, for $l=0$ boxes, we must return the empty list to halt. At initialization of the **FFstate** class instance, the scaled momenta and energy of the bra state are stored as **self.pbra** and **self.ebra**. In (1) we see that these quantities are taken relative between bra and ket, so at $l=0$, we begin counting momentum and energy from there, and these are also built up recursively.

Another variable calculated at initialization is **self.U**. It is a deeply nested list of lists. The first index $l$ denotes the excitation, we may have $l < 0$ for holes, as Python indexing wraps around when negative. With first index 0 in **self.U**, this refers to boxes with original filling. The second index enumerates the boxes in the staggered fashion described earlier in (154). The contents of **self.U**$[l][a]$ is a list of tuples, each tuple containing 3 elements: a local shufflings of box $a$, with filling $\phi_a + l$, along with scaled momentum and scaled energy $p, e$ of that shuffling. These were all found with **ind_draw**, described in A.2.

All possible local shufflings are stored. By combining recursively all such local box shufflings, all global microshufflings are produced. In order to know at what multi-index of the **weightrix** to put this state, the local shufflings are enumerated with local index $j_a$, $a$ being the box index, and appended to the indices of the lower index boxes to form a unique microshuffling multi-index.

```python
def in_box_shuffle(self,Bfil):
    a = len(Bfil)
    if a == 0:
        yield [],self.pbra,self.ebra,[]
    else:
        for leftside,p0,e0,indis in self.in_box_shuffle(Bfil[:a-1]):
            j=0
            for nowbox,p,e in self.U[0][a-1]:
                yield leftside + nowbox, p0+p,e0+e,indis+[j]
                j += 1
```

### A.5 Integrate Out All But One Tensor Dimension

| Name: | **disect** |
|---|---|
| Purpose: | Take a multidimensional tensor (Numpy Array), and return a list, for each dimension, of that axis with all other dimensions integrated out. |
| Inputs: | Multidimensional array of floating point numbers, **We**. |
| Outputs: | List of one-dimensional Numpy arrays **w**, whose first index enumerates the dimensions/ axes of **We**, and whose second index corresponds to the position along that axis in **We**. The value of $\mathbf{w}[a][j] = w_{j_a}(a)$ from (160). |

    **disect** takes as argument what is termed the **weightrix** in the main algorithm, at the stage of producing all microshufflings at original box fillings. It returns the proxy weights for each box/dimension, with all other boxes/dimensions integrated out. If a box $a$ only has the trivial filling (**As**$[a] = 1$), there is no need to calculate the normalized proxy weight of the corresponding axis, as it is by definition unit. If, conversely, box $a$ is nontrivial, containing multiple distinct local shufflings, we must sum over all other axes. Then we modify **We**, by summing over the first axis, collapsing the array to a small dimension, as this axis has been extracted and subsequent axes need this axis summed.

```
1   def disect(We):
2       w = []
3       As = We.shape
4       l = len(As)
5       for a in range(l):
6           if As[a] >1:
7               w.append(We.sum(tuple(range(1,l-a))))
8           else:
9               w.append(arr([1]))
10          We = We.sum(0)
11      for a in range(l):
12          if As[a] >1:
13              w[a] /= We
14      return w
```

### A.6 Divide Local Shufflings into Tiers

| Name: | **FFstate.sortout** |
|---|---|
| Purpose: | Divide all local shufflings at a given level and pad into tiers, according to proxy weights. |
| Inputs: | list of Numpy arrays **odw**, each array corresponding to a box, the arrays holding proxy weights. Level **nl** determines where to store the result, pad **pad** determines where to store the result, as well as which boxes are involved. |
| Outputs: | None. The effect is an updated **self.BT** variable. |

    **sortout** is a method of the **FFstate** class that updates the **FFstate** variable **self.BT**. The latter is a deeply nested list of lists that holds the box tiers of the current bra state instance. The first index denotes the excitation: **nl** may be zero, for the original filling of the box, or a positive or negative integer, denoting that many particles or holes in the box, respectively. **self.BT** should only be called with those indices after sufficient exploration. The second index of **self.BT** is the box index. If we are at **pad** $= 0$, this routine treats all boxes $a$ originally

filled in the bra state. If **pad** $> 0$, the routine only treats the outer two values for $a$. The third index of **self.BT** is the tier, so for this index 0, we find a list of tuples, copied over from **self.U**, corresponding to local box shufflings that have the expected highest weight for this given box and this excitation on that box. Higher tiers $> 0$ may or may not exist, and are populated by local shufflings at least a factor **self.eps** smaller in proxy weight than the previous tier. Proxy weights **odw** have be calculated as in (160) if (**pad** $= 0$, **nl** $= 0$), and else by (161).

```
def sortout(self,odw,nl,pad):
    if pad:
        a = self.innerboxes+2*(pad-1)
    else:
        a = 0
    for w in odw:
        argord = (-w).argsort()
        noweps = 1
        bt = []
        for j in argord:
            if not (w[j] > noweps):
                noweps *= self.eps
                bt.append([])
            bt[-1].append(self.U[nl][a][j])
        self.BT[nl][a] = bt
        a += 1
```

## A.7   Generate All States in a Block

| Name: | **FFstate.deepen_gen** |
|---|---|
| Purpose: | Object left in memory, can be prompted to produce volleys, small subsets of states from the block defined by the inputs, and update the operator $\tau$ and the sum rule with them. These states are disjunct from those produced in the **explore** (A.11) phase. |
| Inputs: | **level**, **pad**, and **tier** of the block under consideration |
| Outputs: | The total weight **running** found in each volley, and the number of states **count** in the volley. |

**deepen_gen** is a method of the **FFstate** class that, when called for the next yield, considers a volley of all states corresponding to a single excitation profile from the block, with all tier placements. During the explore phase, the states generated in any block have zero tier on non-extremal boxes, and all possible local shufflings, thus all possible tiers, are used on the extremal boxes. In order to have different states at this phase of the search, at least one local tier must be above zero on a non-extremal box. The variable **PES** is a list, in turn stored listed in the **FFstate** variable **p_h_profiles**, with the following content. The first entry is an excitation profile in the style of (156). The second is a list, length between zero and two, of the indices of extremal boxes. The third is the **walking** variable, a total amount of overlap found at this profile, including phase. This is useful for other investigations into the nature of field theory macrostates. Given the profile **PES[0]** and extremal boxes **PES[1]**, we can produce all possible placements of the local tiers to achieve **tier** global tier: the method **tier_profile**, (see A.8), distributes the global tier in all ways over the available boxes in the profile such that the maximum local tier is never exceeded, and that not all nonzero local tiers are found in the extremal boxes. With each tier profile **tierpf**, we may produce all possible states **ket** and

corresponding momentum and energy **p**, **e**, with the method **shuffle_profile_tier**, from A.9. Inside this subroutine, we call the **FFstate** method **update_tau**, seen in A.10, to update the $\tau$-function and the sum rule. After the volley is done, the **walking** variable is incremented, and the **running** and **count** variables are returned as a criterion for the success of this volley. The more weight returned in the fewer states, the better.

```python
def deepen_gen(self,level,pad,tier):
    for PES in self.p_h_profiles[level][pad]:
        walking = 0
        running = 0
        count = 0
        for tierpf in self.deepen_tier_profile(PES[0],PES[1],tier):
            for ket,p,e in self.shuffle_profile_tier(PES[0],tierpf):
                count += 1
                w,w2 = self.update_tau(ket,p,e)
                running += w2
                walking += w
        PES[2] += walking
        yield running,count
```

## A.8 Distribute Local Tiers over Boxes

| Name: | **FFstate.tier_profile** |
|---|---|
| Purpose: | Provide all distributions of local tiers over boxes that sum to a given global tier. |
| Inputs: | The excitation **profile** in the format of equation (156), a list of extremal box indices **EJ**, and the global **tier** needed on this profile. |
| Outputs: | Each call to an instance of **tier_profile** returns a list of integers, **tierpf**, each entry of which corresponds to a box in **profile**, and each value of which is the local tier of local shufflings to be concatenated to global states by the function that calls **tier_profile**. |

**tier_profile** is both a method of the **FFstate** class as well as a generator. Its first action is to tabulate in **deps**, from **self.BT**, how many local tiers are available for each box at present filling in the input **profile**. Then rather than calling **boxicles** (see A.3) directly, we must ensure that not all tiers end up in extremal boxes. If there is a single extremal box, or one entry to **EJ**, we can achieve this by artificially lowering (if necessary) the allowed local tier in that box to below the global tier, then naturally the remainder spills over into other boxes. If instead, there are two extremal boxes and two entries to **EJ**, we must distribute the local tiers manually. We determine the total capacity of the extremal boxes, and iterate over placing between none all all but one of the local tiers on them together, and inside one such iteration, divide the local tiers between the two extremal boxes in all ways. Finally, if there are no extremal boxes, straightforward calling of **boxicles**, suffices. We remember that at least one local tier must be placed, if we are to have different states than the exploration phase. Nomenclature is such that the minimal value of **tier** is zero, so **boxicles** is called with **tier** + 1 excitations.

```python
def tier_profile(self,profile,EJ,tier):
    deps = [len(self.BT[profile[j]][j])-1 for j in range(len(profile))]
    if len(EJ)==1:
        deps[EJ[0]] = min(deps[EJ[0]],tier)
```

```
5          if len(EJ)==2:
6              a,b = deps[EJ[0]],deps[EJ[1]]
7              deps[EJ[0]] = min(a+b,tier)
8              deps[EJ[1]] = 0
9              for tierpf in boxicles(tier+1,deps):
10                 if tierpf[EJ[0]] > a:
11                     tierpf[EJ[1]] = tierpf[EJ[0]] - a
12                     tierpf[EJ[0]] = a
13                 yield tierpf
14                 while tierpf[EJ[1]] < b and tierpf[EJ[0]] > 0:
15                     tierpf[EJ[1]] += 1
16                     tierpf[EJ[0]] -= 1
17                     yield  tierpf
18         else:
19             for tierpf in boxicles(tier+1,deps):
20                 yield tierpf
```

## A.9 All Microshufflings at an Excitation Profile and Tier Distribution

| Name: | **FFstate.shuffle_profile_tier** |
|---|---|
| Purpose: | Provide all microshufflings at a given excitation profile and tier profile. |
| Inputs: | Two lists: the excitation profile **phpf** in the format of equation (156), and a tier profile **tierpf**, consisting of nonnegative integers. |
| Outputs: | Each call to an instance of **shuffle_profile_tier** returns a tuple of three terms: a microshuffling **tail + locshuf**, the microshuffling's scaled momentum $\mathbf{p} + \mathbf{p0}$, and the microshuffling's scaled energy $\mathbf{e} + \mathbf{e0}$. |

**shuffle_profile_tier** is both a method of the **FFstate** class as well as a generator that calls itself recursively. It builds the microshufflings from the left. At the deepest level of recursion, both profiles are length 0, and we return the empty shuffling, and the gauge choice of momentum and energy **self.pbra** and **self.ebra**. On top of this, for each possible left part of the microshuffling, **tail**, found by calling **shuffle_profile_tier** with curtailed profiles, we return sequentially all local shufflings from **self.BT** at the correct particle-hole-excitation **phpf**$[-1]$ and the correct tier **tierpf**$[-1]$ for the next box to the right, along with that box's momentum and energy. The outermost layer of this function produces all full length microshufflings.

```
1      def shuffle_profile_tier(self,phpf,tierpf):
2          l = len(phpf)
3          if l==0:
4              yield [],self.pbra,self.ebra
5          else:
6              for tail,p,e in self.shuffle_profile_tier(phpf[:-1],tierpf[:-1]):
7                  for locshuf,p0,e0 in self.BT[phpf[-1]][l-1][tierpf[-1]]:
8                      yield tail+locshuf,p+p0,e+e0
```

## A.10 Update $\tau$-function

| Name: | **FFstate.update_tau** |
|---|---|
| Purpose: | Take a set of momentum quantum numbers, calculate the overlap to the bra state $\langle \mathbf{g} |$ used in the initialization, update the progress of the $\tau$-function and sumrule |
| Inputs: | Quantum numbers **ket**, and corresponding sum, or total scaled momentum **p**, and sum of squares of total scaled energy, **e**. |
| Outputs: | The overlap $\langle g | k \rangle$ and its squared modulus, the weight **w2** of the state. |

Also updates **FFstate** variables **self.tau**, **self.sum_rule** and **self.states_done**, which should be self-explanatory after section 7.1.

```
def update_tau(self,ket,p,e):
    detti = det(1/(self.abra-sorted(ket)-self.nu))
    w = detti*self.presin
    w2 = w*w
    self.tau += w2*exp(p*self.Place+e*self.Time)
    self.sum_rule += w2
    self.states_done +=1
    return w,w2
```

## A.11 Prepare Hilbert Space Section at New Pad and Level

| Name: | **FFstate.explore** |
|---|---|
| Purpose: | Create excitation profiles and local tiers needed to produce all states at a given pad and level, and produce the first of these states in the process. |
| Inputs: | The **level** and **pad** to be treated. |
| Outputs: | None. The effect is an updated **self.BT** variable with local tiers, updated **self.p_h_profiles** variable with excitation profiles, an updated **self.blocks** and **self.cart** variable reflecting the new available blocks to explore. |

**explore** is a method of the **FFstate** class that performs an important function in the Hilbert space search, by preparing all objects needed to produce all states in the blocks over a given pad-level combination. If the level is zero, there are no excitations and the pads can have no particles. There are no new states here. The script only needs to expand the size of **FFstate** objects **self.BT** and **self.blocks** to allow the algorithm to call these objects at larger pad indices, although they return empty lists as fillings. However, if the level is nonzero, new states must be produced. The algorithm prepares some auxiliary variables, such as the number of boxes to be considered **b0** and variables of the **FFstate** class **self.spind** and **self.spox**. These variables are updated by the closely related **FFstate** method **parasite**, described in A.12. **self.spind** has two entries, the first for particles and the second for holes, and each keeps track of the index of the enumerated local shufflings of an extremal box, if one is present. This means, as we build up global microshufflings recursively, at the depth where we iterate over an extremal box, **self.spind**[0] holds the integer counting which local shuffling is currently being used in the iteration, for the extremal box with $\lambda$ particles in it, if $\lambda$ is the level. **self.spind**[1] does the same for $\lambda$ holes. **self.spox** on the other hand holds the box index of the extremal box, again the first entry for particles, the second for holes. These data are useful in order to put the proxy weight in the correct location in the **weightrix** that is filled during the course of **explore**. **self.spox** and **self.spind** are updated by the method **parasite**, discussed later on.

The next division is whether the pad is nonzero. If yes, then the only extremal boxes can be the outer two, and only with particles, as there can be no holes in empty boxes. Moreover, if the level is above the box size $\beta$, or **self.B**, then there are no extremal boxes because the largest box excitations have already been treated at level $\lambda = \beta$. So for nonzero pad, $\lambda \leq \beta$, we create a **weightrix** with two indices for the two outer boxes, and in each of those, $\binom{\beta}{\lambda}$ zeros to hold the proxy weight, one for each local shuffling. Then for the boxes introduced in the new pad, we must add new entries to variables holding local shufflings. The first is the **FFstate** variable **self.U**, whose first index indicates the level, and whose second index the box number, and which must now be extended in this second dimension. The content of an entry of **self.U** at a given level and box is a list of tuples. Such a tuple contains a local shuffling, its scaled momentum and scaled energy. They are created by the method **get_local_shuffle**, which essentially applies **ind_draw** (see A.2) and sums its piecewise first and second power. The next variable to be updated is **self.BT**, which is more subtle and introduces our companion function **parasite**. **self.BT** holds the local shufflings, organized per signed level: i.e. $[\lambda]$ for $\lambda$ particles, $[-\lambda]$ for $\lambda$ holes in the box as the first index, box index as **self.BT**'s second index and tier as the third index. This variable is queried by combinatoric methods such as **shuffle_profile_tier** (see A.9). However, at present time, the tiers are not known for the extremal boxes. The solution is to have the extremal boxes, when they arise in the excitation profiles, behave as if they only have the lowest tier, zero, and produce all states with global tier zero. This includes all possible local shufflings in extremal boxes, and only the actual lowest local tier local shufflings in other boxes. During, we wish to keep track of which local shuffling is used in the extremal boxes, in order to add the weights of these states to the entry of the **weightrix** that corresponds to said shuffling. Instead of a simple list, we install a tailor-made generator, **parasite**, at the appropriate location in **self.BT**. In python, entering negative indices loops around to the end of the list: for a list **my_list** of length **l**, **my_list[l-a] = my_list[-a]**. For this reason, the structure of the first index of **self.BT** is {no excitation, 1 particle, 2 particles, ..., 2 holes, 1 hole}. As we wish to query **self.BT** for larger numbers of holes and particles, we must take care to insert the higher excitations in the middle of the first index, the outer list.

The following loops keep track of how many particles are in the outer pads, at least one and at most the level, or $2\beta$, then how these particles are divided between left and right, and then how the rest of the particles and holes distribute among the remaining boxes, by means of **boxicles** (A.3). This produces all possible excitation profiles at this level and pad, which are stored, and per profile states are produced as described above and their weights entered into the weightrix. The **FFstate** variable **p_h_profiles** is extended in order of decreasing state-averaged weight found in the profiles, the necessary tiers are created with **sortout** (A.6), and **self.blocks** is updated with the generator method **deepen_gen**, from A.7.

Conversely, if the pad is zero, the **weightrix** has an extra first index to allow for extremal boxes with holes, and we do not need to artificially place any number of particles in the outer boxes. Besides that, much is the same.

A final check is that the created block has any actual states at the lowest untreated global tier, which is tier one, at **self.blocks[level][pad][0]**. If so, the block is added to the **self.cart** and may be prompted for more states in the future. One final caveat: empirically it is desirable to suppress newly minted blocks from being searched directly. When they become the current searched block, even more are explored, which is computationally expensive. For this reason, the results of **explore** are initially added to the cart with an average weight that is scaled by a constant, called **self.carteps**. Testing has indicated that the most efficient search is performed with a constant **self.carteps** $= \frac{1}{5}$.

```
1    def explore(self,level,pad):
2        if level:
```

```
3                  b0 = self.innerboxes+2*pad
4                  flat = [0 for _ in range(b0)]
5                  self.spind = [None,None]
6                  self.spox = [None,None]
7                  if len(self.blocks[level-1]) <= pad:
8                      self.explore(level-1,pad)
9                  if pad:
10                     if level <= self.B:
11                         partconf = ncr(self.B,level)
12                         weightrix = [[0 for _ in range(partconf)] for _ in [0,0]]
13                         for boxj in range(b0-2,b0):
14                             self.U[level].append(
15                             self.get_local_shuffle(level,self.box2qn(boxj)))
16                             self.BT[level].append([self.parasite(level,boxj)])
17                     profnweight = []
18                     for oparts in range(1,min(level,2*self.B)+1):
19                         for opl in range(max(0,oparts-self.B),min(oparts,self.B)+1):
20                             opr = oparts-opl
21                             for partplace in boxicles(
22                             level-oparts,self.boxemp+[self.B for _ in range(2*(pad-1))]):
23                                 holesdeps = []
24                                 for j in range(self.innerboxes):
25                                     if partplace[j]:
26                                         holesdeps.append(0)
27                                     else:
28                                         holesdeps.append(self.boxfil[j])
29                                 for holeplace in boxicles(level,holesdeps):
30                                     profile = partplace+[opl,opr]
31                                     running = 0
32                                     walking = 0
33                                     count = 0
34                                     for jj in range(self.innerboxes):
35                                         profile[jj] -= holeplace[jj]
36                                     for ket,p,e in self.shuffle_profile_tier(profile,flat):
37                                         w,w2 = self.update_tau(ket,p,e)
38                                         running += w2
39                                         walking += w
40                                         count += 1
41                                         if self.spox[0] != None:
42                                             weightrix[self.spox[0]%2][self.spind[0]] += w2
43                                     if self.spox[0] == None:
44                                         extholes = []
45                                     else:
46                                         extholes = [self.spox[0]]
47                                         self.spox[0]=None
48                                     profnweight.append(
49                                     [running/max(1,count),[profile,extholes,walking]])
50                     self.p_h_profiles[level].append(
51                     [xp for _,xp in sorted(profnweight,reverse=True)])
52                     if level <= self.B:
53                         odw = []
54                         for jw in [0,1]:
55                             sw = sum(weightrix[jw])
56                             odw.append(arr(weightrix[jw])/sw)
57                         self.sortout(odw,level,pad)
58                     self.blocks[level].append([self.deepen_gen(level,pad,0)])
59                 else:
60                     profnweight = []
61                     if level <= self.B:
```

```
62                        weightrix = []
63                        for nl in [-level,level]:
64                            weightrix.append([[0 for _ in range(ncr(self.B,bb-nl))]
65                                for bb in self.boxfil])
66                            nou = []
67                            self.BT.insert(level,[[self.parasite(nl,boj)]
68                                for boj in range(self.innerboxes)])
69                            s=0
70                            for boxj in range(self.innerboxes):
71                                nou.append(self.get_local_shuffle(self.boxfil[boxj]+nl,s))
72                                if s<0:
73                                    s = -s
74                                else:
75                                    s = -s-self.B
76                            self.U.insert(level,nou)
77                    for partplace in boxicles(level,self.boxemp):
78                        holesdeps = []
79                        for j in range(self.innerboxes):
80                            if partplace[j]:
81                                holesdeps.append(0)
82                            else:
83                                holesdeps.append(self.boxfil[j])
84                        for holeplace in boxicles(level,holesdeps):
85                            profile = [partplace[jj]-holeplace[jj]
86                                for jj in range(self.innerboxes)]
87                            running = 0
88                            walking = 0
89                            count = 0
90                            for ket,p,e in self.shuffle_profile_tier(profile,flat):
91                                w,w2 = self.update_tau(ket,p,e)
92                                running += w2
93                                walking += w
94                                count += 1
95                                for hw in [0,1]:
96                                    if self.spox[hw] != None:
97                                        weightrix[hw][self.spox[hw]][self.spind[hw]] += w2
98                            extholes = []
99                            for sw in [0,1]:
100                                if self.spox[sw]!= None:
101                                    extholes.append(self.spox[sw])
102                                    self.spox[sw] = None
103                            profnweight.append(
104                                [running/max(1,count),[profile,extholes,walking]])
105                    self.p_h_profiles.append(
106                    [[xp for _,xp in sorted(profnweight,reverse=True)]])
107                    if level <= self.B:
108                        for jw in [0,1]:
109                            odw = []
110                            for jww in range(self.innerboxes):
111                                sw = sum(weightrix[jw][jww])
112                                odw.append(arr(weightrix[jw][jww])/sw)
113                            self.sortout(odw,[level,-level][jw],0)
114                    self.blocks.append([[self.deepen_gen(level,0,0)]])
115              for rm,cm in self.blocks[level][pad][0]:
116                  if cm:
117                      self.cart.append([rm/cm*self.carteps,[level,pad,0]])
118                      break
119          else:
120              self.BT[0].extend([[[[[],0,0]]],[[[[],0,0]]]])
```

```
121            self.blocks[0].append([[]])
```

## A.12 Placeholder Tier Zero

| Name: | **FFstate.parasite** |
|---|---|
| Purpose: | Employed by **explore**, from A.11. Take the place of the tier-zero list of local shufflings of an extremal box, inside **self.BT**, and return all local shufflings while updating auxiliary variables used to fill the **weightrix**. |
| Inputs: | The signed level **nl**, positive for particles and negative for holes, and the box index at which it is located, **boxj**. |
| Outputs: | Each call to an instance of **parasite** yields a local box filling as stored in **self.U**. |

An instance of **parasite** is created, and located at **self.BT[nl][boxj][0]**, by **explore**, while the level is **nl**, and **boxj** is the index a possible extremal box. While **explore** is using **shuffle_profile_tier** from A.9 to concatenate microfillings recursively, from time to time it needs the tiers of the unexplored extremal boxes. In that case, instead, **parasite** provides all possible fillings, while storing in the **FFstate** variables **self.spox** the box number of the extremal box, and in **self.spind**, the index of the local shuffling returned. For either particles (**ind** $= 0$) or holes (**ind** $= 1$), there is at most a single extremal box at a time. When all microshufflings of an excitation profile have been yielded, which share the same **self.spox**, both these auxiliary variables are reset. This setup is meant to mimic exactly the behavior of the boxes whose tiers are known at other locations in **self.BT**, from the point of view of **shuffle_profile_tier**, while allowing **explore** to fill the weights calculated into the correct locations in the **weightrix** for all local shufflings. As each box is iterated over a variable amount of times, **parasite** restores a new instance of itself at the same location **self.BT[nl][boxj][0]** before stopping the iteration and terminating.

```
1        def parasite(self,nl,boxj):
2            ind = int(nl<0)
3            self.spox[ind]=boxj
4            for j,locfil in enumerate(self.U[nl][boxj]):
5                self.spind[ind]=j
6                yield locfil
7            self.BT[nl][boxj][0] = self.parasite(nl,boxj)
```

# B  Orthogonality Catastrophe for a Discontinuous Phase Shift

In this appendix we discuss the scaling of the overlap $\mathcal{F}$, defined in (113) of the effective form-factors in the quasi-$\tau$-function, defined in (104). We consider the case for a discontinuous $\eta(k)$, but specification when the discontinuity goes to zero is straightforward, reproducing the results for sections 5 and 6. Let $M$ be the largest quantum number that corresponds to the effective Fermi sea. In other words it denotes the boundary of the set of consecutive integers $[-M, -M + 1, \ldots, M]$ from which to select indices such as $a$, $b$ and $m$. These indices are eventually meant to iterate over $\mathbb{Z}$, so we have in mind the $M \to \infty$ limit, which must remain well behaved. The system size $L$, though macroscopically large, it is finite. In symbols:

$M \gg L \gg 1$. Thanks to the interpretation of an overlap between infinite Fermi seas, one standard, and one with a momentum dependent shift $\eta(k)$, and the resulting divergence in $L$, we call the result an orthogonality Catastrophe. (For a continuous $\eta(k)$ there is no catastrophe.)

We reiterate the approximation in (106), and introduce nomenclature

$$\eta_a := \eta\left(\frac{2\pi}{L}a\right). \tag{169}$$

The most essential part of the overlap is a Cauchy determinant, which we present in a double product form analogously to (18)

$$\left(\det_{-M \le a,b \le M} \frac{1}{k_a - q_b}\right)^2 \approx \left(\frac{L}{2\pi}\right)^{4M+2} \frac{\prod_{a<b}(b-a)^2 \prod_{a<b}(b-\eta_b-a+\eta_a)^2}{\prod_{a,b}(b-a+\eta_a)^2}$$

$$= \prod_{m=-M}^{M} \left(\frac{L}{2\pi\eta_m}\right)^2 \prod_{-M\le a<b\le M} \frac{\left(1-\frac{\eta_b-\eta_a}{b-a}\right)^2}{\left(1-\frac{\eta_b}{b-a}\right)^2\left(1+\frac{\eta_a}{b-a}\right)^2}, \tag{170}$$

where in the second equality we have split the bottom product into two sectors and the diagonal $a = b = n$, and divided above and below by all factors $(b-a)$. Let us now also multiply the factors of $2\sin(\pi\eta_n)/L$ from (104) onto (170) to find a central object of interest for this appendix, because the scaling behavior mostly is determined by the elements present in the overlap portion of the quasi-$\tau$-function, $\mathcal{F}$, defined in (122). Before we take the $M \to \infty$ limit, it has the form

$$\mathcal{F} = \left(\det_{-M\le a,b\le M} \frac{\sin(\pi\eta_a)}{\pi(a-b-\eta_a)}\right)^2. \tag{171}$$

In section 6, we argued that a candidate for $\eta(k)$ could be expected to be discontinuous at one point, where we jump from one regime to another around $k = x/t$. This would arise in the scenario where we started with the ansatz of equation (132), and immediately took $\varepsilon = 0$. However, we allow for any other scheme to propose a function $\eta(k)$, and simply now consider it to have a discontinuity at $k = x/t$.

$$\Delta := \lim_{\epsilon \to 0}\left[\eta(x/t+\epsilon) - \eta(x/t-\epsilon)\right]. \tag{172}$$

We therefore scrutinize the effect of a discontinuity $\Delta$ as in (172) in the shift function $\eta(k)$ on its functional the quasi-$\tau$-funcion. If there is no such jump, we may specify it to have a magnitude $\Delta = 0$ at the end and reduce to the smooth result. For now, we assume a finite discontinuity. Without loss of generality we may place it infinitesimally to the right of $k = 0$, between $\eta_0$ and $\eta_1$, and translate it at will by $x/t$ later.

In order to find large $M$ limit of $\mathcal{F}$, we split it into a product of three factors

$$\mathcal{F} = F_1^2 F_2^2 F_3^2, \tag{173}$$

where we define

$$F_1 := \prod_{m=1}^{M} \frac{\sin(\pi\eta_m)}{\pi\eta_m} \prod_{1\le a<b\le M} \frac{1-\frac{\eta_b-\eta_a}{b-a}}{\left(1-\frac{\eta_b}{b-a}\right)\left(1+\frac{\eta_a}{b-a}\right)},$$

$$F_2 := \prod_{m=-M}^{0} \frac{\sin(\pi\eta_m)}{\pi\eta_m} \prod_{-M\le a<b\le 0} \frac{1-\frac{\eta_b-\eta_a}{b-a}}{\left(1-\frac{\eta_b}{b-a}\right)\left(1+\frac{\eta_a}{b-a}\right)}, \tag{174}$$

$$F_3 := \prod_{a=-M}^{0}\prod_{b=1}^{M} \frac{1-\frac{\eta_b-\eta_a}{b-a}}{\left(1-\frac{\eta_b}{b-a}\right)\left(1+\frac{\eta_a}{b-a}\right)},$$

with only $F_3$ containing terms on both sides of the discontinuity. We tackle these expressions by expanding their logarithms, and collecting orders of $\eta_a$. Consider the double product of $F_1$ or $F_2$.

$$
\sum_{a<b} \log\left(1 - \frac{\eta_b - \eta_a}{b - a}\right) - \sum_{a<b} \log\left(1 - \frac{\eta_b}{b - a}\right) - \sum_{a<b} \log\left(1 + \frac{\eta_a}{b - a}\right) =
$$
$$
\sum_{m=1}^{\infty} \frac{1}{m}\left[-\sum_{a<b}\left(\frac{\eta_b - \eta_a}{b - a}\right)^m + \sum_{a<b}\left(\frac{\eta_b}{b - a}\right)^m + \sum_{a<b}\left(-\frac{\eta_a}{b - a}\right)^m\right]. \tag{175}
$$

From which it is apparent by setting $m = 1$ that all terms linear in $\eta_a$ vanish identically. At any finite $M$, these sums are finite and may be rearranged freely.

We turn our attention to the terms at $m > 1$. Firstly, the numerator of the double sum of $F_1$,

$$
-\frac{1}{m}\sum_{1 \leq a < b \leq M}\left(\frac{\eta_b - \eta_a}{b - a}\right)^m = -\sum_{1 \leq a < b \leq M}\left(\frac{2\pi}{L}\right)^2 \frac{1}{m}\left(\frac{\eta\left(\frac{2\pi}{L}b\right) - \eta\left(\frac{2\pi}{L}a\right)}{\frac{2\pi}{L}b - \frac{2\pi}{L}a}\right)^m \left(\frac{2\pi}{L}\right)^{m-2}
$$
$$
\approx -\frac{1}{m}\int_0^{2\pi M/L} dp \int_p^{2\pi M/L} dq \left(\frac{\eta(q) - \eta(p)}{q - p}\right)^m \left(\frac{2\pi}{L}\right)^{m-2}, \tag{176}
$$

after identifying $\frac{2\pi}{L}a \mapsto p$, $\frac{2\pi}{L}b \mapsto q$, and absorbing the $dp, dq$ and the sum into a double integral. This which is legitimate because the integrand has no singularities, specifically, it passes smoothly through $p = q$. We assume these integrals are finite, so it follows that we need only keep the order $m = 2$, as higher orders are negligible for large $L$ due to the powers of $\frac{2\pi}{L}$. After taking the $M \to \infty$ limit, we find,

$$
-\sum_{m=2}^{\infty} \frac{1}{m}\sum_{1 \leq a < b \leq M}\left(\frac{\eta_b - \eta_a}{b - a}\right)^m = -\frac{1}{2}\int_{-\infty}^{0} dp \int_p^0 dq \left(\frac{\eta(q) - \eta(p)}{q - p}\right)^2 + \mathcal{O}\left(\frac{1}{L}\right). \tag{177}
$$

The integrand is symmetric in $p \leftrightarrow q$, which means we may integrate over the whole upper right quadrant of $\mathbb{R}^2$, not just the half $p < q$, at the price of a factor $\frac{1}{2}$. We define the constant

$$
U_1 := \frac{1}{2}\int_{p,q \geq 0} dp\, dq \left(\frac{\eta(q) - \eta(p)}{q - p}\right)^2. \tag{178}
$$

Moving on, we consider the product expansion of the sinc-function, seen before in (31),

$$
\log\left(\frac{\sin(\pi\eta_a)}{\pi\eta_a}\right) = \sum_{n=1}^{\infty} \log\left(1 + \frac{\eta_a}{n}\right)\left(1 - \frac{\eta_a}{n}\right) = -\sum_{m=1}^{\infty} \frac{1}{m}\sum_{n=1}^{\infty}\left[\left(\frac{\eta_a}{n}\right)^m + \left(\frac{-\eta_a}{n}\right)^m\right] \tag{179}
$$

so after expanding the log around 1 in orders of $\eta_a$, it appears the sinc-function terms only contribute at even orders $m$.

Secondly, we ply the $m \geq 2$ terms of the log of the numerator of the double product of $F_1$,

$$
\frac{1}{m}\left[\sum_{1 \leq a < b \leq M}\left(\frac{\eta_b}{b - a}\right)^m + \sum_{1 \leq a < b \leq M}\left(-\frac{\eta_a}{b - a}\right)^m\right]
$$
$$
= \frac{1}{m}\left[\sum_{b=2}^{M}\sum_{a=1}^{b-1}\left(\frac{\eta_b}{b - a}\right)^m + \sum_{a=1}^{M-1}\sum_{b=a+1}^{M}\left(-\frac{\eta_a}{b - a}\right)^m\right] \tag{180}
$$
$$
= \frac{1}{m}\sum_{a=1}^{M}\left[\sum_{n=1}^{a-1}\left(\frac{\eta_a}{n}\right)^m + \sum_{n=1}^{M-a}\left(-\frac{\eta_a}{n}\right)^m\right],
$$

where we have extended the sums over $a$ and $b$ by a single, extremal term in order to unify them in the single outer sum in the final expression, along with substituting $b - a \mapsto \pm n$. We cancel the contributions of the denominator in (180) against part of the sinc function in (179), and collect the remaining sum into the functional

$$d_m[\eta] := \sum_{a=1}^{M} \eta_a^m \sum_{n=a}^{\infty} \frac{1}{n^m}. \tag{181}$$

We employ this object, and the auxiliary $\tilde{\eta}_a := -\eta_{M-a+1}$, resulting in the identity

$$\log F_1 \approx -\frac{U_1}{2} - \sum_{m=2}^{\infty} \frac{d_m[\eta]}{m} - \sum_{m=2}^{\infty} \frac{d_m[\tilde{\eta}]}{m}. \tag{182}$$

If $y_n := y\left(\frac{2\pi}{L}n\right)$ is a smooth function of one variable, and $m > 1$, one can easily show that

$$\sum_{n=1}^{M} \frac{y_n}{n^m} = y_1 \sum_{n=1}^{M} \frac{1}{n^m} + \left(\frac{2\pi}{L}\right)^m \sum_{n=1}^{M} \frac{y_n - y_1}{(2\pi n/L)^m} = y_1 \sum_{n=1}^{\infty} \frac{1}{n^m} + \mathcal{O}\left(\frac{1}{L}\right), \tag{183}$$

as the factor $\left(\frac{2\pi}{L}\right)^m$ kills the integral resulting in the second sum, when we take the limit

$$\lim_{L \to \infty} \left(\frac{2\pi}{L}\right)^{m-1} \int_0^{\frac{2\pi}{L}M} dk \frac{y(k) - y(0)}{k^m} = \lim_{L \to \infty} \left(\frac{2\pi}{L}\right)^{m-1} \int_0^{\frac{2\pi}{L}M} dk \frac{y'(k)}{(1-m)k^{m-1}} = 0, \tag{184}$$

and increasing the upper boundary from $M$ to $\infty$ is trivial in (183) the $M \to \infty$ limit. Adding to this we take an observation borrowed from [105] about what is essentially the *Hurwitz Zeta function*,

$$\sum_{n=M}^{\infty} \frac{1}{n^m} = \frac{1}{(m-1)M^{m-1}} + \mathcal{O}\left(\frac{1}{M^m}\right), \tag{185}$$

we may rewrite back and forth,

$$d_m[\eta] \approx \sum_{a=1}^{M} \frac{\eta_a^m}{(m-1)a^{m-1}} \approx \frac{\eta_1^m}{m-1} \sum_{a=1}^{M} a^{1-m} \approx \eta_1^m \sum_{a=1}^{\infty} \sum_{n=a}^{\infty} \frac{1}{n^m}$$

$$= \eta_1^m \sum_{n=1}^{\infty} \sum_{a=1}^{n} \frac{1}{n^m} = \eta_1^m \sum_{n=1}^{\infty} \frac{1}{n^{m-1}} = \zeta(m-1)\eta_1^m \tag{186}$$

using (185) in the first and third approximation, and (183) in the second. We also extended the upper bound from $a = M$ to $a = \infty$, which is negligible for $m > 2$. The final equality introduces the *Riemann Zeta function* $\zeta$, and is only convergent for $m \geq 3$. A alternative treatment of $d_2[\eta]$ is found by separating it to

$$d_2[\eta] = \sum_{a=1}^{M} \left\{ \eta_a^2 \left(-\frac{1}{a} + \sum_{n=a}^{\infty} \frac{1}{n^2}\right) + \frac{\eta_a^2 - \eta_1^2}{a} + \eta_1^2 \frac{1}{a} \right\}. \tag{187}$$

We approximate the terms one by one. The first,

$$\sum_{a=1}^{M} \eta_a^2 \left(-\frac{1}{a} + \sum_{n=a}^{\infty} \frac{1}{n^2}\right) \approx \eta_1^2 \sum_{a=1}^{M} \left(-\frac{1}{a} + \sum_{n=a}^{\infty} \frac{1}{n^2}\right) =$$

$$\eta_1^2 \left\{ \sum_{a=1}^{M} \left(-\frac{1}{a} + \sum_{n=a}^{M} \frac{1}{n^2}\right) + \sum_{a=1}^{M} \sum_{n=M+1}^{\infty} \frac{1}{n^2} \right\} = \eta_1^2 M \psi^{(1)}(M+1), \tag{188}$$

invoking (183) in the first approximation of (188). We have also used

$$\sum_{a=1}^{M}\sum_{n=a}^{M}\frac{1}{n^2} = \sum_{n=1}^{M}\sum_{a=1}^{n}\frac{1}{n^2} = \sum_{n=1}^{M}\frac{1}{n} \tag{189}$$

in order to cancel the first term in the penultimate expression of (188). The second term is by definition equal to $M$ times the first derivative of the *digamma* function $\psi$.

From here, we invoke the asymptotic expansion at large $M$ of the first derivative of the digamma function,

$$\psi^{(1)}(M) = \sum_{m=0}^{\infty}\frac{B_m}{(M)^{m+1}} \approx \frac{1}{M} + \mathcal{O}\left(\frac{1}{M^2}\right), \tag{190}$$

where $\{B_m\}$ are the *Bernoulli* numbers with $B_0 = 1$, to find that the final expression of (188) has the limit

$$\lim_{M\to\infty}\eta_1^2 M\psi^{(1)}(M+1) = \eta_1^2. \tag{191}$$

Summarizing,

$$\lim_{M\to\infty}\sum_{a=1}^{M}\eta_a^2\left(-\frac{1}{a} + \sum_{n=a}^{\infty}\frac{1}{n^2}\right) \approx \eta_1^2. \tag{192}$$

Next up, with the usual substitution,

$$U_0[\eta] := \sum_{a=1}^{M}\frac{\eta_a^2 - \eta_1^2}{a} \approx \int_0^{\frac{2\pi}{L}M}dk\,\frac{\eta(k)^2 - \eta(0)^2}{k}. \tag{193}$$

And finally,

$$\eta_1^2\sum_{a=1}^{M}\frac{1}{a} = \eta_1^2(\log(M) + \gamma + O(1/M)), \tag{194}$$

where $\gamma \approx 0.57721$ is the Euler-Mascheroni constant.

One can sum up all the elements of equation (182) by using an alternative series expansion of the Barnes-G function, seen earlier in equation (21).

$$\log G(1-z) = \frac{z}{2}(1 - \log 2\pi) - \frac{(1+\gamma)z^2}{2} - \sum_{m=3}^{\infty}\frac{\zeta(m-1)}{m}z^m, \tag{195}$$

which, collecting the results from (186) to here, leads us to

$$-\sum_{m=2}^{\infty}\frac{d_m[\eta]}{m} \approx -\frac{\eta_1^2}{2}\log M + \log G(1-\eta_1) + \frac{\eta_1}{2}(\log 2\pi - 1) - \frac{U_0[\eta]}{2}. \tag{196}$$

Exponentiating (182),

$$F_1^2 \approx \frac{G(1-\eta_1)^2 G(1+\eta_M)^2}{M^{\eta_1^2 + \eta_M^2}}\left(\frac{2\pi}{e}\right)^{\eta_1 - \eta_M}e^{-U_2 - U_0[\eta] - U_0[\tilde{\eta}]}, \tag{197}$$

where we remind ourselves that $\eta_1$ is just to the right of the discontinuity, and for our case of interest $M \gg L \gg 1$, $\eta_M$ is the value at infinity. At this point, we make a crucial demand on $\eta(k)$: it must vanish as $|k| \to \infty$. This is a reasonable choice: it means the matrix in (104) approaches the identity far from the origin, in other words, we may curtail the determinant to a finite domain without considerable inaccuracy. All the physics takes place within some distance of zero momentum. On what scale this occurs is not immediately important and will

in general depend on the shape of the distribution $\rho(q)$ that implicitly defines $\eta(k)$. What is important is that this scale is independent of $M$, so for some large $k \approx \frac{2\pi}{L}M$, we may assume $\eta(k) \approx 0$, or $\eta_M = 0$. This allows us to simplify expression (197).

Although (197) is elegant, we prefer to keep our expressions in terms of sums instead of Barnes functions and integrals. This will facilitate the analysis of $\mathcal{F}$ later on. With this in mind we collect the lessons of (179), (180), (182) and (186) to rewrite $F_1$ as

$$\log F_1 \approx -\frac{U_1}{2} - \frac{1}{2}\sum_{a=1}^{M}\eta_a^2\left(\sum_{n=M-a+1}^{\infty}\frac{1}{n^2} + \sum_{n=a}^{\infty}\frac{1}{n^2}\right) - \sum_{m=3}^{\infty}\frac{\zeta(m-1)}{m}\eta_1^m. \tag{198}$$

Similarly, we can obtain for $F_2$, where we must discard the functionals of $\eta$ in favor of the $\tilde{\eta}$ terms, as well as setting $\eta_{-M} = 0$,

$$\log F_2 \approx -\frac{U_2}{2} - \frac{1}{2}\sum_{a=-M}^{0}\eta_a^2\left(\sum_{n=M+a+1}^{\infty}\frac{1}{n^2} + \sum_{n=1-a}^{\infty}\frac{1}{n^2}\right) - \sum_{m=3}^{\infty}\frac{\zeta(m-1)}{m}(-\eta_0)^m. \tag{199}$$

$\eta_0$ is the value just left of the discontinuity. Additionally, recall the definition of $U_1$ in (178) and compare to the analogous

$$U_2 := \frac{1}{2}\int_{p,q\leq 0}dp\,dq\left(\frac{\eta(q)-\eta(p)}{q-p}\right)^2. \tag{200}$$

In order to evaluate $F_3$ from (174) we follow a similar procedure. The sole difference, due to the discontinuity in $\eta$, is that the cubic and higher terms in the expansion of the log of the numerator do not vanish. Instead, $\eta_a$ can be replaced by the corresponding values at the boundaries, namely for $m \geq 3$

$$\sum_{a=-M}^{0}\sum_{b=1}^{M}\left(\frac{\eta_b-\eta_a}{b-a}\right)^m \approx \sum_{j=1}^{N/2-1}\sum_{i=N/2}^{N}\left(\frac{\Delta}{b-a}\right)^m \approx \zeta(m-1)\Delta^m, \qquad \Delta := \eta_1 - \eta_0. \tag{201}$$

This way, we obtain

$$\log F_3 = -\frac{1}{2}\sum_{a=-M}^{0}\sum_{b=1}^{M}\frac{(\eta_b-\eta_a)^2-\eta_b^2-\eta_a^2}{(b-a)^2} - \sum_{m=3}^{\infty}\frac{\zeta(m-1)}{m}\left(\Delta^m - \eta_1^m - (-\eta_0)^m\right), \tag{202}$$

where the terms involving $\eta_1$ and $\eta_0$ come from the denominator of the definition of $F_1$, in a similar fashion. In order to regularize the quadratic terms into having a well-controlled $M \to \infty$ limit we find the identity

$$\sum_{a=-M}^{0}\sum_{b=1}^{M}\frac{(\eta_b-\eta_a)^2}{(b-a)^2} = \sum_{a=-M}^{0}\sum_{b=1}^{M}\frac{(\eta_b-\eta_a)^2-\Delta^2}{(b-a)^2} + \sum_{a=-M}^{0}\sum_{b=1}^{M}\frac{\Delta^2}{(b-a)^2}. \tag{203}$$

The last term can be evaluated explicitly in terms of derivatives of the digamma function $\psi$, namely, for large $M$,

$$\sum_{a=-M}^{0}\sum_{b=1}^{M}\frac{1}{(b-a)^2} \approx 2\psi(M)-\psi(2M)+2M\psi^{(1)}(M)-2M\psi^{(1)}(2M)+\gamma \approx 1+\gamma+\log\frac{M}{2}, \tag{204}$$

again referencing (190), and noting that $\psi(M) \approx \log M$. The rest of the sum can be transformed into an integral

$$\left(\frac{2\pi}{L}\right)^2\sum_{a=-M}^{0}\sum_{b=1}^{M}\frac{(\eta_b-\eta_a)^2-\Delta^2}{(2\pi(b-a)/L)^2} \approx \int_0^{\frac{2\pi}{L}M}dp\int_{-\frac{2\pi}{L}M}^{0}dq\frac{(\eta(q)-\eta(p))^2-\Delta^2}{(q-p)^2}$$

$$\approx -\Delta^2\log\left(\frac{2\pi}{L}M\right) + U_3, \tag{205}$$

where the last equality is the result of numerous applications of integration by parts, and is only valid in the large $M$ limit. We have invoked another constant,

$$
U_3 := \Delta^2 \log 2 - 2 \int_{-\infty}^{0} dp \int_{0}^{\infty} dq\, \eta'(p)\eta'(q) \log|p-q| +
$$
$$
2 \int_{-\infty}^{0} [\eta(k)-\eta_1]\eta'(k)\log(-k)dk + 2\int_{0}^{\infty} [\eta_0 - \eta(k)]\eta'(k)\log(k)dk. \quad (206)
$$

The rest of the sums that involve quadratic terms should be combined with the corresponding terms from $F_1$ and $F_2$. We take the $\eta_b^2$ terms from (202), relabeling $b \mapsto a$, then $a - b \mapsto n$, together with the quadratic terms from (198) to form

$$
\sum_{a=1}^{M} \eta_a^2 \left( \sum_{n=a}^{a+M} \frac{1}{n^2} - \sum_{n=M-a+1}^{\infty} \frac{1}{n^2} - \sum_{n=a}^{\infty} \frac{1}{n^2} \right) =
$$
$$
-\sum_{a=1}^{M} \eta_a^2 \left( \psi^{(1)}(M-a+1) + \psi^{(1)}(M+a+1) \right) \approx -\sum_{a=1}^{M} \eta_a^2 \left( \frac{1}{M-a+1} + \frac{1}{M+a+1} \right) \quad (207)
$$

again invoking the approximation (190). We follow the same steps for the $\eta_a^2$ terms in (202), substituting $b - a \mapsto n$ and quadratic terms from (199), obtaining

$$
\sum_{a=-M}^{0} \eta_a^2 \left( \sum_{n=1-a}^{M-a} \frac{1}{n^2} - \sum_{n=M+a+1}^{\infty} \frac{1}{n^2} \sum_{n=1-a}^{\infty} \frac{1}{n^2} \right) =
$$
$$
-\sum_{a=-M}^{0} \eta_a^2 \left( \psi^{(1)}(M+a+1) + \psi^{(1)}(M-a+1) \right) \approx -\sum_{a=-M}^{0} \eta_a^2 \left( \frac{1}{M+a+1} + \frac{1}{M-a+1} \right). \quad (208)
$$

Expressions (207) and (208) can be added together in the exponent, and can be approximated by the integral

$$
\lim_{M\to\infty} -\int_{-\frac{2\pi}{L}M}^{\frac{2\pi}{L}M} dk \eta^2(k) \left( \frac{1}{\frac{2\pi}{L}M+k} + \frac{1}{\frac{2\pi}{L}M-k} \right) = 0, \quad (209)
$$

due to vanishing of $\eta(k)$ at both infinities. Also the neglected terms of the expansion of $\psi^{(1)}$ in (190) would vanish, as they feature even larger powers of $M$ in the denominator.

The cubic and higher terms that involve $\eta_0$ and $\eta_1$ cancel manifestly between (202), (198) and (199).

Combining $F_3$ in (202), using (204), (205) and the identities of $F_1$ and $F_2$ in (198) and (199), respectively, taking care to cancel the results of (207) and (208), we finally obtain

$$
\log \mathcal{F} \approx -\Delta^2 \left( 1 + \gamma + \log \frac{M}{2} \right) - 2 \sum_{m=3}^{\infty} \frac{\zeta(m-1)}{m} \Delta^m + \Delta^2 \log \left( \frac{2\pi}{L}M \right) - U_1 - U_2 - U_3. \quad (210)
$$

Taking into account that for a smooth function $\eta(k)$,

$$\int_{z_0}^{z_1} dp \int_{z_0}^{z_1} dq \left( \frac{\eta(q) - \eta(p)}{q - p} \right)^2 = \int_{z_0}^{z_1} dp \int_{z_0}^{z_1} dq \frac{\eta(q)\eta'(p) - \eta'(q)\eta(p)}{q - p} +$$

$$2 \int_{z_0}^{z_1} dk\, \eta'(k) \left[ (2\eta(k) - \eta(z_0)) \log \frac{k - z_0}{z_1 - z_0} - (2\eta(k) - \eta(z_1)) \log \frac{z_1 - k}{z_1 - z_0} \right]. \quad (211)$$

We see that we can rewrite $U_1$ (178) and $U_2$ (200) to

$$U_1 = \frac{1}{2} \int_0^\infty dp \int_0^\infty dq \frac{\eta(q)\eta'(p) - \eta'(q)\eta(p)}{q - p} + \int_0^\infty dk\, \eta'(k)(2\eta(k) - \eta_1)\log(k),$$

$$U_2 = \frac{1}{2} \int_{-\infty}^0 dp \int_{-\infty}^0 dq \frac{\eta(q)\eta'(p) - \eta'(q)\eta(p)}{q - p} - \int_{-\infty}^0 dk\, \eta'(k)(2\eta(k) - \eta_0)\log(-k). \quad (212)$$

Or integrating by parts one more time

$$U_1 = -\int_0^\infty dp \int_0^\infty dq\, \eta'(p)\eta'(q)\log|p - q| + 2\int_0^\infty dk\, \eta'(k)(\eta(k) - \eta_1)\log(k),$$

$$U_2 = -\int_{-\infty}^0 dp \int_{-\infty}^0 dq\, \eta'(p)\eta'(q)\log|p - q| - 2\int_{-\infty}^0 dk\, \eta'(k)(\eta(k) - \eta_0)\log(-k). \quad (213)$$

Comparing to (206), this form allows us to collect $U_1 + U_2 + U_3$ efficiently. Additionally, substituting the series in the $\zeta$-functions along the lines of (195) into (210), we obtain

$$\mathcal{F} \approx \left( \frac{2\pi}{L} \right)^{\Delta^2} G^2(1 - \Delta) \left( \frac{2\pi}{e} \right)^\Delta \times$$

$$\exp\left( -\int dq \int dp\, \eta'(p)\eta'(q)\log|p - q| + 2\Delta \int dk\, \dot{\eta}(k)\log|k| \right) \quad (214)$$

or equivalenly

$$\mathcal{F} \approx \left( \frac{2\pi}{L} \right)^{\Delta^2} G^2(1 - \Delta) \left( \frac{2\pi}{e} \right)^\Delta \times$$

$$\exp\left( -\frac{1}{2} \int dq \int dp \frac{\eta(q)\dot{\eta}(p) - \dot{\eta}(q)\eta(p)}{q - p} + \Delta \int dk\, \dot{\eta}(k)\log|k| \right). \quad (215)$$

Here the derivative $\dot{\eta}$ of a piece-wise continuous function is understood as the derivative of the continuous parts only. For instance, if

$$\eta(k) = y_-(k)\theta(-k) + y_+(k)\theta(k) \Rightarrow \dot{\eta}(k) = y'_-(k)\theta(-k) + y'_+(k)\theta(k), \quad (216)$$

in other words, we omit any Dirac-$\delta$ peaks that would occur at discontinuities.

The effect of (214) scaling as a power law of $L$ is reminiscent of the orthogonality catastrophe of section 3. The lesson is that we cannot take the TDL immediately from the definition of $\tilde{\tau}$, (104), with a discontinuous $\eta$. Instead, we must assume $\eta$ continuous in a fashion that

approaches the discontinuous version in the TDL, controlling the limit. That approach is performed with the ansatz in equation (133). Nonetheless, we feel (214) may be useful to future researchers considering $\eta(k)$ functions that are discontinuous at any system size[35]. For this reason, we have left this more general result here.

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
