# Peer review of "On the Dynamics of Free-Fermionic Tau-Functions at Finite Temperature"

_SciPost Physics Core, doi:SciPost Phys. Core 5, 006 (2022)_

## Round 1 · Referee Report · Anonymous · 2021-11-24

Report
The authors study the \tau-function for free fermions. Before doing so they discuss a simple physical setup, namely a many-particle version of the Aharonov-Bohm problem, where the \tau-function naturally appears. This section is very useful to clarify the definition, motivation and basic properties pf the \tau-function. In the following section they discuss why the \tau-function is hard to calculate by linking it to Andersons orthogonality catastrophe. After that they go on to analyse it using several complementary approaches. While to topic of the paper may seem a bit dry and the paper itself rather lengthy, I believe that such technical articles deserve publication in journals like SciPost Physics Core.
Regarding the presentation, my main concern is with the introduction, since I find the definition of the quantity of interest (1) not easy to understand. For example, the following points remain unclear: (i) Is there any restriction on the integers n_a? (ii) The k_a are defined as specific shifts of the g_a, in particular I deduce that there is precisely one state |k>. Why is there a summation over k then? (iii) It is stated that the summation is over ordered integers (the n_a I suppose) , but the n_a are given by |g>? Some of these points become clear in the following section, but at first the reader may be irritated. Thus I ask the authors to revise this part and make it more self-contained, or at least clearly refer to later parts of the manuscript.
Some further remarks:
-The authors mention that the number of terms in a form-factor expansion grows exponentially and thus cannot be performed at finite temperatures. There have been attempts to overcome this using some stochastic sampling of the Hilbert space [PRB 84, 094203 (2011); PRB 88, 085323 (2013)] which the authors may want to mention.
-It is unclear where (2) comes from. At least it should be pointed out that this formula will be derived later.
-Eqs. (1) and (3) use the same notation for \tau but the RHSs are different quantities.
Author: Daniel Chernowitz on 2022-01-18 [id 2101]
(in reply to Report 1 on 2021-11-24)
We thank the reviewer for their time and attention. They were correct to point out the double use of the n_a in the notation. This has been remedied (the summed over Hilbert space will now be written with m_a as its integers).
We have added a few sentences to help the reader understand where the introductory formulae originate.
We agree with and have incorporated all of the 'further remarks'.
Author: Daniel Chernowitz on 2022-01-18 [id 2100]
(in reply to Report 2 on 2021-12-13)We thank the reviewer for their time and attention.
In regards to the weaknesses, indeed not everywhere have we kept quantitative track of the errors. However, where it is crucial (in the main text) we have estimated the error where it depends on, e.g. the \epsilon parameter which is taken to 0 in the thermodynamic limit.
The conflicting circumlocution and coarseness probably splits down the line separating the two authors. I imagine the 'lack of details' is referring to CH5: written mainly by OG who is the more senior academic. I, DC, might do better to get to the point more quickly.
This is fair feedback.

---

## Round 1 · Referee Report · Anonymous · 2021-12-13

Strengths
1- Thorough study of spectral sums and their asymptotics in the free case, including analytical and numerical approaches
2- Well presented and well explained
Weaknesses
1- Often the sign $\approx$ is used instead of a $O$ notation that would make clear what is neglected
2- There is a strange simultaneous lack of concision and lack of details sometimes in the calculations
Report
The authors study the so-called $\tau$ functions, defined as a spectral sum over "abstract" form factors that have a Cauchy determinant form. This kind of sums arises in computation of correlation functions in certain free models, such as the Tonks-Girardeau gas. It is well known since Korepin and Slavnov's works in the 90s that such spectral sum can be expressed as a single determinant, whose thermodynamic limit is a Fredholm determinant. For interacting models this spectral sum cannot be computed exactly, and various approximation schemes (analytical or numerical) have been developed in the past years, so this is a timely subject. The point of the paper is to use this particular exactly computable case to gain insights on or derive the contributions of each of the terms in the spectral sum. In this respect, sections 6 and 7 are the newest and most relevant.
In section 6 is explained an approach that consists in introducing effective form factors that produce the same thermodynamic limit, but such that only one state contributes to the spectral sum. This was introduced in a previous paper of one of the authors for the equal time case, and is generalized to the dynamical case here. The asymptotic decay of the dynamical $\tau$ function is obtained, including the subleading power-law decay. I find this approach interesting, it seems to me that a more detailed section 6 could have been a paper on its own.
In section 7 is presented in great detail an algorithm to scan the Hilbert space and evaluate numerically (approximately) the spectral sum, in the spirit of the ABACUS algorithm. It will certainly be very useful to people interested in these approaches. However I wonder to which extent this amount of numerical code (around 1/4 of the paper) would be more relevantly displayed in a GitHub file than printed in a physics paper. Besides, this is in the end an approach quite independent from sections 5 and 6, and contributes to make the paper look lengthy. But this is the choice of the authors.
The paper is good and mostly well presented, I recommend publication.

---

## Round 1 · Referee Report · Pieter W. Claeys · 2021-12-29

Strengths
1- Various approaches to calculate the free-fermionic $\tau$-function at finite temperature are proposed from both a theoretical and numerical point of view, all in excellent agreement.
2- Clear and complete description of the calculations and the numerical algorithm, including code.
3- The described algorithm to perform partial summations over the Hilbert space can be generalized to more involved problems.
Weaknesses
1- Section 7.2 can be hard to read.
2- Section 5 is slightly disconnected from the rest of the paper.
Report
In this manuscript the authors calculate free-fermionic $\tau$-functions at a finite temperature. These functions are defined in terms of a Cauchy-like overlap between free-fermionic states where one of the states has a shifted momentum, with a momentum- and energy-dependent phase, and averaged over a statistical ensemble. This averaging involves a summation over the full Hilbert space, making the calculation unfeasible for large system sizes. Such functions appear as correlation functions for Jordan-Wigner strings, and in Section 2 the authors explicitly motivate their $\tau$-function by showing how it appears as a generalized Loschmidt echo in a many-body Aharonov-Bohm setup or as a generating function for the correlation function in the Tonks-Girardeau gas. In Section 3, the authors argue that these systems typically exhibit an "orthogonality catastrophe", such that the necessary number of terms in the summation over the Hilbert space grows dramatically with system size. For the ground state this orthogonality catastrophe consists of the 'diagonal' overlap vanishing as a power law in the number of particles, whereas for the excited states an exponential vanishing is observed (using both theoretical and numerical arguments).
The bulk of the paper then consists of different ways of evaluating the $\tau$-function. In Section 4 an analytic expression is obtained as a Fredholm determinant, first for a single eigenstate (no averaging) and then for a Gibbs ensemble (with averaging). In Section 5 the authors return to the case of zero temperature, and show how a partial summation over 'soft modes' is sufficient to counteract the orthogonality catastrophe for the ground state and return the same asymptotics as the Fredholm determinant expression. In Section 6 a scheme is proposed to replace the full averaging by the evaluation of a single effective form factor that returns the same asymptotics, by replacing the momentum-independent phase shift $\nu$ by a momentum-dependent shift in the obtained expressions. The necessary momentum-dependent shift necessarily exhibits a discontinuity, and the role of this discontinuity and the resulting asymptotics are discussed in detail. Remarkably simple expressions for the scaling of the effective $\tilde{\tau}$-function are obtained, exhibiting exponential decay with an additional power law, and the dependence on temperature and $x/t$ are discussed. A complementary approach is proposed in Section 7, which details an algorithm to perform a partial summation over the Hilbert space — similar to the ABACUS algorithm but for finite-entropy states. All approaches show an excellent agreement.
The paper is written in a very clear way and presents, in my opinion, an important contribution to the literature. Both the theoretical and numerical contributions are valuable in themselves, and it is impressive to see them both combined and confirming each other to such a high degree in this work. Furthermore, while the theoretical results are specific to this problem, the numerical algorithm can be directly extended to more general problems.
I am happy to recommend this paper for publication in SciPost Physics Core. Some minor remarks can be found below, but these are only meant for clarification or as suggestion and not as 'Requested changes'. The authors can choose to address these comments, but I recommend this paper for publication either way.
Requested changes
1- In the introduction, it might be worthwhile to emphasize the importance of $\nu$ when it is introduced, since right now it is easy to read over its introduction.
2- When discussing the orthogonality catastrophe, the scaling of 'diagonal overlaps' between states with the same quantum numbers is discussed, and the scaling with $N$ is derived. It is repeatedly claimed that all other 'off-diagonal' overlaps are necessarily smaller, but is this an exact claim or just intuition? If the former, it would be useful to provide a reference and/or argument.
3- In Figure 3, it might be instructive to also plot the asymptotic expression (22) as a dotted black line.
4- In the derivation of the Fredholm determinant, a summation over $k$ is replaced by a contour integral enclosing the appropriate poles, after which the (fast oscillating) integrand is replaced by its average in order to obtain an expression for the thermodynamic limit. Is there a way to directly perform this averaging starting from Eq. (37) without introducing the contour integral? I'm wondering since the final expression should not depend on the specific form of the integrand provided it has the correct residues.
5- On page 16, the authors write "In keeping with tradition". Some references to the literature would be useful in order to motivate this statement and the following expressions.
6- As mentioned in 'Weaknesses', Section 5 seems to be disconnected from the overall goal of the paper of calculating $\tau$-functions at finite temperature. It presents a rather technical calculation for a partial summation over the Hilbert space, and the bosonization techniques in this section are also disconnected from the bulk of the paper. It might be useful to flag this more explicitly in the manuscript.
7- In Fig. 7, what are the peaks in the right panel and at what values of $k_F x$ do they occur? Is this at $\pm k_Ft$? If so, it might be useful to mark these points on the figure or provide the explicit value of $k_F t$.
9- When discussing the momentum-dependent phase shift, after Eq. (106) the authors write that the shifts can be slightly perturbed into the complex plane. I might be misunderstanding the argument, but this seems to imply that the poles in the integral are also pushed away from the real line. Is it then guaranteed that a contour at distance $\epsilon$ from the real line still encloses all necessary poles in the limit $\epsilon \to 0^+$?
10- The discussion at the end of page 38 is somewhat confusing, e.g." Contrast to being off by 30 − 46% without it, growing with time." It might be worthwhile to slightly extend/clarify this discussion.
11- In section 7.1 it is not clear to me what is meant by "Notably, the infinite suppressed boxes the left and right with the trivial empty filling".
12- While Section 7.2 is undoubtedly extremely useful to anyone implementing the algorithm, it is hard to read and might be better suited to an appendix, since Section 7.1 already explains the algorithm in a readable way. Similarly, I suspect it will be easier for other people to adapt the code if it is made available on GitHub or a similar platform rather than in the text of a manuscript.
13- In the code at the bottom of page 69, a comment reads "Should this be abs or not?", which should probably be answered or removed.
14- Assorted typos: 'spinfull', 'asympototics', 'explicitely', 'Tonks-Giradeau' (twice), 'caluclating', 'the the', 'is could'

---

## Round 3 · Author Response

Individual reply to anonymous report nr 1:

We thank the reviewer for their time and attention. They were correct to point out the double use of the n_a in the notation. This has been remedied (the summed over Hilbert space will now be written with m_a as its integers). We have added a few sentences to help the reader understand where the introductory formulae originate. We agree with and have incorporated all of the 'further remarks'.

Individual reply to anonymous report nr 2:

We thank the reviewer for their time and attention. In regards to the weaknesses, indeed not everywhere have we kept quantitative track of the errors. However, where it is crucial (in the main text) we have estimated the error where it depends on, e.g. the \epsilon parameter which is taken to 0 in the thermodynamic limit. The conflicting circumlocution and coarseness probably splits down the line separating the two authors. I imagine the 'lack of details' is referring to CH5: written mainly by OG who is the more senior academic. I, DC, might do better to get to the point more quickly. This is fair feedback.

Individual reply to report by Pieter Claeys, nr 3:

The authors thank Pieter for his meticulous and fair review. We will address most of his feedback. About weaknesses: Indeed ch5 is somewhat disconnected, however it has been a long-term project of OG and we feel this paper is the best home for the content. It does show a completely separate machinery to arrive at a signal that is the limiting (T=0) case of the main topic of the paper. In that sense it serves as another confirmation of the validity of the techniques in the paper. It is true that a printed paper might not be the best way to distribute Python code, however we feel there is merit in understanding the algorithm on the ground level. Not everyone will be interested in ch 7.2 and it can safely be skipped by those who aren't. No other parts depend on it. In regards to the enumerated list of requested changes, we agree with and have incorporated them straightforwardly, or have cleared up the confusion. Exceptions are addressed below. 2- We have not succeeded in proving (by means of some integer/discrete mathematics) that the diagonal overlaps are largest. It is most certainly true, though, as evinced by a large amount of numerics. We think a proof should be possible. 3- That line was plotted, as the black dots. There is no need for numerical sampling of the ground state, there is only one ground state. The caption was, however, confusing, and we amended it. 4- We believe a contour integral is still the preferred way to address the sum. If anyone has a way to short-circuit the calculation with a smart expansion or identity, please cite us. 7- Indeed, the peaks in both panels (on the left they are not clearly visible) represent singularities in the asymptotic formulas that occur at the "light" cone $x=\pm k_F t$. In reality, these singularities are absent meaning that the asymptotic expansion should be trusted only far from the "light" cones. Our plots demonstrate that, in fact, asymptotic expansion starts to be relevant already in the close vicinity of the light cone. The asymptotic exactly on the light cone is discussed below in chapter 5. As for the numeric position: the time is fixed to $t =100 E_F = 50 k_F^2 $, which means that for $k_F=1$ the singularities are located at $x=\pm 50$. (We have expanded caption of the picture to clarify this fact) 8- This would be too much work to remedy, apologies. 12- We have left ch 7.2 as is, and beg for leniency, as restructuring would probably introduce more errors than is warranted. DC has, however, created a github and linked to it. That is most likely practical for prospective users.

---

## Round 3 · List of Changes

• Signposted the importance of various quantities, such as the shift \nu, the origin of certain expressions such as the definition of \tau, or the roles of certain sections such as chapters 5 and 7, more clearly.
  • Cleared up definition issue in the summation of Hilbert space (n_a -> m_a).
  • Improved the captions of certain figures, sich as figures 3 and 7.
  • Improved references at various points.
  • Clarified the scale on which poles of the effective form factors can deviate from the real line.
  • Improved clarity of some ambiguous formulations.
  • Created and linked to a github for the codebase.
  • Removed assorted typos.

---

## Editorial Decision

published